palaeontology, evolution

Palaeocene, mammals, locomotion, end-Cretaceous, extinction, evolution

**Author for correspondence:**
Sarah L. Shelley
e-mail: sarah.shelley@ed.ac.uk

# Quantitative assessment of tarsal morphology illuminates locomotor behaviour in Palaeocene mammals following the end-Cretaceous mass extinction

Sarah L. Shelley[1,2], Stephen L. Brusatte[1] and Thomas E. Williamson[3]

[1]School of Geosciences, University of Edinburgh, Edinburgh, UK
[2]Section of Mammals, Carnegie Museum of Natural History, Pittsburgh, USA
[3]New Mexico Museum of Natural History and Science, Albuquerque, NM, USA

SLS, 0000-0001-6628-0226

Mammals exhibit vast ecological diversity, including a panoply of locomotor behaviours. The foundations of this diversity were established in the Mesozoic, but it was only after the end-Cretaceous mass extinction that mammals began to increase in body size, diversify into many new species and establish the extant orders. Little is known about the palaeobiology of the mammals that diversified immediately after the extinction during the Palaeocene, which are often perceived as 'archaic' precursors to extant orders. Here, we investigate the locomotor ecology of Palaeocene mammals using multivariate and disparity analyses. We show that tarsal measurements can be used to infer locomotor mode in extant mammals, and then demonstrate that Palaeocene mammals occupy distinctive regions of tarsal morphospace relative to Cretaceous and extant therian mammals, that is distinguished by their morphological robustness. We find that many Palaeocene species exhibit tarsal morphologies most comparable with morphologies of extant ground-dwelling mammals. Disparity analyses indicate that Palaeocene mammals attained similar morphospace diversity to the extant sample. Our results show that mammals underwent a post-extinction adaptive radiation in tarsal morphology relating to locomotor behaviour by combining a basic eutherian bauplan with anatomical specializations to attain considerable ecomorphological diversity.

## 1. Introduction

Mammals have evolved an array of locomotor behaviours, from cursoriality in ungulates and branch-swinging in primates, to deep-diving in cetaceans and powered flight in bats [1]. The foundations of this ecomorphological diversity were established in the Mesozoic, in a miscellany of species distantly related to extant crown mammal groups [2]. Following the Cretaceous–Palaeogene (K-Pg) mass extinctiona at 66 Ma, therian mammals diversified to fill empty niches, rapidly evolving into new species [3,4] and increasing in body size [5,6]. It therefore stands to reason that much of the locomotor diversity of extant mammals emerged during this post-extinction transition, the classic example upon which the theory of adaptive radiations was developed [7]. However, relatively little is known about the ecomorphological diversity and palaeobiology of the mammals that thrived after the extinction.

The nature of mammal diversification following the K-Pg extinction is contentious. Most extant placental mammal orders cannot be traced back in the fossil record to the interval immediately following the extinction. Instead, the early

Palaeogene (Palaeocene) saw the proliferation of so-called 'archaic' eutherians, largely known from the northern hemisphere fossil record, whose relationships to extant groups are poorly resolved and whose palaeobiology is difficult to study because most species are poorly represented, often solely by teeth, and lack any obvious extant analogues [8]. Palaeocene eutherians have often been typecast as 'archaic' and have confounded researchers since the nineteenth century. Historically they have been regarded as 'inadaptive forms' [9], derided as 'arrested or persistently archaic in structure' [10]. Some groups are considered a 'generalized' stock from which the extant placental orders evolved [8,11]. They are critical for understanding the evolution of mammals as they constituted a significant proportion of mammal diversity during much of the first third of the Cenozoic.

Anatomical and functional insights on Palaeocene mammals have previously either been limited to qualitative inferences [12–15], or focused on a specific extinct taxon [16–18], or a small and/or closely related sample of extinct taxa [19,20]. These findings challenge the concept that Palaeocene mammals were non-specialized generalists and indicate a wider range of morphological diversity than previously alluded. However, no study has investigated the diversity of locomotor ecologies exhibited by a larger sample of Palaeocene mammals. Locomotion is strongly tied to many aspects of organismal behaviour and substrate preference; it is therefore a valuable ecological proxy for the selectivity of the K-Pg extinction and subsequent faunal recovery and provides insight into the Cenozoic radiation of eutherian mammals. Here, we investigate the locomotor ecology of Palaeocene eutherians, in comparison to a sample of Cretaceous cladotherians and extant therian mammals.

## 2. Methods

### (a) Dataset

We compiled a dataset comprising 29 tarsal measurements: 17 astragalar and 12 calcaneal (electronic supplementary material, figure S1). Tarsals allow for accurate locomotor inferences [21,22] and are commonly preserved as fossils, allowing for the incorporation of a broad sample of Palaeocene taxa that better encapsulate a range of morphological diversity. Tarsal measurements were taken for 85 extant therian mammals representing 20 orders (electronic supplementary material, dataset S1). We used a broad extant taxonomic sample, with representatives from every pertinent extant order, to capture the variations in morphology associated with different locomotor behaviours and reduce phylogenetic influence. We compiled data for 40 extinct Palaeocene eutherian mammals and 5 Cretaceous cladotherian mammals. Where appropriate fossil taxa were composited to allow them to be included in the analyses (electronic supplementary material, dataset S1). A key novelty of our study derives from the broad taxonomic sample of 40 Palaeocene species, including many taxa which have not been well-described in the recent literature or quantitatively analysed. Furthermore, the inclusion of many Palaeocene taxa allows for quantifiable comparisons and differentiation between taxa within the functional context provided by the extant sample and the temporal context provided by including the Cretaceous taxa. Measurements of extant taxa were taken directly from specimens using digital callipers. Measurements of extinct taxa were taken from specimens, photographs and published literature using ImageJ [23].

Extant taxa were classified into six locomotor groups (locomotor definitions and classifications are provided in the electronic

supplementary material). We used tarsal measurements, allowing us to incorporate a broad sample of Palaeocene taxa, but note that Palaeocene taxa often exhibit an amalgam of morphologies in their skeletons [12,13,19,24]. Future studies incorporating other parts of the skeleton will be informative, as well as combining phylogenetic and multivariate data, allowing for locomotor diversity to be investigated through the Palaeocene at a finer temporal scale.

### (b) Multivariate analyses

It is widely recognized that morphometric shape data can be strongly correlated with body size [25]. In this study, we are interested in tarsal shape, minimizing the influence of size. To achieve this, the data were Box-Cox transformed and subsequently standardized by way of a $z$-transformation using the geometric mean (electronic supplementary material, dataset S1).

We conducted principal component analysis (PCA) on the variance–covariance matrix using the prcomp function in R to determine the linear combination of variables that maximizes variance. For the PCA visualization, the data were grouped by Cretaceous, Palaeocene and extant subsets (additional visualization of placental versus marsupial morphospaces are available in the electronic supplementary material).

Two discriminant analyses (DAs) were conducted in R using the MASS [26] and klaR packages [27]. Linear discriminants analysis (LDA) was used to visualize the combination of variables that maximizes separation among the extant species when grouped by locomotor mode. The fossil taxa were then superimposed onto the morphospace based on the criteria for maximal separation as determined by the extant data. A regularized discriminant analysis (RDA) model was used to predict fossil locomotor classification. Error rates associated with the RDA optimization parameters were estimated using cross-validation and bootstrapping. The principal component (PC) scores and the discriminant functions (DFs) were exported for further analyses (electronic supplementary material, dataset S1).

### (c) Statistical analyses

We used PERMANOVA tests to assess whether groups were significantly separated in morphospace. A PERMANOVA test was run on the 29 PCs derived from the PCA (electronic supplementary material, dataset S1). Taxa were grouped into Cretaceous, Palaeocene and extant subsets. Additional comparisons to assess the separation between Palaeocene species and extant marsupial and placental mammals are also provided in the electronic supplementary material. Probability values were adjusted for multiplicity using the Benjamini and Yekutieli false discovery rate method [28]. Throughout this study, statistically significant differences are denoted when $p < 0.005$, with $p < 0.05$ but greater than 0.005 being suggestive of significance. In addition, we have supplemented all probability values in this study with an upper Bayes factor or Bayes factor bound (BFB) (electronic supplementary material, equation 2), data-based odds representing the strongest case for the alternative hypothesis relative to the null hypothesis [29–31]. A probability value of 0.005 corresponds to a BFB of 13.88 (values are in odds to one, so 13.88 : 1). PERMANOVA tests were also run using the same protocol on the DFs for the extant locomotor groups and for the Palaeocene taxa, divided into a locomotor group according to the RDA classification. Calculations were conducted in R using the 'pairwiseAdonis' package [32].

We assessed the effects of phylogeny on the extant taxa included in our morphospaces using Blomberg's K statistic. We were not able to include the extinct taxa in this assessment given the lack of well-resolved phylogeny for Palaeocene eutherians. We downloaded 10 000 birth-death node-dated trees from the www.vertlife.org database for our sample of extant mammals. A 50% majority-rule consensus tree was generated using the ape

package in R [33] (electronic supplementary material, dataset S2). Blomberg's K statistic was then calculated for the first three axes of the PC and DF morphospaces using the phylosig function in the phytools package [34]. Probability values were calculated with 100 000 randomizations. A statistically insignificant phylogenetic signal along an axis is expected when $p < 0.005$ (the phylogenetic and multivariate data are significantly different).

## (d) Disparity analyses

We used the PCs and DFs to calculate morphological disparity. Taxa were grouped into Cretaceous, Palaeocene and extant subsets (additional placental versus marsupial comparisons are available in the electronic supplementary material). We quantified morphospace occupation using the sum of ranges (= overall spread of species in morphospace), the sum of variances (= average dissimilarity among species in morphospace relative to the group mean) and mean pairwise distance (= average density of species in morphospace relative each other). Disparity metrics, for all 29 PC axes and 5 DF axes, were calculated in R using the 'dispRity' package v. 1.41 (electronic supplementary material, dataset S3) [35]. To allow for between-group comparisons, the data were bootstrapped with 200 pseudoreplicates [36], and confidence intervals were generated for disparity within each subset. To mitigate sampling biases, the data were fully rarefied at all sample sizes between the minimum and maximum number of taxa in the group, and the mean of those values was taken.

Three statistical tests were used to compare disparity across groups. A Wilcoxon rank-sum test with pairwise comparisons was used to assess whether group mean ranks differ. The Bhattacharyya coefficient was calculated with pairwise comparisons to quantify the probability of overlap between group distributions. Both tests were conducted in R using the 'dispRity' package. Finally, we used permutation tests to further assess whether the difference in disparity values between groups was statistically significant (electronic supplementary material, dataset S4). The test calculates the statistical significance of the difference between the observed disparity of the two groups under consideration (calculated without resampling) and the expected disparity (empirical mean of 10 000 resampled disparity values) of the two groups, so as to stringently account for differences in sample size [37]. Calculations were conducted in R. Each disparity metric was calculated as before, but without resampling, using the 'dispRity' package.

The full rationale for our methods is provided in the electronic supplementary material.

# 3. Results

## (a) Multivariate analyses

PCA ordinated our mammal sample into a morphospace with the first three axes accounting for 45.36% of the total variance (figure 1; electronic supplementary material, figures S4–S8 and tables S2–S3). The PCA maximizes variance between all species based on their tarsal morphology, with the morphospaces illustrating how Palaeocene taxa differ from extant and Cretaceous taxa.

PC1 (21.40% of the total variance) arrays taxa along a spectrum of decreasing tarsal robustness and increasing tarsal stability. Tarsal robustness is captured by the mediolateral proportions of the astragalus and calcaneum. Tarsal stability relates to the loss of rotational movement between the astragalus and calcaneum from low to high PC1 scores. A detailed description of the measurements and functional inferences for the first three PC axes is provided in the electronic supplementary material, figures S4–S8. PC2 (12.46%) arrays taxa

by increasing capability of multiaxial movement at the cruropedal joint coupled with less efficient but more rapid lever action of the calcaneum. PC3 (11.50%) arrays taxa by how loads are dissipated through the tarsus with decreasing PC3 values being associated with more equal weight distribution through the ectal and sustentacular facets.

Relative to the extant sample, Palaeocene species are widely dispersed in PC morphospace (figures 1 and 2). They are differentiated from extant placental mammals in that they possess a comparatively more robust and mobile astragalolcalcaneal articulation (electronic supplementary material, figures S4–S8). In this regard, they are more similar to our extant marsupial sample. The Palaeocene taxa differ from the extant marsupials by possessing a variably mobile to more stable cruropedal joint and a range of relative ectal and sustentacular facet proportions more comparable to the extant placental sample.

Statistical tests of PC morphospace separation demonstrate that Palaeocene taxa are significantly separated from the extant taxa and the Cretaceous species across all 29 PC axes (PERMANOVA $p = 0.0005$ [BFB = 89.12] and 0.0045 [BFB = 14.91], respectively) (electronic supplementary material, table S4). Furthermore, the Cretaceous species are significantly separated from the extant sample (PERMANOVA $p = 0.0008$, BFB = 62.81). The Palaeocene sample is also statistically separated from the extant sample when the extant sample is differentiated into marsupial and placental species (PERMANOVA $p < 0.005$) (electronic supplementary material, table S5). The Cretaceous sample, however, is not well separated from the extant marsupial sample (PERMANOVA $p = 0.018$, BFB = 5.00). Our PC morphospace contains no significant phylogenetic signal (PC1 K = 0.0985, $p = 0.00005$ [BFB = 3195.36], PC2 K = 0.0651, $p = 0.00149$ [BFB = 37.93], PC3 K = 0.0860, $p = 0.0004$ [BFB = 399.42]).

DA with extant taxa grouped by locomotor mode ordinated Cretaceous, Palaeocene and extant species into morphospace, with the first three axes accounting for 91.90% of the total variance (figure 1; electronic supplementary material, figures S9–S11 and table S6). The DF morphospace differs to the PC morphospace in that the extant taxa are ordinated with the supervised input of their predetermined locomotor mode. In ordinating the extinct species to best fit the extant data, the DA effectively demonstrates which tarsal morphologies are comparable between Palaeocene and extant mammals with less emphasis on how they differ.

DF1 (67.02% of the total variance) arrays taxa by morphologies concomitant with stance. Low DF scores are associated with a fully plantigrade stance, progressing through a heel-elevated stance to digitigrady and unguligrady as DF values increase. A detailed description of the measurements and functional inferences for the first three DF axes is provided in the electronic supplementary material, figures S9–S11. DF2 (16.92%) captures the force and speed capability of the lever action of the calcaneum with the astragalus at the cruropedal joint with movements increasing from a relatively more rapid but less forceful lever at the low end of DF2 towards more forceful and efficient movements at the higher end of DF2. DF3 (7.96%) arrays taxa by rotational movement between the astragalus and calcaneum from habitual eversion (low DF3 scores) through to habitual inversion (high DF3 scores). This axis is similar to PC1 but differs in that morphologies associated with habitual eversion and inversion are distinguished along a single axis. Our DF

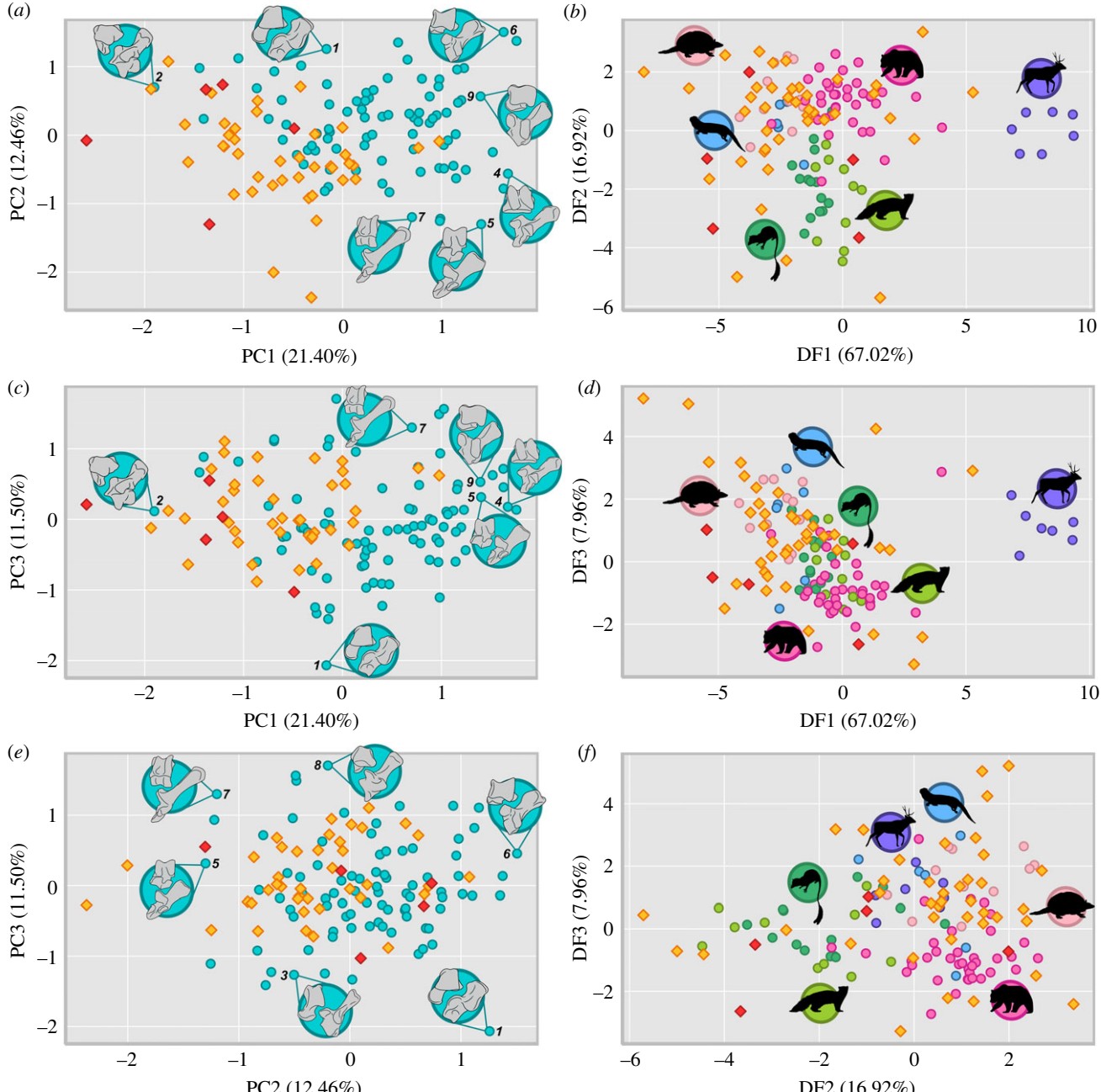

**Figure 1.** Morphospace occupation of Palaeocene mammals relative to extant and Cretaceous mammals derived from tarsal measurements. (*a*) PC1 versus PC2, (*b*) DF1 versus DF2, (*c*) PC1 versus PC3, (*d*) DF1 versus DF3, (*e*) PC2 versus PC3 and (*f*) DF2 versus DF3. Extant species are denoted by circular datapoints, extinct species are denoted by diamond datapoints. In PC morphospace, taxa are grouped by time subset with Cretaceous (red), Palaeocene (yellow) and extant (blue) groups. Exemplar tarsals: *Alouatta* (1), *Didelphis* (2), *Eira* (3), *Equus* (4), *Hexaprotodon* (5), *Hystrix* (6), *Odocoileus* (7), *Thylacinus* (8), *Ursus* (9). In DF morphospace, extant taxa are grouped by locomotor mode: arboreal (dark-green), cursorial (purple), scansorial (yellow-green), semi-aquatic (light-blue), semi-fossorial (light-pink) and terrestrial (hot-pink). (Online version in colour.)

morphospace contains limited significant phylogenetic signal (DF1 K = 0.145, *p* = 0.00001 [BFB = 3195.36], DF2 K = 0.073, *p* = 0.00059 [BFB = 83.85], DF3 K = 0.0612, *p* = 0.00588 [BFB = 12.18]). DF1 and DF2 contain no substantial phylogenetic influence. DF3, which accounts for 7.96% of the total variance contains a low but not insubstantial phylogenetic signal.

Relative to the extant sample, Palaeocene mammals are widely dispersed in DF locomotor morphospace, occupying and extending beyond all sampled extant locomotor regions, with the exception of the extant cursorial group (figures 1 and 2; electronic supplementary material, figure S11). Generally, they are more heavily concentrated in areas of DF morphospace associated with a plantigrade to heel-

elevated stance and forceful pedal movements. They are widely distributed along DF3 indicating a range of medial to lateral movement and loading through the astragalus and calcaneum.

RDA classifies 91.765% of extant species correctly into pre-determined locomotor groups, thus validating the tarsal dataset as a practical proxy for inferring locomotor mode (electronic supplementary material, tables S7–S9). Cross-validation and bootstrapping of the optimization parameters result in greater average misclassification rates with a 42.125% and 47.869% misclassification rate, respectively. However, incorrect classifications are explained by overlapping locomotor behaviour or phylogeny, as evidenced by the posterior

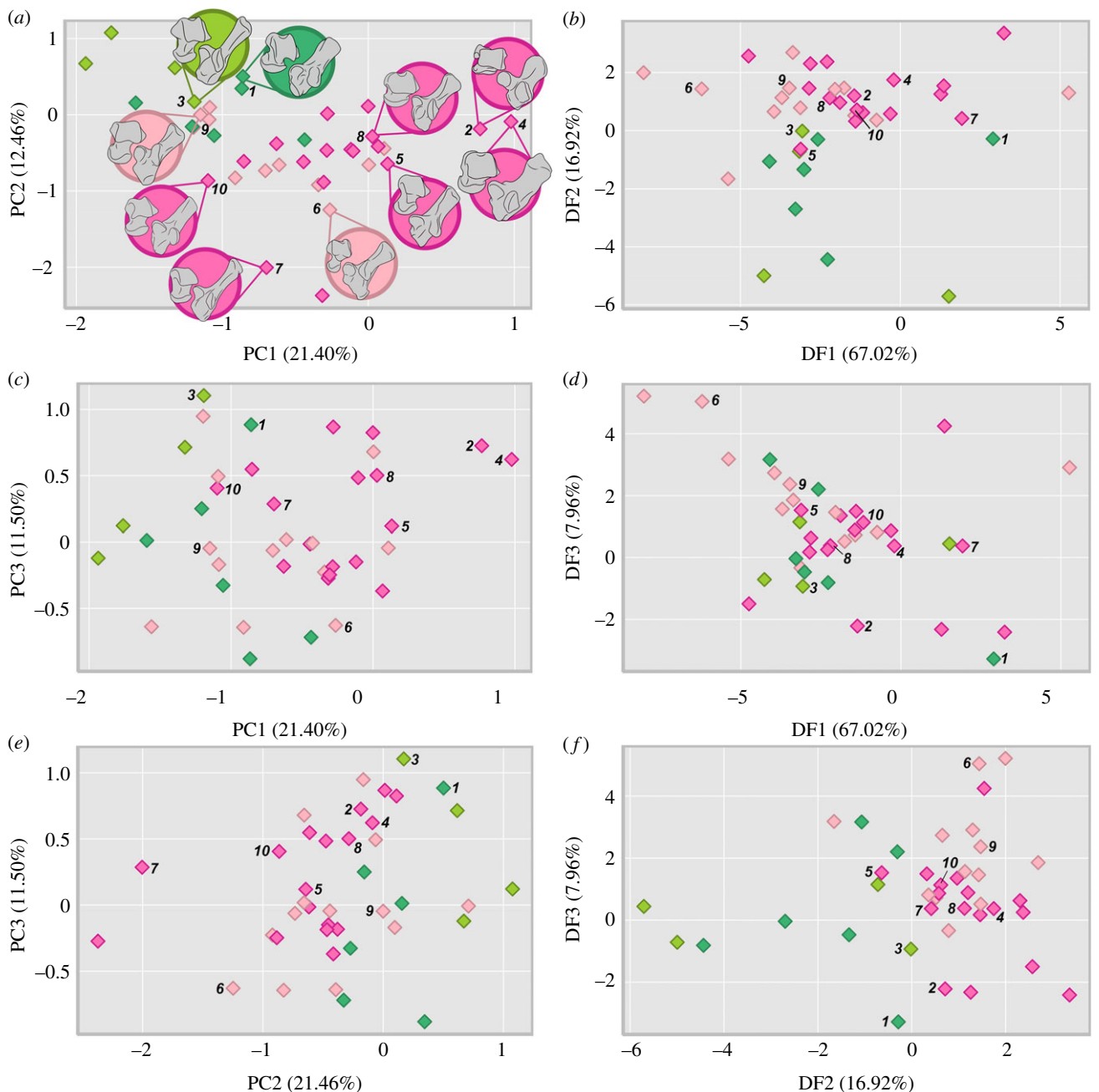

**Figure 2.** Morphospace occupation of Palaeocene mammals with locomotor group as predicted by the RDA (*a*) PC1 versus PC2, (*b*) DF1 versus DF2, (*c*) PC1 versus PC3, (*d*) DF1 versus DF3, (*e*) PC2 versus PC3 and (*f*) DF2 versus DF3. Exemplar tarsals: *Arctocyon* (1), *Cardonia* (2), *Colbertia* (3), *Ectoconus* (4), *Goniacodon* (5), *Orthaspidotherium* (6), *Pachyaena* (7), *Periptychus* (8), *Procerberus* (9), *Tetraclaenodon* (10). Datapoint coloration corresponds to inferred locomotor mode: arboreal (dark-green), cursorial (purple), scansorial (yellow-green), semi-aquatic (light-blue) and semi-fossorial (light-pink), terrestrial (hot-pink) (see electronic supplementary material, figure S12 for a labelled version). (Online version in colour.)

values (electronic supplementary material, tables S7–S9). We used the RDA model to predict the locomotor modes of the extinct mammals in our dataset (figure 2; electronic supplementary material, figure S12 and table S10). A plurality of the Palaeocene species was classified as terrestrial (19) and semi-fossorial (13). The remaining Palaeocene species were inferred to be arboreal (9 species) and scansorial (4). Within the Palaeocene sample, there is variable statistical separation in DF morphospace between extinct taxa grouped by their inferred locomotor behaviour, indicating that these groups have somewhat distinctive morphologies (figure 2; electronic supplementary material, figure S12 and table S13). It is worth noting that these distinctions are not

as numerically significant as those found between the equivalent extant locomotor groups (electronic supplementary material, table S15).

In sum, the PCA demonstrates that Cretaceous, Palaeocene and extant mammals (as a whole and when differentiated into marsupials and placentals) all have distinctive tarsal morphologies. Despite their distinctiveness in PC morphospace, the LDA visualization and RDA classification show that the fossil species share similar tarsal functions to extant mammals in terms of stance and movement and loading through the tarsus but are often exploiting locomotor strategies using different combinations of morphologies distinguished by their underlying robustness. These unusual

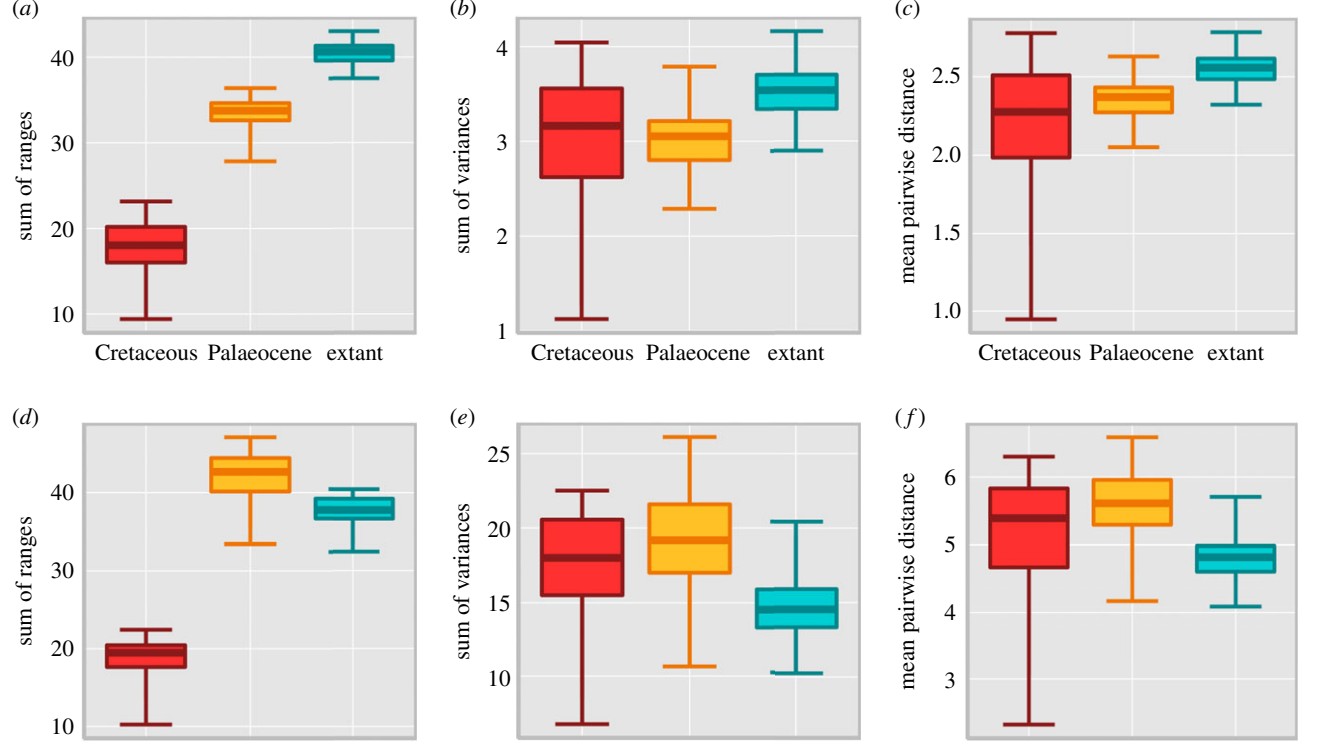

**Figure 3.** Morphological disparity calculated for PC and DF scores with Cretaceous (red), Palaeocene (yellow), extant (blue) and 95% confidence intervals. (*a*) PC sum of ranges, (*b*) PC sum of variances, (*c*) PC mean pairwise distance, (*d*) DF sum of ranges, (*e*) DF sum of variances and (*f*) DF mean pairwise distance. (Online version in colour.)

anatomical combinations are defined by their robustness and lack of stability, as illustrated by the PC morphospace.

## (b) Disparity analyses

Palaeocene mammals have considerable morphological disparity in their tarsals (figure 3). Statistical tests of PC morphospace occupation (WRS tests and Bhattacharyya coefficients on sum of ranges disparity values) show that Palaeocene taxa occupy a significantly smaller area of PC morphospace compared to our extant sample, but a larger area compared to Cretaceous forms, with a little statistical overlap in their distributions (figure 3*a*; electronic supplementary material, table S14). However, when tested further with a permutation test that more stringently accounts for sample size differences and distributions, Palaeocene taxa do not occupy a significantly smaller area of PC morphospace compared to the extant sample (electronic supplementary material, table S14 and figure S16). The permutation result for the Palaeocene versus Cretaceous comparison also does not support a statistically significant difference between these two groups ($p < 0.005$; electronic supplementary material, table S14 and figure S16). When PC morphospace occupation is quantified using the sum of variance disparity to describe dissimilarity within groups relative to their mean, Palaeocene taxa exhibit lower average disparity than both the extant and Cretaceous mammals (figure 3*b*; electronic supplementary material, table S14). However, no comparisons are statistically significant when subjected to permutation tests (electronic supplementary material, table S14 and figure S16). When PC morphospace occupation is quantified using mean pairwise distances to describe average dissimilarity within groups as a measure of datapoint density, Palaeocene taxa exhibit

marginally greater average dissimilarity compared to Cretaceous taxa and marginally lower average dissimilarity compared to the extant sample (figure 3*c*; electronic supplementary material, table S14). The results of the WRS indicate a significant difference between the group means (with the exception of the Palaeocene versus Cretaceous comparison); however, the permutation tests do not maintain this significance (electronic supplementary material, table S14).

Statistical tests of observed DF morphospace occupation (WRS and Bhattacharyya coefficients on the sum of ranges disparity values sum of ranges disparity) show that Palaeocene taxa occupy a significantly larger area of DF morphospace compared to both Cretaceous and extant mammals sampled (figure 3*d*; electronic supplementary material, table S14). A permutation test on the Palaeocene versus extant comparison also supports this finding (electronic supplementary material, table S14 and figure S17). The sum of variance disparity shows Palaeocene taxa exhibit the highest dissimilarity within groups relative to their mean compared to both the extant and Cretaceous mammals sampled (WRS $p < 0.005$; figure 3*e*; electronic supplementary material, table S14). However, the permutation test does not find the difference between Palaeocene and extant mammals to be statistically significant (electronic supplementary material, table S14 and figure S17). Palaeocene mammals also exhibit the highest disparity when quantified using mean pairwise distances (WRS $p < 0.005$; figure 3*f*; electronic supplementary material, table S14). The Palaeocene versus extant comparison is again statistically significant when subjected to a permutation test (electronic supplementary material, table S14 and figure S17).

These disparity trends are largely upheld when comparing the Palaeocene sample to the extant placental sample (electronic supplementary material, figures S15, S18–S23, and

table S15). However, the extant marsupial sample is less spread out in PC and DF morphospace (lower sum of range disparity values) and exhibits high dissimilarity between forms (variance disparity) in PC morphospace following appropriate adjustment for sampling.

## 4. Discussion

The Palaeocene followed a catastrophic mass extinction and the abolition of complex and highly structured ecosystems dominated by dinosaurs, and heralded the beginning of the Age of Mammals. The Palaeocene mammal fauna is often perceived from two philosophical standpoints. On the one hand, it is the product of classic adaptive radiation marked by the proliferation of eutherian mammals more 'advanced' than their Cretaceous antecedents [5,7,38]. On the other hand, many Palaeocene mammals have been regarded as 'primitive' and 'archaic' forms compared with their extant relations, a view especially prominent in the historical literature [9–11]. Seldom are Palaeocene mammal fauna considered on their own objective merit.

Our results show that the prevailing archaic typecast of Palaeocene mammals is a misconception. Instead, Palaeocene mammals are characterized by robust limb proportions compared to extant mammals (electronic supplementary material, figure S24 and dataset S5) and occupy their own distinct area in PC morphospace defined by morphologies that are relatively more mobile, but were also experimenting with more efficient and forceful tarsal movements as shown in the DF morphospace (figures 1 and 2; electronic supplementary material, figures S4–S11). We hypothesize that the distinctive robustness of Palaeocene mammals is related to a lack of prominent osseous stabilizers in their joint anatomy (e.g. relatively weak to little trochleation of the astragalus in animals considered to be otherwise more terrestrially adapted, lack of tightly interlocking articulations). Instead they were more reliant on soft tissues to stabilize their joints, which is often manifested by prominent soft tissue attachment areas (e.g. [12,13,17,24]). This feature of stabilizing through soft tissues may have resulted from, and been beneficial to, rapid evolution of larger body sizes and novel proportions to exploit empty ecospace following the K-Pg mass extinction. Furthermore, it is the subsequent development of osseous stabilizers in the mostly post-Palaeocene members of waning 'archaic' and emergent extant orders that provide key phylogenetic characters that readily diagnose and differentiate taxa, such as the double pulley astragalus of Artiodactyla [39]. This partially explains why the phylogeny of Palaeocene taxa has been so difficult to resolve.

Palaeocene mammals exhibited a high diversity of tarsal morphologies, as shown by their notable range of morphospace occupation when visualized using both PCA and DA ordination techniques (figures 1 and 2). Our results show Cretaceous cladotherians included in our sample were restricted to somewhat smaller and separate regions of morphospace and are inferred to have been arboreal or terrestrial (electronic supplementary material, table S10). This contrasts with the greater variety of locomotor styles of non-cladotherian Mesozoic mammals with which cladotherians likely competed [2,40,41]. Following the K-Pg mass extinction, eutherians diversified in the Palaeocene, resulting in a greater morphospace occupation and an increase in observed range

disparity (figure 3; electronic supplementary material, figures S15–S23). These disparity comparisons are limited by the small Cretaceous sample available, although this limitation may not entirely be the result of sampling bias but simply reflect the observed increase in species richness after the extinction [3]. Furthermore, the distribution of the sample of Cretaceous species in PC and DF morphospace, although constrained compared to Palaeocene and extant mammals in terms of size (range disparity), still indicates considerable tarsal shape diversity (variance disparity) and differing locomotor strategies within similar niches (figure 3; electronic supplementary material, figures S15–S23).

We hypothesize that the increase in the spread of taxa in morphospace (sum of range disparity), from the Cretaceous into the Palaeocene, was the consequence of a post-extinction adaptive radiation. This inference is consistent with trends of increasing species richness and a greater range of body sizes in Palaeocene mammals [3–6], although changes in dietary ecology may have been delayed until the Eocene [3,42]. Given the resolution of our study, we are unable to determine whether the increase in locomotor disparity occurred suddenly after the extinction or more gradually across the Palaeocene. However, the wide dispersion of Puercan-aged species (66.043–63.3 Ma) in morphospace suggests that locomotor diversification was an early-burst trait (figure 2; electronic supplementary material, figure S12 and dataset S1).

Observed variance disparity does not show the same increase from the Cretaceous in the Palaeocene as range disparity (figure 3; electronic supplementary material, figure S15–S23). This disjunction implies that mammals attained diversity in tarsal shape prior to the extinction, during the Cretaceous, and although they may have been exploiting similar (and/or fewer) niches (electronic supplementary material, table S10), they were doing so in different ways. We note that although our data has been corrected for sampling bias, there may be a phylogenetic and temporal component to this diversity, but the lack of fossil material limits further investigation at present. Following the K-Pg mass extinction, mammals evolved from a bottleneck [43–45] and underwent a shift in tarsal shape morphology, increasing their spread (range) in tarsal shape morphospace but to a lesser extent their dissimilarity (variance).

When comparing the results of the disparity analysis on the PC versus the DF scores, the Palaeocene group exhibits higher DF disparity compared to the PC data (figure 3; electronic supplementary material, figure S15). Differences in tarsal morphology between the Palaeocene and our extant mammal samples, as exemplified by the PCA, result in the DA plotting a substantial number of the Palaeocene species on the periphery of the extant data or as statistical outliers (figure 1; electronic supplementary material, table S16) resulting in higher disparity values. This distribution is explained by Palaeocene species co-opting unusual morphologies defined by a robust but mobile tarsus but with adaptations towards more efficient movement, which is different to both extant placental and marsupial mammals. We use an RDA to predict locomotor groups for the Palaeocene species based on their tarsal morphology, but do not intend for these classifications to be definitive given how divergent some of the Palaeocene tarsal morphologies are (electronic supplementary material, table S16). They do, however, provide a quantitative means for inferring tarsal function based on explicit comparison to extant species. We emphasize that our classifications

should be taken in conjunction with our morphospaces and the existing literature (electronic supplementary material, table S17). See our electronic supplementary material for further discussion on fossil classifications.

A substantial number of Palaeocene taxa were classified as terrestrial (20 species) or semi-fossorial (16 species) by the RDA (figure 2; electronic supplementary material, figure S12 and table S10) which raises two hypotheses. Given the predictive capability of the RDA model, there is a reasonable likelihood that ground-dwelling locomotor strategies were key adaptive traits for surviving and thriving after the extinction. Burrowing capabilities were probably advantageous for surviving the initial impact [46]. Furthermore, palaeobotanical records of the K-Pg event show an approximately 50% loss in plant diversity over the extinction boundary and a fern spike in the interval immediately following the impact [47,48], indicating a substantial reduction of arboreal habitat which may have favoured more ground-dwelling mammals while the flora recovered. It is therefore plausible that following the extinction, Palaeocene mammals diversified from a small, ground-dwelling bauplan (retaining some traits of that morphotype) into a range of morphologically robust forms under the unique selective pressures of the Palaeocene environment [49–51]. Note that this hypothesis of a robust post-extinction survivor morphotype is decoupled from the concept of the primitive eutherian condition morphotype (a small, scansorial to arboreal insectivore that lived prior to the K-Pg mass extinction) [52,53].

Alternatively, given the uncertainty of the predictive model used to classify the fossil taxa, the tarsal morphologies of the Palaeocene mammals sampled might have been convergent with extant terrestrialists and semi-fossors because they are similarly morphologically robust, and therefore the Palaeocene species may not have necessarily exhibited similar locomotor behaviours. When comparing the distribution of extant versus Palaeocene locomotor groups in morphospace there is a substantial separation between extinct and extant mammals when grouped by locomotor behaviour. For example, Palaeocene taxa classified as terrestrial are significantly separated from extant terrestrial mammals in both PC and DF morphospace (PERMANOVA $p < 0.005$; electronic supplementary material, figures S13–S14 and table S13). In this scenario that Palaeocene species exhibit convergent morphologies with extant terrestrialists and semi-fossors, it might have been the robust morphologies themselves and associated reliance on soft tissues for joint stabilization that promoted survival and diversification around the extinction. This trait of morphological robustness might also not have been unique to eutherians. Other mammal clades, specifically the morphologically robust multituberculates but also monotremes, metatherians and gondwanatheres, survived and radiated following the K-Pg extinction event [44,54,55]. Testing the survivorship of morphologically robust mammals using homologous robusticity metrics would be an informative avenue for future work to understand the complexity of faunal recovery following the K-Pg mass extinction, but currently would likely be impeded by a limited fossil record.

We hypothesize that reliance on soft tissues for joint stabilization may have been physiologically beneficial to eutherians during the Palaeocene. Soft tissue-stabilized joints could be adapted more rapidly to meet functional requirements of movement given that post-natal myogenesis parallels embryonic development allowing for the functional development of soft tissue [56], whereas post-natal osteogenesis is restricted to remodelling without the functional development of stabilizing features [57]. Myological modifications would have likely favoured the development of osteological adaptations over time; however, adaptable, soft tissue-stabilized joint anatomies may have been advantageous for survival during an adaptive radiation (in conjunction with other physiological and adaptive traits such as changes in growth and body size and exploitation of new dietary niches, [3,6]).

Overall, Palaeocene mammals exploited a considerable area of PC and DF morphospace compared to extant mammals. This finding is particularly notable because the Palaeocene was dominated by a more homogeneous global environment than the present day, with subtropical and tropical rainforest biomes extending to high latitudes [49–51]. The high morphological disparity in these more globally uniform ecosystems emphasizes the intensity of the adaptive radiation that was unfolding. Following the Palaeocene–Eocene thermal maximum, the diversity of habitats increased in concert with the development of increasingly drier and cooler climates and more open and fragmented landscapes [51,58]. Over time, this environmental change may have been less conducive to the survivorship of robust mammals adapted to dense tropical forest environments, perhaps explaining why many Palaeocene groups succumbed to extinction.

In summary, Palaeocene mammals are significantly dissimilar from both Cretaceous and extant mammals in their skeletal robusticity and tarsal morphology and exhibit a high disparity of locomotion-related tarsal morphologies, thus indicating considerable ecomorphological diversity. Far from being generalized ancestral stock for extant mammal orders, Palaeocene mammals were experimenting with tarsal anatomies, combining a basic placental bauplan [59] with numerous specializations, which exemplifies the ability of mammals to adapt and evolve following catastrophic environmental upheaval.

Data accessibility. Data is provided as part of the electronic supplementary material. These include: Electronic Supplementary Material PDF containing supplementary text, figures and tables. Dataset S1 Tarsal Dataset. Separate file formatted as an Excel file (.xlsx) and includes four sheets: sheet 1, tarsal measurements and associated metadata; sheet 2, Z-transformed tarsal data; sheet 3, Principal Component scores; sheet 4, Discriminant Function scores. Dataset S2: Phylogenetic dataset. Separate file formatted as a Nexus file (.nex) containing the 50% majority-rule consensus tree derived from the 10 000 birth-death node dated trees downloaded from the www.vertlife.org database. Dataset S3: Disparity code. Separate file formatted as Rez source code for opening in R or as a text file and includes the disparity code used in this study. Dataset S4: Permutation code. Separate file formatted as Rez source code for opening in R or as a text file and includes the permutation code used in this study. Dataset S5 Regression Dataset. Separate file formatted as an Excel file (.xlsx) and includes three sheets: sheet 1, humerus robustness index values; sheet 2, femur robustness index values; sheet 3, calcaneum robustness index values.

Authors' contributions. S.L.S.: conceptualization, data curation, formal analysis, investigation, methodology, resources, visualization, writing-original draft, writing-review and editing; S.L.B.: methodology, supervision, writing-review and editing; T.E.W.: resources, supervision, writing-review and editing

All authors gave final approval for publication and agreed to be held accountable for the work performed therein.

Competing interests. We declare we have no competing interests

**Funding.** This study was funded by Marie Curie Career Integration Grant, Natural Environment Research Council, National Science Foundation (DEB 1654949, DEB 1654952 and EAR 1325544) and H2020 European Research Council (PalM 756226).

**Acknowledgements.** We thank C. Argot, J. Galkin, A. Millhouse and Z. Timmons for access to specimens. Thanks also go to the reviewers who provided helpful feedback on previous versions of this manuscript which greatly improved the final version. We also thank J. Wible for constructive comments and discussion and P. dePolo and Z. Kynigopoulou for assistance with data collection. We would also like to acknowledge and thank www.phylopic.org for providing content used in our figures; individual credits are provided in the supplementary PDF.

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
