## [Peer Review File · Proceedings of the Royal Society B: Biological Sciences]

Review History

RSPB-2020-0494.R0 (Original submission)

Review form: Reviewer 1 (Peter Bishop)

Recommendation

Major revision is needed (please make suggestions in comments)

Scientific importance: Is the manuscript an original and important contribution to its field?

Acceptable

General interest: Is the paper of sufficient general interest?

Good

Quality of the paper: Is the overall quality of the paper suitable?

Acceptable

Is the length of the paper justified?

Yes

Should the paper be seen by a specialist statistical reviewer?

Yes

Do you have any concerns about statistical analyses in this paper? If so, please specify them explicitly in your report.

Yes

It is a condition of publication that authors make their supporting data, code and materials available - either as supplementary material or hosted in an external repository. Please rate, if applicable, the supporting data on the following criteria.

Is it accessible?

Yes

Is it clear?

Yes

Is it adequate?

Yes

Do you have any ethical concerns with this paper?

No

Comments to the Author

See attached comments file, and also attached annotated manuscript. (See Appendix A)

Review form: Reviewer 2

Recommendation

Major revision is needed (please make suggestions in comments)

Scientific importance: Is the manuscript an original and important contribution to its field?

Excellent

General interest: Is the paper of sufficient general interest?

Good

Quality of the paper: Is the overall quality of the paper suitable?

Acceptable

Is the length of the paper justified?

Yes

Should the paper be seen by a specialist statistical reviewer?

No

Do you have any concerns about statistical analyses in this paper? If so, please specify them explicitly in your report.

No

It is a condition of publication that authors make their supporting data, code and materials available - either as supplementary material or hosted in an external repository. Please rate, if applicable, the supporting data on the following criteria.

Is it accessible?

Yes

Is it clear?

Yes

Is it adequate?

Yes

Do you have any ethical concerns with this paper?

No

Comments to the Author

This manuscript makes important contributions to our understanding of both mammalian functional morphology and Paleocene mammalian ecology. The manuscript demonstrates that a discriminant function analysis (DFA) of linear tarsal measurements can distinguish functional groups among modern placental mammals with substantial accuracy. When the resulting discriminant function is used to classify Paleocene mammals, sampled taxa are recovered as either semi-fossorial or (non-cursorial) terrestrial, a somewhat surprising conclusion in at least a few cases (e.g., *Purgatorius*, recently considered arboreally adapted). This manuscript has profound positive implications for our ability to infer locomotor ecology from isolated tarsal remains and for our understanding of locomotor diversity and disparity in early Cenozoic mammals. Unfortunately, these important results are obscured by more problematic aspects of the authors' study. Substantial revisions are required before this manuscript is ready for publication. Major concerns are described below. The attached files detail additional issues with the main and supplemental text.

1. The manuscript places too much emphasis on the results of a principal components analysis (PCA), at the expense of the DFA. PCA is informative in visualizing a morphospace and is best used for data exploration. It recovers successive combinations of variable loadings that account for the most variance in the overall sample, without regard to user-defined categories. In contrast, a DFA looks for the combination of variable loadings that best sorts samples into user-defined categories. In the case of this study, the discriminant analysis finds the combination of factor loadings that best sorts the taxa into locomotor categories, regardless of the contribution of that variation to overall variation.

In fact, when I graph CF1 versus CF2, the DFA does an impressive job of discriminating functional categories among extant placentals. Aside from locomotor categories that blend into each other (scansorial and arboreal) or are frequently noted as having extensive functional overlap (semifossorial and semiaquatic), locomotor groups are well separated by the DFA. When Paleocene taxa are added, a few forms extend somewhat beyond the modern data, but most taxa fall within the envelope formed by extant placentals. In contrast, the PCA is much messier. Along PC1 and PC2, Fig. S2a demonstrates extensive overlap between every locomotor category aside from terrestrial. It certainly doesn't look like something that could be used to confidently infer locomotor preferences in mammals of unknown habits. Making it worse is the fact that there is very little overlap between Paleocene and modern mammals. Most Paleocene mammals fall outside of the morphospace occupied by any locomotor group.

Given this contrast, the DFA results should be front and center in the manuscript. Instead, the results and discussion place far too much emphasis on the PCA results. For instance, the authors describe and interpret the variables with the largest loadings along PC1-3, but never indicate what variables have the largest loadings along CF1-3. Those are going to be the variables that are most relevant to assessing locomotor function. Additionally, an extensive discussion of morphological disparity is predicated on the assumption that morphological disparity is reflective of locomotor diversity. However, the PCA results demonstrate that overall morphological disparity is poorly correlated with locomotor diversity. It might be worth looking at disparity in variables with substantial loadings in CF1 and CF2, but overall morphological

disparity doesn't appear to be reflective of locomotor diversity.

2. The sample of Paleocene eutherians included needs to be expanded. The sample used in this study is not representative of the taxonomic diversity of Paleocene mammals known from published tarsal remains. For instance, there are nine "condylarths", but only one plesiadapiform. Why are Plesiadapis, Dryomomys, Carpolestes, and Ignacius excluded? Several major groups for which the tarsus has been described are excluded entirely (e.g., Pseudictopidae, Arctostylopidae, Dinocerata, Adapisoriculidae, Oxyaenidae). As a result, the documented diversity/disparity of Paleocene eutherian tarsals is not reflected by the sample used in this manuscript. This casts doubt on the authors' conclusions concerning diversity and disparity in Paleocene eutherian tarsal morphology and locomotor ecology.

In addition, some additional accounting for which specific taxa were chosen should be provided. I can understand not including every condylarth, but why were specific taxa chosen and not others? Why are peripitychids represented by both Peripitychus and Ectoconus but not Hemithlaeus? Why are the Tiupampa condylarths excluded?

3. Sampled material of two included taxa (Colbertia and Carodnia) come from a locality (Itaboraí) that consensus now considers to be Eocene in age (e.g., Gelfo et al., 2009; Woodburne et al., 2014; Antoine et al., 2015; Rangel et al., 2019). Carodnia does occur in faunas that are considered Paleocene (the "zona de Carodnia" in Patagonia), although tarsals of the Paleocene species are not known. Colbertia is not known from Paleocene faunas. What is the justification for including either taxon? With regard to Carodnia, if you are including tarsals from Eocene representatives of genera with Paleocene species, why not Azygonyx, Apheliscus, Haplomylus, Chriacus, or Didymictis?

4. I don't think there's enough Cretaceous material available to support the role it's been given in this study. Of the five Cretaceous taxa included, only Deccanolestes is known from the last 10 million years of the Cretaceous. The remainder includes three taxa from more than 25 ma before the K/T boundary, two of which are non-eutherians. I don't see how you can meaningfully compare Paleocene eutherians with their Cretaceous antecedents on this basis.

5. The "size + shape" analyses don't add anything to the manuscript and should be removed. Aside from confirming the well-known maxim that PC1 represents size in an untransformed PCA, the "size + shape" analyses don't tell us anything beyond what "shape"-only analyses already revealed. The "size + shape" DFA actually performs slightly worse than the "shape"-only DFA in correctly classifying modern taxa. Bizarrely, it flips the largest early Cenozoic taxon in the analysis (Carodnia) from terrestrial (which seems fairly reasonable) to arboreal (which does not).

6. The three axis graphs in Figure 2 are visually appealing but very difficult to read, undermining the utility of the figure. Visualizing three axes at once can be valuable when the graph is interactive and the viewer can manipulate and rotate the image. When they're compressed into a static, 2-D image, as they are here, they're simply difficult to interpret. For instance, part 2a doesn't convey how much separation there is between functional groups because the Y and Z axes are smooshed together. A 3D plot is visually more interesting than bivariate plots, but scientifically, it's much less useful. Figs. 2a-b should be replaced with bivariate plots, while 2c, showing results of the "size + shape" analysis, should be dropped entirely.

7. The introduction and discussion lean too heavily on papers that range from 30 to over 100 years old and do not adequately represent current thinking on Paleocene mammals. Aside from Rose (2009), citations supporting the authors' claims that Paleocene mammals are viewed as "primitive" or "archaic" are uniformly outdated. Paleocene mammals are not as well understood as Eocene and younger mammals, largely due to more limited material, but I don't think it's fair to claim that they are presently viewed as primitive or uniformly generalized.

References cited in this review that are not cited in the manuscript:

Antoine, P.-O., G. Billet, R. Salas Gismondi, J. V. Tejada-Lara, P. Baby, S. Brusset, and N. Espurt. 2015. A new *Carodnia* Simpson, 1935 (Mammalia, Xenungulata) from the early Eocene of northwestern Peru and a phylogeny of xenungulates at species level. *Journal of Mammalian Evolution* 22:129-140.

Gaudin, T. J., R. J. Emry, and J. R. Wible. 2009. The phylogeny of living and extinct pangolins (Mammalia, Pholidota) and associated taxa: a morphology based analysis. *Journal of Mammalian Evolution* 16:235-305.

Gelfo, J. N., F. J. Goin, M. O. Woodburne, and C. d. Muizon. 2009. Biochronological relationships of the earliest South American Paleogene mammalian faunas. *Palaeontology* 52:251-269.
Rangel, C. C., L. M. Carneiro, L. P. Bergqvist, É. V. Oliveira, F. J. Goin, and M. J. Babot. 2019. Diversity, affinities and adaptations of the basal sparassodont *Patene* (Mammalia, Metatheria). *Ameghiniana* 56:263-289.

Szalay, F. S. 1994. *Evolutionary History of the Marsupials and an Analysis of Osteological Characters*. Cambridge University Press, Cambridge, 481 pp.

Thewissen, J. G. M. 1991. Limb osteology and function of the primitive Paleocene ungulate *Pleuraspidotherium* with notes on *Tricuspidon* and *Dissacus*. *Geobios* 24:483-495.

Woodburne, M. O., F. J. Goin, M. Sol Raigemborn, M. Heizler, J. N. Gelfo, and É. V. Oliveira. 2014. Revised timing of the South American early Paleogene land mammal ages. *Journal of South American Earth Sciences* 54:109-119.

Review form: Reviewer 3

Recommendation

Accept with minor revision (please list in comments)

Scientific importance: Is the manuscript an original and important contribution to its field?

Excellent

General interest: Is the paper of sufficient general interest?

Excellent

Quality of the paper: Is the overall quality of the paper suitable?

Good

Is the length of the paper justified?

Yes

Should the paper be seen by a specialist statistical reviewer?

No

Do you have any concerns about statistical analyses in this paper? If so, please specify them explicitly in your report.

Yes

It is a condition of publication that authors make their supporting data, code and materials available - either as supplementary material or hosted in an external repository. Please rate, if applicable, the supporting data on the following criteria.

Is it accessible?

Yes

Is it clear?

Yes

Is it adequate?

Yes

Do you have any ethical concerns with this paper?

No

Comments to the Author

Please see attached document. (See Appendix B)

Decision letter (RSPB-2020-0494.R0)

18-May-2020

Dear Dr Shelley:

I am writing to inform you that your manuscript RSPB-2020-0494 entitled "Unique locomotor habits in Palaeocene mammals after the end-Cretaceous mass extinction" has, in its current form, been rejected for publication in Proceedings B.

This action has been taken on the advice of referees, who have recommended that substantial revisions are necessary. With this in mind we would be happy to consider a resubmission, provided the comments of the referees are fully addressed. However please note that this is not a provisional acceptance.

To upload a resubmitted manuscript, log into <http://mc.manuscriptcentral.com/prsb> and enter your Author Centre, where you will find your manuscript title listed under "Manuscripts with

Decisions." Under "Actions," click on "Create a Resubmission." Please be sure to indicate in your cover letter that it is a resubmission, and supply the previous reference number.

Sincerely,
Dr Locke Rowe
mailto: proceedingsb@royalsociety.org

Associate Editor
Board Member: 1
Comments to Author:

Thank you for the opportunity to review this paper. Apologies to authors for the time taken to reach this point. We had a difficult time finding reviewers, probably reflecting the current disruption to working routines. However, the three reviewers have provided both timely and exhaustive comments on the paper, and I hope these are helpful going forward.

It is my opinion that this paper has promise, but needs considerable re-evaluation before it is ready for Proceedings B. The two aspects that jumped out at me on reading the paper were the lack of explicit consideration of body size in the analysis, and the apparent discrepancy (lack of overlap) in morphospace between extant and extinct animals that are ultimately concluded to be of the same locomotor ecology. These issues are noted by at least one (sometimes two) of the reviewers, in addition to a number of other potential statistical issues. Despite these issues, all three reviewers remain positive about the potential contribution this study could make to field. It is therefore my recommendation that the authors be given the opportunity to produce a revised version of the study that addresses the issues raised in the reviews, in particular the comments on the statistical analyses. Competition for space in Proceedings B is extremely intense and so acceptance of the paper would rely not only a robust re-evaluation of the statistics but also a strong take-home message surviving this re-evaluation.

Reviewer(s)' Comments to Author:

Referee: 1
Comments to the Author(s)
See attached comments file, and also attached annotated manuscript.

Referee: 2
Comments to the Author(s)
This manuscript makes important contributions to our understanding of both mammalian functional morphology and Paleocene mammalian ecology. The manuscript demonstrates that a discriminant function analysis (DFA) of linear tarsal measurements can distinguish functional groups among modern placental mammals with substantial accuracy. When the resulting discriminant function is used to classify Paleocene mammals, sampled taxa are recovered as either semi-fossorial or (non-cursorial) terrestrial, a somewhat surprising conclusion in at least a few cases (e.g., *Purgatorius*, recently considered arboreally adapted). This manuscript has profound positive implications for our ability to infer locomotor ecology from isolated tarsal remains and for our understanding of locomotor diversity and disparity in early Cenozoic mammals. Unfortunately, these important results are obscured by more problematic aspects of the authors' study. Substantial revisions are required before this manuscript is ready for publication. Major concerns are described below. The attached files detail additional issues with the main and supplemental text.

1. The manuscript places too much emphasis on the results of a principal components analysis (PCA), at the expense of the DFA. PCA is informative in visualizing a morphospace and is best used for data exploration. It recovers successive combinations of variable loadings that account for the most variance in the overall sample, without regard to user-defined categories. In contrast, a DFA looks for the combination of variable loadings that best sorts samples into user-defined categories. In the case of this study, the discriminant analysis finds the combination of

factor loadings that best sorts the taxa into locomotor categories, regardless of the contribution of that variation to overall variation.

In fact, when I graph CF1 versus CF2, the DFA does an impressive job of discriminating functional categories among extant placentals. Aside from locomotor categories that blend into each other (scansorial and arboreal) or are frequently noted as having extensive functional overlap (semifossorial and semiaquatic), locomotor groups are well separated by the DFA. When Paleocene taxa are added, a few forms extend somewhat beyond the modern data, but most taxa fall within the envelope formed by extant placentals. In contrast, the PCA is much messier.

Along PC1 and PC2, Fig. S2a demonstrates extensive overlap between every locomotor category aside from terrestrial. It certainly doesn't look like something that could be used to confidently infer locomotor preferences in mammals of unknown habits. Making it worse is the fact that there is very little overlap between Paleocene and modern mammals. Most Paleocene mammals fall outside of the morphospace occupied by any locomotor group.

Given this contrast, the DFA results should be front and center in the manuscript. Instead, the results and discussion place far too much emphasis on the PCA results. For instance, the authors describe and interpret the variables with the largest loadings along PC1-3, but never indicate what variables have the largest loadings along CF1-3. Those are going to be the variables that are most relevant to assessing locomotor function. Additionally, an extensive discussion of morphological disparity is predicated on the assumption that morphological disparity is reflective of locomotor diversity. However, the PCA results demonstrate that overall morphological disparity is poorly correlated with locomotor diversity. It might be worth looking at disparity in variables with substantial loadings in CF1 and CF2, but overall morphological disparity doesn't appear to be reflective of locomotor diversity.

2. The sample of Paleocene eutherians included needs to be expanded. The sample used in this study is not representative of the taxonomic diversity of Paleocene mammals known from published tarsal remains. For instance, there are nine "condylarths", but only one plesiadapiform. Why are Plesiadapis, Dryomomys, Carpolestes, and Ignacius excluded? Several major groups for which the tarsus has been described are excluded entirely (e.g., Pseudictopidae, Arctostylopidae, Dinocerata, Adapisoriculidae, Oxyaenidae). As a result, the documented diversity/disparity of Paleocene eutherian tarsals is not reflected by the sample used in this manuscript. This casts doubt on the authors' conclusions concerning diversity and disparity in Paleocene eutherian tarsal morphology and locomotor ecology.

In addition, some additional accounting for which specific taxa were chosen should be provided. I can understand not including every condylarth, but why were specific taxa chosen and not others? Why are periptychids represented by both Periptychus and Ectoconus but not Hemithlaeus? Why are the Tiupampa condylarths excluded?

3. Sampled material of two included taxa (Colbertia and Carodnia) come from a locality (Itaboraí) that consensus now considers to be Eocene in age (e.g., Gelfo et al., 2009; Woodburne et al., 2014; Antoine et al., 2015; Rangel et al., 2019). Carodnia does occur in faunas that are considered Paleocene (the "zona de Carodnia" in Patagonia), although tarsals of the Paleocene species are not known. Colbertia is not known from Paleocene faunas. What is the justification for including either taxon? With regard to Carodnia, if you are including tarsals from Eocene representatives of genera with Paleocene species, why not Azygonyx, Apheliscus, Haplomyilus, Chriacus, or Didymictis?

4. I don't think there's enough Cretaceous material available to support the role it's been given in this study. Of the five Cretaceous taxa included, only Deccanolestes is known from the last 10 million years of the Cretaceous. The remainder includes three taxa from more than 25 ma before the K/T boundary, two of which are non-eutherians. I don't see how you can meaningfully compare Paleocene eutherians with their Cretaceous antecedents on this basis.

5. The “size + shape” analyses don’t add anything to the manuscript and should be removed. Aside from confirming the well-known maxim that PC1 represents size in an untransformed PCA, the “size + shape” analyses don’t tell us anything beyond what “shape”-only analyses already revealed. The “size + shape” DFA actually performs slightly worse than the “shape”-only DFA in correctly classifying modern taxa. Bizarrely, it flips the largest early Cenozoic taxon in the analysis (*Carodnia*) from terrestrial (which seems fairly reasonable) to arboreal (which does not).
6. The three axis graphs in Figure 2 are visually appealing but very difficult to read, undermining the utility of the figure. Visualizing three axes at once can be valuable when the graph is interactive and the viewer can manipulate and rotate the image. When they’re compressed into a static, 2-D image, as they are here, they’re simply difficult to interpret. For instance, part 2a doesn’t convey how much separation there is between functional groups because the Y and Z axes are smooshed together. A 3D plot is visually more interesting than bivariate plots, but scientifically, its much less useful. Figs. 2a-b should be replaced with bivariate plots, while 2c, showing results of the “size + shape” analysis, should be dropped entirely.
7. The introduction and discussion lean too heavily on papers that range from 30 to over 100 years old and do not adequately represent current thinking on Paleocene mammals. Aside from Rose (2009), citations supporting the authors claims that Paleocene mammals are viewed as “primitive” or “archaic” are uniformly outdated. Paleocene mammals are not as well understood as Eocene and younger mammals, largely due to more limited material, but I don’t think its fair to claim that they are presently viewed as primitive or uniformly generalized.

References cited in this review that are not cited in the manuscript:

- Antoine, P.-O., G. Billet, R. Salas Gismondi, J. V. Tejada-Lara, P. Baby, S. Brusset, and N. Espurt. 2015. A new *Carodnia* Simpson, 1935 (Mammalia, Xenungulata) from the early Eocene of northwestern Peru and a phylogeny of xenungulates at species level. *Journal of Mammalian Evolution* 22:129-140.
- Gaudin, T. J., R. J. Emry, and J. R. Wible. 2009. The phylogeny of living and extinct pangolins (Mammalia, Pholidota) and associated taxa: a morphology based analysis. *Journal of Mammalian Evolution* 16:235-305.
- Gelfo, J. N., F. J. Goin, M. O. Woodburne, and C. d. Muizon. 2009. Biochronological relationships of the earliest South American Paleogene mammalian faunas. *Palaeontology* 52:251-269.
- Rangel, C. C., L. M. Carneiro, L. P. Bergqvist, É. V. Oliveira, F. J. Goin, and M. J. Babot. 2019. Diversity, affinities and adaptations of the basal sparassodont *Patene* (Mammalia, Metatheria). *Ameghiniana* 56:263-289.
- Szalay, F. S. 1994. *Evolutionary History of the Marsupials and an Analysis of Osteological Characters*. Cambridge University Press, Cambridge, 481 pp.
- Thewissen, J. G. M. 1991. Limb osteology and function of the primitive Paleocene ungulate *Pleuraspidothierium* with notes on *Tricuspidon* and *Dissacus*. *Geobios* 24:483-495.
- Woodburne, M. O., F. J. Goin, M. Sol Raigemborn, M. Heizler, J. N. Gelfo, and É. V. Oliveira. 2014. Revised timing of the South American early Paleogene land mammal ages. *Journal of South American Earth Sciences* 54:109-119.

Referee: 3

Comments to the Author(s)

Please see attached document.

Author's Response to Decision Letter for (RSPB-2020-0494.R0)

See Appendices C & D.

RSPB-2020-2330.R0

Review form: Reviewer 1 (Peter Bishop)

Recommendation

Accept with minor revision (please list in comments)

Scientific importance: Is the manuscript an original and important contribution to its field?

Excellent

General interest: Is the paper of sufficient general interest?

Good

Quality of the paper: Is the overall quality of the paper suitable?

Excellent

Is the length of the paper justified?

Yes

Should the paper be seen by a specialist statistical reviewer?

No

Do you have any concerns about statistical analyses in this paper? If so, please specify them explicitly in your report.

No

It is a condition of publication that authors make their supporting data, code and materials available - either as supplementary material or hosted in an external repository. Please rate, if applicable, the supporting data on the following criteria.

Is it accessible?

Yes

Is it clear?

Yes

Is it adequate?

Yes

Do you have any ethical concerns with this paper?

No

Comments to the Author

I am thoroughly pleased with the revised manuscript. The authors have done an excellent job in addressing my comments, concerns and queries, and I think the new version is much stronger and more suitable for publication. Again, the figures are superb.

I have attached the manuscript with some very minor grammatical/punctuation suggestions annotated throughout, and I encourage the authors to carefully run their eye over the supplementary text as well in this regard. I do not feel the need to see any further revisions (should they be required), and I look forward to seeing the paper in print.

Review form: Reviewer 2

Recommendation

Major revision is needed (please make suggestions in comments)

Scientific importance: Is the manuscript an original and important contribution to its field?

Good

General interest: Is the paper of sufficient general interest?

Good

Quality of the paper: Is the overall quality of the paper suitable?

Acceptable

Is the length of the paper justified?

No

Should the paper be seen by a specialist statistical reviewer?

No

Do you have any concerns about statistical analyses in this paper? If so, please specify them explicitly in your report.

Yes

It is a condition of publication that authors make their supporting data, code and materials available - either as supplementary material or hosted in an external repository. Please rate, if applicable, the supporting data on the following criteria.

Is it accessible?

Yes

Is it clear?

Yes

Is it adequate?

Yes

Do you have any ethical concerns with this paper?

No

Comments to the Author

The revised version of this manuscript is a significant improvement over the first draft. The inclusion of an expanded Paleocene sample is a welcome addition, and it paints a clearer picture of the distribution of Paleocene eutherians in tarsal morphospace. The manuscript now also strikes a more appropriate balance between the PCA and DFA analyses. However, I think there are still significant issues that need to be addressed, largely centered on the validity of the locomotor predictions made by the DFA. I've outlined my primary concerns below, with additional comments included in the attached pdfs.

1. While the DFA does an impressive job of separating extant taxa by locomotor groups, I'm still concerned by how many fossil taxa fall outside of the discriminant morphospace defined by extant taxa. Many of the "arboreal" and "semifossorial" taxa, in particular, seem to be well outside the envelope defined by all extant taxa, which suggests that their morphology may be too divergent for locomotor patterns to be accurately inferred based on the morphology of extant taxa. Even when fossil taxa do fall within the extant morphospace, many are classified into groups that seem nonintuitive, at least based on the DF1 versus DF2 plot. The "terrestrial" taxa, for instance, include several taxa (Protungulatum, Goniacodon, Vincelestes) that look like they would be more appropriately classified as semifossorial or semiaquatic based on where they plot in discriminant function morphospace.

I think you need to investigate whether it's appropriate to use DFA to classify unknowns they fall well outside the range of the known comparative sample. As great a job as the discriminant functions do of accurately classifying extant eutherians, I'm not sure it's an appropriate sample for classifying Cretaceous and Paleocene mammals. You wouldn't use a regression derived from shrews to estimate the mass of an elephant. I'm concerned that what you're doing here is more or less equivalent.

One potentially relevant issue is whether extant placentals are the ideal extant comparator for Cretaceous "cladotheres" and even some Paleocene eutherians. Horowitz (2000) notes that some Cretaceous eutherians lack important features that characterize most or all extant placentals, including a pulley-shaped astragalar trochlea, complete encasement of the astragalus by the fibula, presence of a plantar tubercle, and a straight calcaneal tuber. In this regard, Cretaceous "cladotheres" and Paleocene eutherians are more similar to extant marsupials than placentals. A modern sample that includes both marsupials and placentals would be more appropriate than a placental-only sample. I would recommend adding a sample of marsupials across the same locomotor categories to the modern sample. If you can still get good separation and classification of modern locomotor categories, the fossil categories would be more robust.

2. Related to the first point, a number of classifications of fossil taxa are unexpected and conflict with previous assessments of the locomotor ecology of these taxa. You've focused on three taxa (Purgatorius, Ectoganus, Dissacus), but there are several other classifications that are equally, if not more concerning, including Pachyaena and Kulbeckia, both of which have been considered cursorial based on holistic assessments of postcranial anatomy (O'Leary and Rose, 1995; Chester et al., 2010; Averianov and Archibald, 2017). In these and other instances, the fact that overall postcranial anatomy does not support the locomotor classification derived from the tarsal DFA should be a cause for concern, as it serves as a partial test of the validity of the DFA classifications.

Additionally, I'm concerned by the number of changes in predicted locomotor ecology between the first draft and the current draft. Of the 25 fossil taxa that were included in the original draft, the predicted locomotor group of 7 (28%) has changed in the revised draft. I assume this reflects the change from LDA to RDA, but it doesn't leave me with much confidence in the reliability of your results.

3. The key paragraph in the discussion from lines 339 to 361 requires extensive modification. First, you're trying to have it both ways. On the one hand, maybe the DFA is accurately classifying Paleocene mammals and they were robust because they were semifossorial. On the other hand, maybe the DFA isn't accurate and Paleocene mammals only looked similar to modern semifossors due to robusticity unrelated to locomotor ecology. The latter would undermine the significance of the DFA.

The argument that a lack of tightly interlocking joints was beneficial during the early stages of the post-K/T eutherian adaptive radiation is interesting, but it needs quite a bit of improvement. First, as written, it comes across as almost Lamarckian (selection for acquired myological adaptations). I don't think that's what you're arguing, but it needs to be rewritten for clarity. Additionally, it's not clear why this would specifically benefit eutherians. Paleocene multituberculates and metatherians have similarly loose joints (if anything, they're less tightly interlocking). Why were they unable to radiate to the extent seen in eutherians?

Although it may be a moot point given the problems with the DFA classifications, you might want to consider an alternative link between postcranial morphology and eutherian success. Robertson et al. (2004) have argued that burrowing and submersion were important mechanisms for surviving the initial K/T event. If many K/T interval eutherians were semifossorial as suggested by the DFA results, this might explain the differential survival of eutherians.

4. The basis for assigning extant taxa to locomotor groups also needs to be more rigorously documented. While you give a general overview, it's impossible to determine how the locomotor group of any particular taxon was determined. For instance, was *Atilax*, which is a marsh dwelling herpestid that falls out close to semiaquatic taxa in the DFA determined to be terrestrial based on the literature, comparisons to closely related species, or observations based on behavior. The same question could be asked of *Pekania pennanti* (you should update the generic assignment, by the way) which you classify as terrestrial. It falls out with scansorial/arboreal taxa in the DFA and Heinrich and Rose (1995) classify it as scansorial.

5. I'm not sure how appropriate it is to apply disparity analysis to DFA scores. I'm having trouble finding examples of similar analyses in the literature, and it doesn't make much sense intuitively, especially given the questions surrounding the applicability of the DFA to the fossil sample. If the DFA loadings don't reliably correlate with locomotion in Cretaceous/Paleocene mammals, I'm not sure what disparity in those loadings would even mean. I would suggest adding citations to the literature used to determine that this analysis was appropriate.

6. Some statistical information is missing or at least not obvious. With regard to the DFA, how many terms are used in the classifying equation? In the Supplemental Information, scores for 5 linear discriminant functions are provided. Did all 5 contribute to the equation?

In the previous draft, you stated that "We used all 29 PC axes to calculate morphological disparity". That statement has been removed, but nothing has replaced it. How many PC and LD scores were used in the disparity analyses? Was anything done to scale the axes to account for the different amounts of variation explained by each axis? For instance, PC1 accounts for ~23% of total variance, while PC29 accounts for 0.084%. Was variance along these axes treated equivalently by the disparity analyses? I would think that variance along PC1 should be given much more weight than variance along PC29.

References cited in this review that are not cited in the text or SI:

Chester, S. G. B., E. J. Sargis, F. S. Szalay, J. D. Archibald, and A. O. Averianov. 2010. Mammalian distal humeri from the Late Cretaceous of Uzbekistan. *Acta Palaeontologica Polonica* 55:199-211.

Heinrich, R. E., and K. D. Rose. 1995. Partial skeleton of the primitive carnivoran *Miacis petilus* from the early Eocene of Wyoming. *Journal of Mammalogy* 76:148-162.

Robertson, D. S., M. C. McKenna, O. B. Toon, S. Hope, and J. A. Lillegraven. 2004. Survival in the first hours of the Cenozoic. *GSA Bulletin* 116:760-768.

Decision letter (RSPB-2020-2330.R0)

20-Oct-2020

I am writing to inform you that this version of your manuscript RSPB-2020-2330 entitled "Unique locomotor habits in Palaeocene mammals after the end-Cretaceous mass extinction" has, in its current form, been rejected for publication in *Proceedings B*.

This action has been taken on the advice of referees, who have recommended that substantial revisions are necessary. With this in mind we would be happy to consider a resubmission, provided the comments of the referees are fully addressed. However please note that this is not a provisional acceptance.

Please find below the comments made by the referees, not including confidential reports to the Editor, which I hope you will find useful.

- 1) A 'response to referees' document including details of how you have responded to the comments, and the adjustments you have made.
- 2) A clean copy of the manuscript and one with 'tracked changes' indicating your 'response to referees' comments document.
- 3) Line numbers in your main document.
- 4) Please read our data sharing policies to ensure that you meet our requirements <https://royalsociety.org/journals/authors/author-guidelines/#data>.

Sincerely,
Dr Locke Rowe
mailto:proceedingsb@royalsociety.org

Associate Editor Board Member
Comments to Author:

I would agree with the sentiment expressed by both reviewers that the resubmission represents an improvement on the first version of the paper. However, given the intense competition for space in Proceedings B, I am inclined to agree more with reviewer 2: that the paper still requires more work to substantiate findings that are of broad enough significance for the journal readership. The crux of the matter is summarised to an extent in the juxtaposition between the title of the paper ("Unique locomotor habits") and the results that the paper, in its present form, currently actually delivers. Currently the paper does an excellent job of demonstrating the links between tarsal morphology and broad locomotor habit in extant mammals, but I agree with reviewer 2 that it falls short of providing a clear predictive picture of the locomotor habits of extinct taxa. It is of course interesting that extinct forms appear different to extant forms morphologically, but this view was qualitatively supported to an extent previously. The issue comes with the lack of a clear translation to differences/similarities in function and locomotor ecologies. Reviewer 2 provides some suggestions as to how the analysis might be amended and extended to perhaps improve this situation. My recommendation is that authors be given the opportunity to address these reviews in a resubmission, but I would emphasise again that acceptance of the paper would rely a robust response and a strong take-home message about the evolution of locomotor ecology surviving in the resubmission. This might be quite a tough task.

A further minor issue for me, not really discussed by the reviewers, is the addition of the long bone robusticity analysis (Fig 2). I'm not sure this is particularly robust with an $N = 4$ for the

extinct taxa, and I don't think its legitimate to extend the best-fit for this group beyond the range of those four taxa.

Reviewer(s)' Comments to Author:

Referee: 1

Comments to the Author(s).

I am thoroughly pleased with the revised manuscript. The authors have done an excellent job in addressing my comments, concerns and queries, and I think the new version is much stronger and more suitable for publication. Again, the figures are superb.

I have attached the manuscript with some very minor grammatical/punctuation suggestions annotated throughout, and I encourage the authors to carefully run their eye over the supplementary text as well in this regard. I do not feel the need to see any further revisions (should they be required), and I look forward to seeing the paper in print.

Referee: 2

Comments to the Author(s).

The revised version of this manuscript is a significant improvement over the first draft. The inclusion of an expanded Paleocene sample is a welcome addition, and it paints a clearer picture of the distribution of Paleocene eutherians in tarsal morphospace. The manuscript now also strikes a more appropriate balance between the PCA and DFA analyses. However, I think there are still significant issues that need to be addressed, largely centered on the validity of the locomotor predictions made by the DFA. I've outlined my primary concerns below, with additional comments included in the attached pdfs.

1. While the DFA does an impressive job of separating extant taxa by locomotor groups, I'm still concerned by how many fossil taxa fall outside of the discriminant morphospace defined by extant taxa. Many of the "arboreal" and "semifossorial" taxa, in particular, seem to be well outside the envelope defined by all extant taxa, which suggests that their morphology may be too divergent for locomotor patterns to be accurately inferred based on the morphology of extant taxa. Even when fossil taxa do fall within the extant morphospace, many are classified into groups that seem nonintuitive, at least based on the DF1 versus DF2 plot. The "terrestrial" taxa, for instance, include several taxa (Protungulatum, Goniacodon, Vincelestes) that look like they would be more appropriately classified as semifossorial or semiaquatic based on where they plot in discriminant function morphospace.

I think you need to investigate whether its appropriate to use DFA to classify unknowns they fall well outside the range of the known comparative sample. As great a job as the discriminant functions do of accurately classifying extant eutherians, I'm not sure its an appropriate sample for classifying Cretaceous and Paleocene mammals. You wouldn't use a regression derived from shrews to estimate the mass of an elephant. I'm concerned that what you're doing here is more or less equivalent.

One potentially relevant issue is whether extant placentals are the ideal extant comparator for Cretaceous "cladotheres" and even some Paleocene eutherians. Horovitz (2000) notes that some Cretaceous eutherians lack important features that characterize most or all extant placentals, including a pulley-shaped astragalar trochlea, complete encasement of the astragalus by the fibula, presence of a plantar tubercle, and a straight calcaneal tuber. In this regard, Cretaceous "cladotheres" and Paleocene eutherians are more similar to extant marsupials than placentals. A modern sample that includes both marsupials and placentals would be more appropriate than a placental-only sample. I would recommend adding a sample of marsupials across the same locomotor categories to the modern sample. If you can still get good separation and classification of modern locomotor categories, the fossil categories would be more robust.

2. Related to the first point, a number of classifications of fossil taxa are unexpected and conflict with previous assessments of the locomotor ecology of these taxa. You've focused on three taxa (Purgatorius, Ectoganus, Dissacus), but there are several other classifications that are equally, if not more concerning, including Pachyaena and Kulbeckia, both of which have been considered

cursorial based on holistic assessments of postcranial anatomy (O'Leary and Rose, 1995; Chester et al., 2010; Averianov and Archibald, 2017). In these and other instances, the fact that overall postcranial anatomy does not support the locomotor classification derived from the tarsal DFA should be a cause for concern, as it serves as a partial test of the validity of the DFA classifications.

Additionally, I'm concerned by the number of changes in predicted locomotor ecology between the first draft and the current draft. Of the 25 fossil taxa that were included in the original draft, the predicted locomotor group of 7 (28%) has changed in the revised draft. I assume this reflects the change from LDA to RDA, but it doesn't leave me with much confidence in the reliability of your results.

3. The key paragraph in the discussion from lines 339 to 361 requires extensive modification.

First, you're trying to have it both ways. On the one hand, maybe the DFA is accurately classifying Paleocene mammals and they were robust because they were semifossorial. On the other hand, maybe the DFA isn't accurate and Paleocene mammals only looked similar to modern semifossors due to robusticity unrelated to locomotor ecology. The latter would undermine the significance of the DFA.

The argument that a lack of tightly interlocking joints was beneficial during the early stages of the post-K/T eutherian adaptive radiation is interesting, but it needs quite a bit of improvement.

First, as written, it comes across as almost Lamarckian (selection for acquired myological adaptations). I don't think that's what you're arguing, but it needs to be rewritten for clarity.

Additionally, it's not clear why this would specifically benefit eutherians. Paleocene multituberculates and metatherians have similarly loose joints (if anything, they're less tightly interlocking). Why were they unable to radiate to the extent seen in eutherians?

Although it may be a moot point given the problems with the DFA classifications, you might want to consider an alternative link between postcranial morphology and eutherian success.

Robertson et al. (2004) have argued that burrowing and submersion were important mechanisms for surviving the initial K/T event. If many K/T interval eutherians were semifossorial as suggested by the DFA results, this might explain the differential survival of eutherians.

4. The basis for assigning extant taxa to locomotor groups also needs to be more rigorously documented. While you give a general overview, it's impossible to determine how the locomotor group of any particular taxon was determined. For instance, was *Atilax*, which is a marsh dwelling herpestid that falls out close to semiaquatic taxa in the DFA determined to be terrestrial based on the literature, comparisons to closely related species, or observations based on behavior.

The same question could be asked of *Pekania pennanti* (you should update the generic assignment, by the way) which you classify as terrestrial. It falls out with scansorial/arboreal taxa in the DFA and Heinrich and Rose (1995) classify it as scansorial.

5. I'm not sure how appropriate it is to apply disparity analysis to DFA scores. I'm having trouble finding examples of similar analyses in the literature, and it doesn't make much sense intuitively, especially given the questions surrounding the applicability of the DFA to the fossil sample. If the DFA loadings don't reliably correlate with locomotion in Cretaceous/Paleocene mammals, I'm not sure what disparity in those loadings would even mean. I would suggest adding citations to the literature used to determine that this analysis was appropriate.

6. Some statistical information is missing or at least not obvious. With regard to the DFA, how many terms are used in the classifying equation? In the Supplemental Information, scores for 5 linear discriminant functions are provided. Did all 5 contribute to the equation?

In the previous draft, you stated that "We used all 29 PC axes to calculate morphological disparity". That statement has been removed, but nothing has replaced it. How many PC and LD scores were used in the disparity analyses? Was anything done to scale the axes to account for the different amounts of variation explained by each axis? For instance, PC1 accounts for ~23% of total variance, while PC29 accounts for 0.084%. Was variance along these axes treated

equivalently by the disparity analyses? I would think that variance along PC1 should be given much more weight than variance along PC29.

References cited in this review that are not cited in the text or SI:

Chester, S. G. B., E. J. Sargis, F. S. Szalay, J. D. Archibald, and A. O. Averianov. 2010. Mammalian distal humeri from the Late Cretaceous of Uzbekistan. *Acta Palaeontologica Polonica* 55:199-211.

Heinrich, R. E., and K. D. Rose. 1995. Partial skeleton of the primitive carnivoran *Miacis petilus* from the early Eocene of Wyoming. *Journal of Mammalogy* 76:148-162.

Robertson, D. S., M. C. McKenna, O. B. Toon, S. Hope, and J. A. Lillegraven. 2004. Survival in the first hours of the Cenozoic. *GSA Bulletin* 116:760-768.

Author's Response to Decision Letter for (RSPB-2020-2330.R0)

See Appendices E & F.

RSPB-2021-0393.R0

Review form: Reviewer 2

Recommendation

Accept with minor revision (please list in comments)

Scientific importance: Is the manuscript an original and important contribution to its field?

Excellent

General interest: Is the paper of sufficient general interest?

Good

Quality of the paper: Is the overall quality of the paper suitable?

Acceptable

Is the length of the paper justified?

Yes

Should the paper be seen by a specialist statistical reviewer?

No

Do you have any concerns about statistical analyses in this paper? If so, please specify them explicitly in your report.

No

It is a condition of publication that authors make their supporting data, code and materials available - either as supplementary material or hosted in an external repository. Please rate, if applicable, the supporting data on the following criteria.

Is it accessible?

Yes

Is it clear?

Yes

Is it adequate?

Yes

Do you have any ethical concerns with this paper?

No

Comments to the Author

Overall, this is a much improved draft of this manuscript. The addition of extant marsupials to the modern dataset has substantially improved the analysis, and I'm now comfortable that DFA provides insights into the locomotor behavior of Paleocene mammals. I agree with the authors that its not necessary that every locomotor classification be perfectly aligned with other data. My major concern, that too many taxa were falling well outside the DFA morphospace defined by extant placentals, has been addressed. I think this is now a reasonable dataset to investigate patterns of locomotor diversification. There are still issues to be addressed. There are several broad issues as well as smaller issues presented with line numbers in lieu of attaching an annotated copy.

1. The broad conclusion that correlation between the tarsal morphology of extant mammals and locomotor behavior can be used to infer patterns behavior in Paleocene mammals is sound. However, this conclusion is repeatedly undercut (e.g., lines 321-323, 346-348, 365-368) by statements that the relationship between morphology and locomotion in Paleocene mammals may not have been the same as in extant mammals. You don't clearly explain where this skepticism about your results is coming from. As a reader, I'm left with the impression that you don't have much confidence in your analyses.

If I'm not mistaken, I think the basic issue you're trying to confront is as follows: extant semifossorial mammals have robust tarsals; Paleocene mammals have robust tarsals across the board; the robusticity of Paleocene mammal tarsals could be predisposing them to be misclassified as semifossorial (or in the semifossorial-adjacent area of terrestriality). If that is the case, it needs to be addressed clearly and directly. State the problem and the evidence for it, explain why you don't (or do) think it undermines your conclusions, and describe potential future tests that could address the issue.

2. I agree with your argument that individual classifications of extinct taxa don't need to be overanalyzed and that the overall pattern is more important. However, I think you could strengthen your presentation by briefly discussing the outlier taxa. There are three taxa that are outliers along discriminant axis 1 (Orthaspidotherium, Pleuraspidotherium, and Tribosphenomys) and three that are outliers along axis 2 (Afrodon, Bustylus, and Carpolestes). It would be worth noting that two of the three in each category are closely related, since it makes the overall proportion of outliers less dramatic.

You should also consider excluding or reanalyzing another outlier, Tribosphenomys. While the other outlier taxa have genuinely weird tarsals, the tarsals illustrated for Tribosphenomys don't seem to be unusual. Its surprising that its so divergent. I think the problem may be the source of data. Based on data set S1, Tribosphenomys was scored from Meng and Wyss (2001). However, the calcaneal figures in Meng and Wyss are not oriented in a way that should allow them to be used for this study. The dorsal and ventral views are actually dorsomedial and ventrolateral views, respectively. Among other things, you shouldn't be able to see the entire plantar tubercle in dorsal view. At least 4 measurements used in the study (C3, C4, C5 and C12) cannot be reliably measured from the photos in Meng and Wyss (2001). I would suggest replacing the calcaneal data from Meng and Wyss with measurements taken from Fostowicz-Frelik et al. (2018) or excluding Tribosphenomys from the analysis.

3. Your use of the term morphospace can be confusing (e.g., line 375). Its not always clear if it applies to the overall morphospace captured by the PCA or the functional morphospace defined

by the DFA or both. Defining and using distinct terms to discuss those would make the discussion easier to follow.

4. My previous criticism of how extant locomotor classifications were determined still needs to be addressed. I specifically requested that this be documented in the manuscript. If Powell (1981) was used to determine the locomotor category of *Pekania*, it needs to be cited somewhere. Addition of a column to Dataset_S1 after "LOCOMOTOR GROUP" with references for each modern taxon would be ideal.

5. The addition of more taxa and their effect on the discriminant analysis is making Fig. 1b, d, and f and Fig. S4b, d, and f difficult to read. In particular, the extinct taxa are obscuring the pattern of the extant taxa. I would suggest adding either a column to Fig. S4 or a separate figure showing all of the extant taxa without the fossil taxa included.

6. Figs. S12d-e and S13d-e are impossible to read given fonts and the density of text, markers, and lines. These graphs (especially the "e" portions) need to be presented at a larger scale so they're legible.

Additional comments:

Lines 155-157: In the previous draft, these distinctions also applied to placentals. Is this no longer true?

Line 156: Remove "," after "joint". There are only two items in the list.

Lines 213-215: I think this should be reworded for emphasis. What the LDA is indicating is that, despite having distinctive morphologies, most fossil taxa appear to be exploiting locomotor strategies similar to extant mammals.

Lines 249-250: The statement described here should be added.

Lines 309-311: I don't think this is the correct interpretation of the pattern you've observed. Putting the two values (range and variance) together, they indicate that Paleocene mammals occupied the same area of morphospace as Cretaceous mammals (range), but at a greater density (variance). That's consistent with greater numerical diversity, which makes sense given that you've sampled more Paleocene taxa. I don't think it implies that Cretaceous mammals were exploiting niches in different ways.

Lines 319-321: Can you document this connection? Right now, this reads as speculation. It seems possible, but some evidence should be provided. In particular, do Paleocene taxa with strong negative scores along PC1 and PC2 (further from the modern sample) tend to be on the periphery of the DFA functions?

Lines 333-335: Simplify, change:

"these ground-dwelling locomotor strategies (or morphologies most similar to those exhibited by extant mammals that exploit these locomotor strategies) were key adaptive traits for"

to:

"ground-dwelling locomotor strategies were key to"

I don't think the qualifications are needed here, they just make the statement more difficult to parse.

Lines 342-342: Change:

"semi-fossorial(-like) bauplan (retaining some traits of that morphotype)"

to:

"ground dwelling bauplan"

Again, the qualifications aren't necessary.

Lines 349-350: I renew my objection to including this statement despite your response. This is a completely circular argument. You use discriminant scores to classify taxa as either terrestrial or semifossorial. You then take the discriminant scores for taxa assigned to each group and use them to demonstrate that the mean values are significantly different. To the extent that this says anything at all, it's that the discriminant functions do a good job of reliably separating functional groups.

Line 350: It's not clear what scenario the phrase "In this scenario" is referring to. It doesn't make sense with regard to the previous sentence.

Lines 353-354: Please discuss how this hypothesis could be tested. If you're correct, Paleocene eutherians should be more robust than Paleocene metatherians or multituberculates, which would account for their divergent fates. More robust placental subclades should exhibit differential survival in the Paleocene. More robust mammals should show greater survival across the K/T boundary. I doubt any of these predictions can currently be tested, but you should indicate what data could be used to test this hypothesis in the future.

Line 358: Change "given rise to" to "have favored development of"

Lines 370-373: Aside from Adapisoriculidae and Pleuraspidothériidae, the fact that Paleocene mammals fall within or very close to the functional morphospace defined by the DFA suggests that their locomotor strategies are easily compared to extant mammals.

Line 374: Change "amount" to "area"

Lines 374-375: Again, this is one place where discriminating between overall and functional morphospace would be very helpful. It's not clear if you're referring to overall tarsal morphospace or functional morphospace.

Line 376: Delete "warmer and". It's surprising that Paleocene mammals showed so much locomotor diversity because the global environment was more homogenous, not because it was warmer. The warmer climate may have driven the greater homogeneity, but the mere fact that it was warmer isn't why it's surprising that Paleocene mammals showed so much locomotor diversity.

Lines 378-379: Delete "which were nonetheless punctuated by hyperthermals". Again, this is distracting from your point.

Lines 385-394: I don't think this needs to be here. If you were focused on the specific locomotor reconstructions of individual taxa, this would be appropriate. Since you're looking at broader patterns and trying to deemphasize individual taxa, this just breaks the flow of the discussion.

Lines 397-398: Again, aside from Pleuraspidothériidae and Adapisoriculidae, where is the evidence for distinctiveness in locomotion? Paleocene mammals certainly occupy a distinctive area of tarsal morphospace, but functionally they seem comparable to extant mammals given the amount of overlap in the DFA.

SI, page 24, bottom, "Higher DF3 scores are associated with morphologies that permit a greater degree of rotational movement between the astragalus and calcaneum": Wouldn't movement between the astragalus and calcaneum facilitate inversion, which is associated with climbing? Artiodactyls are really the exception here where the subastragalar joint is involved in terrestrial locomotion. More typically, terrestrial taxa lock down the subastragalar joint and its arboreal/scansorial taxa that have mobility here.

Decision letter (RSPB-2021-0393.R0)

10-Mar-2021

Dear Dr Shelley:

Your manuscript has now been peer reviewed and the reviews have been assessed by an Associate Editor. The reviewers' comments (not including confidential comments to the Editor) and the comments from the Associate Editor are included at the end of this email for your reference. As you will see, the reviewers and the Editors have raised some concerns with your manuscript and we would like to invite you to revise your manuscript to address them.

Please make every effort to fully address all of the comments at this stage. If deemed necessary by the Associate Editor, your manuscript will be sent back to one or more of the original reviewers for assessment. If the original reviewers are not available we may invite new reviewers. Please note that even at this stage we cannot guarantee eventual acceptance of your manuscript at this stage.

Research ethics:

Use of animals and field studies:

It is a condition of publication that you make available the data and research materials supporting the results in the article (<https://royalsociety.org/journals/authors/author-guidelines/#data>). Datasets should be deposited in an appropriate publicly available repository and details of the associated accession number, link or DOI to the datasets must be included in the Data Accessibility section of the article (<https://royalsociety.org/journals/ethics->

policies/data-sharing-mining/). Reference(s) to datasets should also be included in the reference list of the article with DOIs (where available).

Please submit a copy of your revised paper within three weeks. If we do not hear from you within this time your manuscript will be rejected. If you are unable to meet this deadline please let us know as soon as possible, as we may be able to grant a short extension.

Best wishes,
Dr Locke Rowe
mailto: proceedingsb@royalsociety.org

Associate Editor Board Member

Comments to Author:

Thank you for revising and resubmitting this interesting work on tarsal morphology in extinct and extant mammals. This is the third submission of the paper, and again represents an improvement on the previous iteration. In particular, expansion of the extant marsupial data set deepens the analysis and much of the discussion around the nature and significance of the findings are now much improved. However, the reviewer still has, in my opinion, a number of legitimate concerns that need to be addressed, but these are now probably less than “major” in scale. The first couple of points raised by the reviewer overlap to an extent with comments I raised about the previous version, which the authors didn’t feel were entirely clear. The title of your paper is “Unique locomotor habits in Palaeocene mammals” so what, exactly, is the unique “habit(s)” (which I interpret to be a synonym for habitat or locomotor ecology) that they had? What unique habit/habitat/locomotor ecology were they living in that no other mammal has ever done before or since? In the current version, this very, very high predictive bar is almost solely set by the title but in earlier versions it was also present in places within the text too. But

the analysis did not and does not reach that bar because you are directly analysing morphology in an attempt to infer function and then from that infer habit/ecology; you're not directly analysing mechanics and you're not directly analysing functional performance within an ecology or habitat (because you can't in fossils). My previous comment that there was "a lack of a clear translation to differences/similarities in function and locomotor ecologies" largely reflects the fact that (mostly, but certainly not always) you had not over-interpreted your results in an attempt to reach that bar: you mostly discuss MORPHOLOGICAL diversity/disparity patterns, and not functional/ecological ones. In this version I read through the discussion and found only a few sentences that explicitly talk about locomotor habit/ecology rather than morphology (prior to the final conclusion/summary paragraph). I think this is fine as long as that is where the expectation is set throughout, and I agree wholeheartedly that the morphological analyses and results are novel and very interesting by themselves, but I felt previously that you (by yourselves) were setting the bar too high. In the current version the title and the final sentence of the paper ("thus indicating distinctive locomotor habits and unexpected ecomorphological diversity") are still guilty of this in my opinion. These issues can be addressed by simply rewording, and, for example by separating the solid aspect of your data (morphology) from the inference-based aspect of your work when you set the study up:

"Here, we investigate the locomotor ecology of Paleocene eutherians, in comparison to a sample of Cretaceous cladotherians and extant therian mammals." You could reword to say something like "investigate diversity of morphology in X and Y ways and subsequently these analyses to assess it's relationship to locomotor ecology."

Something like that, so that you are explicitly separate the morphological aspect from the "bigger picture" functional and/or ecological aspects then what you have achieved will be clearer and more fairly represented. I think this approach combined with the suggestions in the reviewers comment 1 would clear these issues up.

Reviewer(s)' Comments to Author:

Referee: 2

Comments to the Author(s).

Overall, this is a much improved draft of this manuscript. The addition of extant marsupials to the modern dataset has substantially improved the analysis, and I'm now comfortable that DFA provides insights into the locomotor behavior of Paleocene mammals. I agree with the authors that its not necessary that every locomotor classification be perfectly aligned with other data. My major concern, that too many taxa were falling well outside the DFA morphospace defined by extant placentals, has been addressed. I think this is now a reasonable dataset to investigate patterns of locomotor diversification. There are still issues to be addressed. There are several broad issues as well as smaller issues presented with line numbers in lieu of attaching an annotated copy.

1. The broad conclusion that correlation between the tarsal morphology of extant mammals and locomotor behavior can be used to infer patterns behavior in Paleocene mammals is sound.

However, this conclusion is repeatedly undercut (e.g., lines 321-323, 346-348, 365-368) by statements that the relationship between morphology and locomotion in Paleocene mammals may not have been the same as in extant mammals. You don't clearly explain where this skepticism about your results is coming from. As a reader, I'm left with the impression that you don't have much confidence in your analyses.

If I'm not mistaken, I think the basic issue you're trying to confront is as follows: extant semifossorial mammals have robust tarsals; Paleocene mammals have robust tarsals across the board; the robusticity of Paleocene mammal tarsals could be predisposing them to be misclassified as semifossorial (or in the semifossorial-adjacent area of terrestriality). If that is the case, it needs to be addressed clearly and directly. State the problem and the evidence for it, explain why you don't (or do) think it undermines your conclusions, and describe potential future tests that could address the issue.

2. I agree with your argument that individual classifications of extinct taxa don't need to be overanalyzed and that the overall pattern is more important. However, I think you could strengthen your presentation by briefly discussing the outlier taxa. There are three taxa that are outliers along discriminant axis 1 (*Orthaspidothorium*, *Pleuraspidotherium*, and *Tribosphenomys*) and three that are outliers along axis 2 (*Afrodon*, *Bustylus*, and *Carpolestes*). It would be worth noting that two of the three in each category are closely related, since it makes the overall proportion of outliers less dramatic.

You should also consider excluding or reanalyzing another outlier, *Tribosphenomys*. While the other outlier taxa have genuinely weird tarsals, the tarsals illustrated for *Tribosphenomys* don't seem to be unusual. It's surprising that it's so divergent. I think the problem may be the source of data. Based on data set S1, *Tribosphenomys* was scored from Meng and Wyss (2001). However, the calcaneal figures in Meng and Wyss are not oriented in a way that should allow them to be used for this study. The dorsal and ventral views are actually dorsomedial and ventrolateral views, respectively. Among other things, you shouldn't be able to see the entire plantar tubercle in dorsal view. At least 4 measurements used in the study (C3, C4, C5 and C12) cannot be reliably measured from the photos in Meng and Wyss (2001). I would suggest replacing the calcaneal data from Meng and Wyss with measurements taken from Fostowicz-Frelik et al. (2018) or excluding *Tribosphenomys* from the analysis.

3. Your use of the term morphospace can be confusing (e.g., line 375). It's not always clear if it applies to the overall morphospace captured by the PCA or the functional morphospace defined by the DFA or both. Defining and using distinct terms to discuss those would make the discussion easier to follow.

4. My previous criticism of how extant locomotor classifications were determined still needs to be addressed. I specifically requested that this be documented in the manuscript. If Powell (1981) was used to determine the locomotor category of *Pekania*, it needs to be cited somewhere. Addition of a column to Dataset_S1 after "LOCOMOTOR GROUP" with references for each modern taxon would be ideal.

5. The addition of more taxa and their effect on the discriminant analysis is making Fig. 1b, d, and f and Fig. S4b, d, and f difficult to read. In particular, the extinct taxa are obscuring the pattern of the extant taxa. I would suggest adding either a column to Fig. S4 or a separate figure showing all of the extant taxa without the fossil taxa included.

6. Figs. S12d-e and S13d-e are impossible to read given fonts and the density of text, markers, and lines. These graphs (especially the "e" portions) need to be presented at a larger scale so they're legible.

Additional comments:

Lines 155-157: In the previous draft, these distinctions also applied to placentals. Is this no longer true?

Line 156: Remove "," after "joint". There are only two items in the list.

Lines 213-215: I think this should be reworded for emphasis. What the LDA is indicating is that, despite having distinctive morphologies, most fossil taxa appear to be exploiting locomotor strategies similar to extant mammals.

Lines 249-250: The statement described here should be added.

Lines 309-311: I don't think this is the correct interpretation of the pattern you've observed. Putting the two values (range and variance) together, they indicate that Paleocene mammals occupied the same area of morphospace as Cretaceous mammals (range), but at a greater density (variance). That's consistent with greater numerical diversity, which makes sense given that

you've sampled more Paleocene taxa. I don't think it implies that Cretaceous mammals were exploiting niches in different ways.

Lines 319-321: Can you document this connection? Right now, this reads as speculation. It seems possible, but some evidence should be provided. In particular, do Paleocene taxa with strong negative scores along PC1 and PC2 (further from the modern sample) tend to be on the periphery of the DFA functions?

Lines 333-335: Simplify, change:

"these ground-dwelling locomotor strategies (or morphologies most similar to those exhibited by extant mammals that exploit these locomotor strategies) were key adaptive traits for"

to:

"ground-dwelling locomotor strategies were key to"

I don't think the qualifications are needed here, they just make the statement more difficult to parse.

Lines 342-342: Change:

"semi-fossorial(-like) bauplan (retaining some traits of that morphotype)"

to:

"ground dwelling bauplan"

Again, the qualifications aren't necessary.

Lines 349-350: I renew my objection to including this statement despite your response. This is a completely circular argument. You use discriminant scores to classify taxa as either terrestrial or semifossorial. You then take the discriminant scores for taxa assigned to each group and use them to demonstrate that the mean values are significantly different. To the extent that this says anything at all, it's that the discriminant functions do a good job of reliably separating functional groups.

Line 350: It's not clear what scenario the phrase "In this scenario" is referring to. It doesn't make sense with regard to the previous sentence.

Lines 353-354: Please discuss how this hypothesis could be tested. If you're correct, Paleocene eutherians should be more robust than Paleocene metatherians or multituberculates, which would account for their divergent fates. More robust placental subclades should exhibit differential survival in the Paleocene. More robust mammals should show greater survival across the K/T boundary. I doubt any of these predictions can currently be tested, but you should indicate what data could be used to test this hypothesis in the future.

Line 358: Change "given rise to" to "have favored development of"

Lines 370-373: Aside from Adapisoriculidae and Pleuraspidothariidae, the fact that Paleocene mammals fall within or very close to the functional morphospace defined by the DFA suggests that their locomotor strategies are easily compared to extant mammals.

Line 374: Change "amount" to "area"

Lines 374-375: Again, this is one place where discriminating between overall and functional morphospace would be very helpful. It's not clear if you're referring to overall tarsal morphospace or functional morphospace.

Line 376: Delete "warmer and". It's surprising that Paleocene mammals showed so much locomotor diversity because the global environment was more homogenous, not because it was warmer. The warmer climate may have driven the greater homogeneity, but the mere fact that it was warmer isn't why it's surprising that Paleocene mammals showed so much locomotor diversity.

Lines 378-379: Delete “which were nonetheless punctuated by hyperthermals”. Again, this is distracting from your point.

Lines 385-394: I don't think this needs to be here. If you were focused on the specific locomotor reconstructions of individual taxa, this would be appropriate. Since you're looking at broader patterns and trying to deemphasize individual taxa, this just breaks the flow of the discussion.

Lines 397-398: Again, aside from Pleuraspidotheriidae and Adapisoriculidae, where is the evidence for distinctiveness in locomotion? Paleocene mammals certainly occupy a distinctive area of tarsal morphospace, but functionally they seem comparable to extant mammals given the amount of overlap in the DFA.

SI, page 24, bottom, “Higher DF3 scores are associated with morphologies that permit a greater degree of rotational movement between the astragalus and calcaneum”: Wouldn't movement between the astragalus and calcaneum facilitate inversion, which is associated with climbing?

Artiodactyls are really the exception here where the subastragalar joint is involved in terrestrial locomotion. More typically, terrestrial taxa lock down the subastragalar joint and its arboreal/scansorial taxa that have mobility here.

Author's Response to Decision Letter for (RSPB-2021-0393.R0)

See Appendix G.

Decision letter (RSPB-2021-0393.R1)

18-Apr-2021

Dear Dr Shelley

I am pleased to inform you that your manuscript entitled "Quantitative assessment of tarsal morphology illuminates locomotor behaviour in Paleocene mammals following the end-Cretaceous mass extinction" has been accepted for publication in Proceedings B.

Data Accessibility section

Open Access

Your article has been estimated as being 9 pages long. Our Production Office will be able to confirm the exact length at proof stage.

Paper charges

Sincerely,

Dr Locke Rowe

Associate Editor:

Board Member

Comments to Author:

Thank you once again to authors for revising the MS and addressing the comments of the reviewer and myself. I think the authors have done a good job (I like the new title). I apologise that this has been a somewhat long/drawn out process but I do feel the changes made (particularly in expansion of the data set and redrafting of the MS) has made a really big difference to the paper.

Appendix A

General comments

This study investigates skeletal measurements (primarily of the major bones of the mammalian tarsus, the astragalus and calcaneum) in relation to locomotor ecology in a range of Palaeocene placental mammals, extant placentals, and a handful of Cretaceous species, to better elucidate locomotor ecology in the extinct species, providing insight into mammalian evolution across the end-Cretaceous mass extinction.

Understanding how form correlates with function and ecology is a topic of perennial interest, and there is a growing body of research being undertaken in regard to understanding locomotor behaviour and ecology in extinct species of all kinds. As I am not terribly conversant with Palaeocene mammal evolutionary history, I am not in a position to remark on how 'novel' or 'impactful' this study is in the context of the field, but I suspect that it will nevertheless form a welcome contribution.

I found the manuscript a very interesting and overall well-written read, with clear motivation and objectives laid out, methods generally described in a clear and detailed fashion, and with results presented in an intelligible format. I have a few important concerns that I believe require addressing, some of which pertain to the statistical methods used. However, I am not a 'statistics guru' by any stretch, and so suggest that this manuscript be reviewed by at least one person who is more conversant with the methods used here than I am. This is an interesting study and I would like to see it published, and to that end I hope that my comments help produce a more solid piece of work.

Specific comments

1. My first big concern revolves around the robustness indices:

- Insufficient detail is given in the Methods text as to what measurements were actually taken to derive the indices. I can work it out from Fig. 1 (and I can also get it from Table S1), but I shouldn't have to do that.
- Why was minimum midshaft circumference measured for the femur, but width was measured for the humerus? I am puzzled by this difference. Also, how was it ensured that humeral width was always measured in the same plane?
- All of the three indices are simple ratios of midshaft (or tuber, in the case of the calcaneum) dimensions to bone length, but this does not account for allometry in the sample. I found this particularly glaring for the limb bones, as there is a wealth of literature on the subject of allometric scaling in limb bones; moreover, the authors address body size in their tarsal morphometric data, but not here, which was confusing. As ratios, different values can reflect differences in the numerator, or denominator, or both. So for example, the result in Fig. 1e of extant eutherians having less robust calcanei on average may be more to do with them having longer calcaneal tubers, which could be related to increasing moment arms of ankle extensor musculature. And are the humeri and femora of Palaeocene eutherians truly more robust, or are these bones actually just shorter for the animal's body size? I think you need to explicitly account for this. One way could be if you had minimal humeral midshaft circumference (of if not, could estimate by treating midshaft as a circle), you could combine that with femoral circumference to estimate body mass in these Palaeocene taxa (e.g., Campione and Evans, 2012, *BMC Biology* 10: 60). Then you could test if Palaeocene species do indeed have shorter propodia for their size, and ergo say if these bones were more robust or not.
- In sum, as there is more that needs to be disentangled, I'm not yet convinced of the results here, and therefore refrain at this stage from commenting on what interpretations may be drawn from them.

2. My other big concern revolves around the linear discriminant analyses (LDA) and its use in predicting locomotor ecology in the extinct species, especially the Palaeocene taxa.

- Firstly, given the success rate for the extant dataset, we can expect two or three of the extinct species to be misclassified. This should be acknowledged in the paper somewhere.
- More pressing for me though is the fact that most Palaeocene taxa plot in a different part of PC morphospace compared to the majority of extant taxa (Figs 2, 3, S2, S3; also shown by the PERMANOVA tests). This in and of itself is a very interesting result, but it raises the question as to how well the LDA is actually working, and if it is even appropriate to use this method for making

predictions for the extinct species. LDA by definition forces an unknown datapoint into an a priori bin to achieve a classification (which is all the more questionable for a fluid aspect like locomotor ecology). Fig. S6 illustrates my concern well – most (16 out of 20) Palaeocene mammals were classified as semi-fossorial, and tend to cluster in one part of PC space, but the extant semi-fossorial sample (light pink) is scattered elsewhere; I would have expected the two to plot closer to each other in this plot. Indeed, the authors note (lines 313–315) that there is statistical separation between the Palaeocene taxa and extant semi-fossorial species in this plot. Unless I'm missing something I am a bit concerned by what I see: if extant taxa are decidedly different from the extinct taxa, how useful are they for making predictions about the extinct taxa in the first place?

- To better dissect the performance of the LDA, it is important to report the posterior probabilities of the classifications (especially for the extinct taxa). It may well be that for some taxa at least their posterior probabilities are teetering on the precipice of falling into a different locomotor classification, potentially explaining some of what I'm noting above. I'm not sure if PAST can compute posterior probabilities, but the R package 'mass' can do this and there are plenty of online examples showing how this can be done.
3. I have a few more minor statistical quibbles as well:
- It doesn't appear that the effect of phylogeny was taken into account (correct me if I am wrong), but it probably should. For instance, given that you have three main groups in your sample (Cretaceous, Palaeogene, extant), and that these roughly occupy different parts of the mammal family tree, might we not expect them to show some differences simply due to phylogeny? Moreover, what if locomotor habit showed some correlation with phylogeny in the extant sample (I suspect it might) – would this not influence how we make predictions for extinct species?
 - Disparity analyses. Could you give some justification as to why you used the particular method you did? In reviews of my own work, I have been persuaded to use Procrustes distance regression (geomorph package) for something that, on the face of it, appears similar to what you are doing. I'm not saying what you did was wrong; I'm just keen to hear why you used that particular method.
4. Text on lines 176–187. I very much welcome the attempt here to more mechanistically link the statistical shape variation to biomechanical aspects that are of more direct relevance to locomotor behaviour. Yet the statements made here – relating to the type and magnitude of joint mobility, stability, or how load is borne – are unsubstantiated, and the reader has to take these as given. Appropriate literature needs to be cited here, and/or quantitative data presented to back up what is being said here. Otherwise it's just here-say and hand-waving. Given the issues I've noted above in regards to the LDA, having a more mechanistic grasp on what the morphologies (or parts thereof) actually mean in relation to locomotion could provide an alternative, perhaps even better, way of assessing locomotor ecology in the extinct taxa sample.
5. This may just reflect my ignorance, but if the 'size + shape' PCA gives a similar arrangement of taxa as per the 'shape only' PCA, and PC1 in the 'size + shape' analysis is strongly influenced by size, is there actually anything to be gained from the 'size + shape' analysis? That is, can the 'shape only' analysis alone suffice?
6. Placentals vs. eutherians: these two terms are used interchangeably in the main text and supplementary information, but I think it would help fluency (especially for the non-specialist) if just one of these were used throughout.
7. Minor nit-pick on page 3 of the supplementary text: "We did not include extant representatives for Chiroptera given they do not use their hindlimbs for sustained locomotion on land." This ignores the mystacinid bats, which spend most of their time on the ground. I'm not asking you to go and measure these (although it would be interesting to see how their morphology compares with your sample), but a bit clearer wording is necessary.
8. Elsewhere in the supplementary text, some sentences at the moment ramble a bit like a train of thought; these require a bit more attention regarding phrasing and the use of commas and such. Also, be careful of using a capital 'C' for the adjective 'cladotherian'.

9. Some datapoints used in the study related to a composite derived from multiple specimens for a particular extinct species. This is an understandable caveat when working with fossil material. What proportion of the sample comprised data for such composite specimens? Also, can such specimens be explicitly highlighted in the supplementary datasets please (for future reference by others, mainly)?

10. The figures are superb and I commend the authors on a job well done there. Just a couple of minor notes:

- Fig. 1a was not cited in the text (the first item cited is Fig. 1b).
- Please check that the colour schemes used are suitable for colour-blind readers.

11. Tables:

- Table 1: For the permutation test results, there's no need to use scientific notation for the sum of ranges P-values.
- Table 2: For the permutation test results, there's no need to use scientific notation for the P-values.

12. In Dataset S1, the original measurement for the bones are not presented, but I'd like to see them included here. Not only does this aid repeatability of the present study, but such data can be useful for future studies. Also, please have a legend for the abbreviations; I can work out what they stand for, but it should be easily displayed in the file itself.

13. I have also added a number of comments regarding wording and so forth in the annotated file attached.

Appendix B

Overview

I think this is an important and interesting contribution to the field of mammalian palaeobiology, and I would be happy to see this published in Proceedings B in due course. The authors investigate the morphological disparity and locomotor mode of Paleocene and some Cretaceous mammals by comparing these taxa to modern mammals. The authors have chosen appropriate measurements from a highly functional limb element which is able to capture morphological disparity across eutherian and stem therian mammals. The element chosen also allows for a good sample of Paleocene mammals to be included. The authors use a suite of statistical tests to rigorously assess their claims which is commendable (although see comment 1a. regarding their LDA) and strengthens their manuscript.

The paper is well written overall, with points clearly explained. I think that this paper would be of interest to a wide audience, and that Proceedings B is a good fit.

Main comments

Assigning fossils to locomotor groups using an LDA

1a. The number of wrongly assigned extant taxa in the jackknife classification is a little worrying. It suggests that entire regions of morphospace for some locomotor groups are populated by only one species, whereby if that species is removed, organisms with this morphology are wrongly assigned. The authors argue that as they use the full dataset when assigning fossils, and that the full dataset has representatives of each tarsal morphology, this is not an issue. However, the authors admit that their sampling of modern mammals is just over 1%, therefore I believe it would be quite difficult to confidently know that the entire breadth of tarsal morphologies had been captured for each locomotor mode, and I would suggest that it is unlikely to be true (although I am sure every effort has been made to sample fairly and inclusively, I imagine it is near impossible to capture the breadth of tarsal ecomorphology with just over 1% of the extant sample. Nearly half of this sample are carnivores, and I do not agree that the full ecomorphology of rodents has been captured with 11 species, despite these species exhibiting a range of locomotor types). Accurate reconstruction of a fossil species locomotor mode would be particularly difficult where there is no overlap with extant species in the CF dataset, which seems to be the case along PC1 at least for some of the Paleocene mammals (although it is difficult to tell in 3D). Furthermore, where a locomotor group is overwhelmingly occupied by mammals from a single clade (i.e. see cursorial), I imagine it would be difficult to assign species from any clade that is particularly phylogenetically removed, as presumably phylogeny is not completely removed from tarsal morphology.

The authors also discuss the issues surrounding discretising what are often complex locomotor behaviours, and there are a number of classifications which could be considered controversial, or at the least highly variable. For example, *Erethizon dorsatum* could be considered arboreal, and there are a number of Carnivora species that could be considered cursorial. Without collecting additional data, one possible test of how robust the fossil assignments are could be to re-assign marginal taxa to their other possible locomotor modes and see whether this changes the predictions for the Paleocene and Cretaceous taxa. I think at the very least, the predicted fossil locomotor modes should be discussed with

extreme caution. I do not find it unrealistic that these taxa could have been primarily terrestrial and semi-fossorial, but neither am I convinced by the LDA.

This isn't necessarily a huge problem for the paper as a whole. There are a number of interesting results regarding morphospace occupation, and the focus could be centred on these results, with assertions about the possible locomotor modes a more secondary point.

1b. Please explain in the supplement the methodology for the resubstitution and jackknife classifications (for anyone unfamiliar with LDAs) and report the percentage predictive accuracy (is 88.89% for the resubstitution classification? Please make clear if so [line 151]). Consider putting the success rate (percentage) in your confusion matrix in the supplement (Tables S3, S4, S9, S10).

PCAs

1c. For the shape dataset, the authors z transform each species to minimise the effect of size. For the size + shape dataset, the authors do not z transform their data. It makes sense not to z transform each species in the size + shape dataset, but as a result PC1, which correlates with tarsal size, accounts for 95% of the variance. It is standard to z transform each trait (rather than species) before subjecting them to a PCA in order to stop one trait from dominating the analysis (i.e. if one trait ranges from 0.1 – 1, but another trait ranges from 1-10, the trait that ranges from 1-10 will carry more 'weight' in the PCA analysis. It might be that PC1 (tarsal size) describes such a large proportion of data not because the relative variance is higher, but because the absolute values are on a larger scale. Presumably, z transforming each trait would not 'remove' the size signal for each species (individual species which are larger will still have relatively larger trait values compared to species that are smaller [similarly, presumably z transforming each species would not completely remove the effect of having traits on different scales – but perhaps the scales are close enough that it is not an issue.]), but it may allow size to be analysed on a level playing field to shape. Body size changes across the boundary could be investigated using a single body size proxy. If there is a reason that the size + shape trait data is not z transformed, please explain.

1d. At the top of page 16, the authors discuss Paleocene morphospace occupation in comparison to their extant sample. It may be worth noting here that there are a number of locomotor types (and therefore presumably morphologies) known among extant mammals that were left out of their analyses (for reasons explained in their supplement). Therefore, they should be clear to discuss morphospace occupation relative to their extant sampling, not extant mammals more generally. I understand the rationale behind excluding them, nevertheless, they are ecomorphologies currently known from Recent mammals which are not recognised among Paleocene mammals and perhaps should be mentioned briefly in the discussion (e.g. fully aquatic, flying, gliding etc [although of course absence of evidence for these morphologies among Paleocene mammals is not necessarily evidence of absence – but we must work with what we have]). Alternatively, discuss around line 325.

Data Visualisation

1d. Although I appreciate the time and effort that has gone into producing visually appealing 3D figures (Fig. 2a-c), I find it near impossible to accurately assess the spread of points in morphospace. In particular along PC2 and PC3. I would prefer to see 2D figures in the main text. Perhaps you could consider moving the 3D figures to the supplement (failing this, ensure there are 2D versions of all your 3D main text figures in the supplement [e.g. like Figure S1]). Particularly, I would like to see 2D figures for your Canonical Functions morphospace (these are not in the in the supplement either). Although the proposed fossil locomotor groups and extant locomotor groups are shown in sup. fig. S2, these are the PCA analyses, and the Paleocene taxa show nearly no overlap with any modern groups. It would be interesting to see the same groupings (fossil and extant mammal assigned groups) in CF morphospace (consider using the same colour for locomotor mode, but a different shape for fossil and extant mammals – i.e. green circle = extant arboreal, same green but diamond = fossil arboreal).

Line by line suggestions

2a. Throughout: 'Paleocene' is misspelt throughout. Even in English, Paleocene is never spelt with an 'a' as it is the Pal-Eocene (the 'old Eocene').

2b. Line 74: 'Extant taxa were classified into six locomotor groups following the literature'. As this is such an integral part of your study, I would list the locomotor groups here in the main text.

2c. Line 76: '...and PanTHERIA Database'. Consider 'and **the** PanTHERIA Database'

2d. Line 81: '...as a means for mitigating against...'. Doesn't read well. Perhaps '**in order to mitigate**'.

2e. Line 124: '...dispRity' package⁵⁵. 55?

2f. Line 134: '...tests to further test...'. Consider '...tests to further **assess**...'

2g. Line 232: '...reveals the same range trends...' This doesn't read well.

2h. Line 254-259. Long sentence. Consider breaking up.

2i. Line 264: '...such as the double pulley astragalus observed in Artiodactyla.' Reference?

2b. Line 274: Word missing from following sentence: '...resulting in greater observed range morphospace occupation (Fig. 4).'

2k. Line 297: 'Then following the K-Pg extinction...'. Sentence beginning with 'Then'.

2l. Line 312: '...but the animals did not...'. Consider '... but the animals **may not have**...'

2m. Page 7 of supplement. Word missing from following sentence: ‘...when resampled this rate drops due to areas of previously morphospace being excluded.’

2n. Page 7 of supplement. Word missing from following sentence: ‘...but for the purpose looking at eutherian locomotion at a broader scale...’

2o. Page 8 of supplement: ‘...don’t necessarily reflect inaccuracy in the data.’ Change to ‘...do not necessarily reflect inaccuracy in the data.’

2p. Page 8 of supplement: ‘...note that although extant hippotamids spend much time in bodies of water they can’t actually swim.’ Change to ‘...note that although extant hippotamids spend much of their time in bodies of water they cannot actually swim.’

2q. Page 9 of the supplement. ‘Variance dissimilarity decreases from the Cretaceous...’. Change to ‘Morphological dissimilarity decreases from the Cretaceous..’.

2r. Page 9 of supplement: ‘...between the Palaeocene compared to the extant sample.’ Change to ‘...between the Palaeocene and the extant sample.’

2s. Page 14 of supplement. Missing space in the following Figure caption: ‘Figure S2. Tarsal ‘shape’morphospace plots’.

Appendix C

Dr. S L. Shelley
Section of Mammals
Edward O'Neil Research Centre
Carnegie Museum of Natural History
5800 Baum Boulevard
Pittsburgh, PA 15206

shelleys@carnegiemnh.org

+1 412 419 8819

18th September 2020

Dear Editors,

In March of this year, my co-authors, Stephen Brusatte and Thomas Williamson, and I submitted a manuscript titled 'Unique locomotor habits in Early Palaeocene mammals after the end-Cretaceous mass extinction' (RSPB-2020-0494) for consideration at Proceedings of the Royal Society B: Biological Sciences. The subsequent reviews were detailed, and the manuscript was declined in its current form in May 2020, but we were encouraged to resubmit our manuscript once we had addressed the reviewers' comments. Since then we have revised our study following the comments and suggestions of the reviewers, which were very helpful, and we are now ready to submit our revised study.

Our paper, for the first time, examines the locomotor diversity of a broad diversity of Palaeocene mammals living after the end-Cretaceous extinction, including the ancestors of living placental mammals. Our study finds a novel and interesting result: Palaeocene mammals exhibited their own diverse and distinctly robust morphologies that belie their historical stereotype as poorly adapted precursors to the evolution of the extant mammal orders. Rather, they flourished in a post-extinction evolutionary radiation and prospered through their own bespoke body plans.

The reviewers made a number of minor comments and suggestions, which we have largely followed (see submitted point-by-point response document). They also made some more substantial suggestions, which we have addressed. We have summarised those concerns here, and provide full rationale in the point-by-point author response document:

1) Reviewer 1 raised concerns with our robustness indices specifically regarding the influence of body size. In our revised submission we have assessed morphological robustness using regression analyses to show how limb bone dimensions scale with body mass for extant mammals and that Palaeocene mammals exhibit comparably short and broad (robust) limb bones for their body mass.

2) All three reviewers had comments regarding the use and suitability of Principal Components Analysis (PCA) and Discriminants Analysis (DA) for our study. These comments were somewhat contradictory, two reviewers raised concerns over the use of the DA while the other suggested that the DA should be the focus of the manuscript. Two reviewers also questioned the usefulness of the 'size+shape' analyses to the overall objectives of the study. In revising our manuscript and after careful consideration of how to reconcile the conflicting reviewer suggestions, we have used PCA and DA on tarsal shape data. In removing the 'size+shape' analyses, we were able to give more attention to the DA (following the suggestion of reviewer 2) and better explain what the analyses show and how they differ (to alleviate the concerns of reviewer 1 and 3). We have provided explicit tests as to show that both the PCA and DA are appropriate for our dataset. We have also provided detailed discussion regarding the morphologies captured by our morphospaces functional interpretation which better demonstrates why our data and methods are appropriate

3) The reviewers had comments regarding our fossil classifications using the DA. We now use a Regularised Discriminant Analysis (RDA) model in R to provide locomotor classifications for the fossil species. This method is more appropriate for our data (details are provided in our methods text) and allows us to include posterior probabilities for our model. We are also more circumspect in how we discuss our locomotor classifications for the

fossil taxa in the manuscript and have provided several case studies in the supplement illustrating why the RDA is useful but also where its limitations lie. In short, we make it clear that we are not focusing on the individual locomotor classifications for specific extinct species, but rather it is their spread in morphospace (both PCA and DA) that is informative.

4) Some reviewers expressed concern that our sample size of extant mammals was not large enough to represent the span of locomotor diversity. We have provided explicit rationale in the main text and supplement, justifying why this sample is a robust proxy dataset for representing the diversity of extant mammals and locomotor behaviours we are interested in. We have also described and figured both the PCA and DA morphospaces to better explain how those spaces work and the morphological trends they capture, we then related those morphologies to locomotor behaviours. Additionally, the RDA demonstrates the strong predictive capability of our dataset. We have also edited our language in the results and discussion section so that when we refer to ‘extant taxa’ it is clear that we are referring to the extant taxa included in our sample.

5) Reviewer 2 made several comments regarding our fossil sampling. We followed the suggestion to revise our Palaeocene sample which now includes 40 species. We decided it was reasonable to incorporate some Eocene species where their generic range extends back into the Palaeocene. Reviewer 2 also raised the concern that sample size differences may be an issue in our disparity comparisons, as some mammal subsets (e.g. Cretaceous species) are considerably smaller than others (e.g., extant mammals). We are aware of this and use two statistical methods that explicitly take into account sample-size differences when making comparisons: a bootstrapping approach and a stringent permutation test (developed specifically to compare disparity in unequal-sized groups during one of our previous papers on morphological disparity: Brusatte et al., 2014, *Current Biology*). We do not think it is necessary to eliminate the disparity analysis because of the sample size differences between groups. It is the norm in palaeontological studies to compare the disparity of unequal sized groups, using proper statistical tests that take into account the size difference (as we have done). We retain the disparity analysis and provide clear discussion of the caveats regarding the Cretaceous sample in the text.

4) Reviewer 2 had comments about how we frame our paper, suggesting that we have set up a ‘straw man’ argument in our introduction and discussion when we say that Palaeocene mammals are often considered to be ‘arrested’ or ‘generalised.’ We understand the reviewer’s point; however, it is true that Palaeocene mammals were frequently typecast as ‘generalised’ in the historical literature, which we support with numerous citations. Based on our work in the field, we feel that such a perception still permeates how these animals are discussed. Many of the Palaeocene species on which the ‘archaic’ stereotype was founded have not been redescribed in light of newer fossil discoveries and inferences. It is also true, as the reviewer outlines, that recent studies have started to change this narrative by arguing for more specialised locomotor habits in some Palaeocene species. These studies, however, have been limited in scope and often focus on individual species or small subgroups. What our study does for the first time is focus on a broad array of early Palaeocene mammals and establishes their locomotor diversity based on explicit, quantitative comparison to a broad range of extant mammals. We have edited the text in our introduction and discussion and stated where more recent studies that have started to change the perception of Palaeocene mammal fauna. We feel that this introduction more accurately establishes the novelty and importance of our study.

With these revisions, we feel that our paper is now substantially stronger. We thank the reviewers for their detailed and helpful comments, and the editors for inviting these reviewers and feel that *Proceedings of the Royal Society B: Biological Sciences* would be an ideal fit for our paper, We hope that the manuscript meets the high standards of your journal group and look forward to receiving a response from you. We are grateful for any help or insight you can provide as we seek a good home for our paper.

Yours Faithfully,

On behalf of the authors

Sarah L. Shelley

Appendix D

Unique locomotor habits in Palaeocene mammals after the end-Cretaceous mass extinction

Shelley, Brusatte & Williamson

Author responses to review comments

Associate Editor

Board Member: 1

Comments to Author:

Thank you for the opportunity to review this paper. Apologies to authors for the time taken to reach this point. We had a difficult time finding reviewers, probably reflecting the current disruption to working routines. However, the three reviewers have provided both timely and exhaustive comments on the paper, and I hope these are helpful going forward.

It is my opinion that this paper has promise, but needs considerable re-evaluation before it is ready for Proceedings B. The two aspects that jumped out at me on reading the paper were **the lack of explicit consideration of body size in the analysis, and the apparent discrepancy (lack of overlap) in morphospace between extant and extinct animals that are ultimately concluded to be of the same locomotor ecology**. These issues are noted by at least one (sometimes two) of the reviewers, in addition to a number of other **potential statistical issues**. Despite these issues, all three reviewers remain positive about the potential contribution this study could make to field. It is therefore my recommendation that the authors be given the opportunity to produce a revised version of the study that addresses the issues raised in the reviews, in particular the comments on the statistical analyses. Competition for space in Proceedings B is extremely intense and so acceptance of the paper would rely not only a **robust re-evaluation of the statistics** but also a **strong take-home message surviving this re-evaluation**.

The below list are brief responses to the issues noted by the editor. More detailed responses are provided in the following text with the associated reviewer comments.

The 'lack of consideration of body size' in the robustness indices has been addressed by swapping out the indexed comparisons for regression analyses which show that the Palaeocene taxa have short and broad limbs for their body size as per the suggestion of reviewer 1. Furthermore, we note that our multivariate (PCA and DA) analyses aim to quantify tarsal shape, minimising body size, as we are most interested in shape and not size (body size has been extensively studied in Palaeocene mammals; locomotor behaviour, represented in our case by tarsal size proxies validated in extant mammals, has not).

The 'apparent discrepancy (lack of overlap) in morphospace between extant and extinct animals that are ultimately concluded to be of the same locomotor ecology' has been addressed by better explaining how the morphospaces work and why we would not expect the Palaeocene species to completely overlap with the extant taxa of equivalent locomotor group in the Principal Component (PC) morphospaces which seek to array all taxa by dissimilarity. As such, the PC morphospace best illustrates tarsal morphologies which distinguish Palaeocene species from extant and Cretaceous mammals and the Discriminants Analysis (DA) morphospace shows how fossil species are similar to extant species when those extant taxa are ordinated with the supervised input of locomotor behaviour. We have also provided a more detail methodology and tests to show that the multivariate analyses are appropriate for our dataset.

We also summarise key revisions to our manuscript here:

1. We have increased our Palaeocene sample to a total of 40 species. All species have a generic temporal range which extends into the Palaeocene. Although we understand that reviewers often ask for ever bigger and more complete samples, this is the largest sample that is available to us. Our techniques require precise measurements, so we cannot easily take grainy or view-limited figures from the literature and incorporate them into our dataset. Given travel limitations because of the Covid-19 pandemic, we also cannot visit additional museum collections to increase our sample. However, this is not a problem: our sample is already by far the largest sample of Palaeocene mammals incorporated into a single dataset, it is representative of the range of species and anatomies of Palaeocene mammals, and because our main argument is that Palaeocene mammals have a large distribution in morphospace, additional samples could only amplify (not decrease) this finding.

2. We have assessed skeletal robustness using regression analyses to account for body size. We find that the Palaeocene taxa do exhibit short and broad limb (robust) limb bones for their body size.
3. We provide equal emphasis on the Principal Components Analysis and Discriminants Analysis in the manuscript. This includes more detailed text and additional figures to illustrate how the morphospaces capture tarsal morphology and how these morphologies relate to locomotor strategies.
4. We replaced the fossil classification procedure, which was previously conducted using a Linear Discriminants Analysis model, with a Regularised Discriminant Analysis (RDA) model. This method is more appropriate for the size and construction of our tarsal dataset. We have provided a full description and justification for this method in our text. We have also provided additional data from the RDA including group priors and posterior probabilities for the extant and extinct locomotor classifications. We note that this procedure still classifies a substantial number of Palaeocene mammals as semi-fossorial.
5. We conduct disparity analyses and associated tests on both the PC scores and the discriminant functions derived from the DA.
6. We have edited our language to be more transparent and cautious when referring to the extant taxa so that it is clear we are talking about the extant taxa included in our sample and not extant mammals as a whole.

Overall, the key findings of our study remain the same but are now presented with more detailed results and discussion. We show that Palaeocene mammals occupy a distinctive region of tarsal morphospace relative to Cretaceous and extant species that is distinguished by their morphological robustness. We find that many Palaeocene species exhibit tarsal morphologies most comparable with morphologies of extant semi-fossorial mammals; however, we acknowledge that these similarities may not be unambiguously associated with semi-fossorial behaviour and present an alternative hypothesis as to why we are finding similarities. Disparity analyses indicate that Palaeocene mammals attained comparable morphospace diversity to the extant sample. In summary, our results show that mammals underwent an adaptive radiation in locomotor behaviour by combining a basic eutherian bauplan with inimitable anatomical specialisations to attain remarkable ecomorphological diversity following the end-Cretaceous mass extinction.

Reviewer(s)' Comments to Author:

Referee: 1

General comments

This study investigates skeletal measurements (primarily of the major bones of the mammalian tarsus, the astragalus and calcaneum) in relation to locomotor ecology in a range of Palaeocene placental mammals, extant placentals, and a handful of Cretaceous species, to better elucidate locomotor ecology in the extinct species, providing insight into mammalian evolution across the end-Cretaceous mass extinction. Understanding how form correlates with function and ecology is a topic of perennial interest, and there is a growing body of research being undertaken in regard to understanding locomotor behaviour and ecology in extinct species of all kinds. As I am not terribly conversant with Palaeocene mammal evolutionary history, I am not in a position to remark on how 'novel' or 'impactful' this study is in the context of the field, but I suspect that it will nevertheless form a welcome contribution.

I found the manuscript a very interesting and overall well-written read, with clear motivation and objectives laid out, methods generally described in a clear and detailed fashion, and with results presented in an intelligible format. I have a few important concerns that I believe require addressing, some of which pertain to the statistical methods used. However, I am not a 'statistics guru' by any stretch, and so suggest that this manuscript be reviewed by at least one person who is more conversant with the methods used here than I am. This is an interesting study and I would like to see it published, and to that end I hope that my comments help produce a more solid piece of work.

Specific comments

1. My first big concern revolves around the robustness indices:

- Insufficient detail is given in the Methods text as to what measurements were actually taken to derive the indices. I can work it out from Fig. 1 (and I can also get it from Table S1), but I shouldn't have to do that.

We have followed the reviewer's suggestion and swapped out the robustness indices for regression analysis on body mass (calculated using the Campione and Evan equation) against a long bone dimension.

- Why was minimum midshaft circumference measured for the femur, but width was measured for the humerus? I am puzzled by this difference. Also, how was it ensured that humeral width was always measured in the same plane?

This comment is no longer applicable given that we have swapped out the indices for regressions. However, to answer to the reviewer's questions: for the original robustness indices we used datasets of Chen and Wilson (2015) and Carrano (1999) to supplement our own measurements and were therefore limited by their measurements/indices. When taking our own measurements, the humerus was held so that the mediolateral axis of the distal end was horizontal.

- All of the three indices are simple ratios of midshaft (or tuber, in the case of the calcaneum) dimensions to bone length, but this does not account for allometry in the sample. I found this particularly glaring for the limb bones, as there is a wealth of literature on the subject of allometric scaling in limb bones; moreover, the authors address body size in their tarsal morphometric data, but not here, which was confusing.

These indices were only ever meant to be a precursor to the main multivariate part of the study but can understand how they are now not as rigorously designed/analysed compared to following analyses. As such, we have followed the below suggestion and now explicitly account for body mass.

As ratios, different values can reflect differences in the numerator, or denominator, or both. So for example, the result in Fig. 1e of extant eutherians having less robust calcanei on average may be more to do with them having longer calcaneal tubers, which could be related to increasing moment arms of ankle extensor musculature. And are the humeri and femora of Palaeocene eutherians truly more robust, or are these bones actually just shorter for the animal's body size? I think you need to explicitly account for this. One way could be if you had minimal humeral midshaft circumference (of if not, could estimate by treating midshaft as a circle), you could combine that with femoral circumference to estimate body mass in these Palaeocene taxa (e.g., Campione and Evans, 2012, BMC Biology 10: 60). Then you could test if Palaeocene species do indeed have shorter propodia for their size, and ergo say if these bones were more robust or not. In sum, as there is more that needs to be disentangled, I'm not yet convinced of the results here, and therefore refrain at this stage from commenting on what interpretations may be drawn from them.

We have followed the reviewer's suggestion and used scatterplots and regression analyses to show that three limb bone dimensions scale with body mass for extant mammals and that Palaeocene mammals exhibit short and broad limb bones for their body size. We note that the Palaeocene taxa are not remarkably separated from the extant sample, but we would not expect them to be based on simple bivariate measures. Interestingly, the Palaeocene taxa included here also appear to be most similar to a number of extant-semi-fossors in the bivariate plots as well as the tarsal morphospaces.

All the raw data including those used for estimating body mass are provided in Dataset S1.

We note that our sample of Palaeocene taxa is limited to four taxa for which we were able to get a complete suite of measurements from associated fossil material. We considered estimating body mass using alternative dental proxies; however, in our experience mass estimate from dental measures can vary greatly so we stuck with the taxa for which we could use the Campione and Evans equation given that these analyses are an 'opener' to the tarsal morphospaces.

2. My other big concern revolves around the linear discriminant analyses (LDA) and its use in predicting locomotor ecology in the extinct species, especially the Palaeocene taxa.

- Firstly, given the success rate for the extant dataset, we can expect two or three of the extinct species to be misclassified. This should be acknowledged in the paper somewhere.

We appreciate the reviewer's concern and never intended for the fossil classifications to be taken as absolutes. In editing this manuscript following the other review suggestions, we have ended up giving more attention to the Discriminant Analysis and have explicitly mentioned how it is useful and where its limitations lie (e.g. paragraph starting L342, specifically L351). We have also been careful in the language we have used, for example saying Palaeocene species are **most similar** to extant semi-fossors. Indeed, the main conclusion of our study is not to say the many Palaeocene species were semi-fossorial but rather they are most similar to extant semi-fossors due to their robustness, which was likely due to the lack of stabilisation in the osteological anatomy, instead they were reliant on their soft tissues which is evident in large muscle attachment areas and heavier mass for their body size.

It is also worth noting that we have used a Regularised Discriminant Analysis (RDA) for predicting locomotor mode for the fossil taxa which is more appropriate for our dataset. We provide a full rationale for using an RDA in the MS and accompanying methods text in the supplement.

- More pressing for me though is the fact that most Palaeocene taxa plot in a different part of PC morphospace compared to the majority of extant taxa (Figs 2, 3, S2, S3; also shown by the PERMANOVA tests). **This in and of itself is a very interesting result, but it raises the question as to how well the LDA is actually working, and if it is even appropriate to use this method for making predictions for the extinct species.** LDA by definition forces an unknown datapoint into an a priori bin to achieve a classification (which is all the more questionable for a fluid aspect like locomotor ecology). Fig. S6 illustrates my concern well – most (16 out of 20) Palaeocene mammals were classified as semi-fossorial, and tend to cluster in one part of PC space, but the extant semi-fossorial sample (light pink) is scattered elsewhere; **I would have expected the two to plot closer to each other in this plot.** Indeed, the authors note (lines 313–315) that there is statistical separation between the Palaeocene taxa and extant semi-fossorial species in this plot. Unless I'm missing something I am a bit concerned by what I see: if extant taxa are decidedly different from the extinct taxa, how useful are they for making predictions about the extinct taxa in the first place?

We do understand the reviewer's concern here. In editing this manuscript to include a greater emphasis on the Discriminant Analysis (DA) we have added text to the methods and results describing how the two morphospace methods differ in their construction (e.g. L252 and methods section of the supplement). We have provided preliminary tests to show that our data meet the requirements of the multivariate analyses and ending up swapping out the LDA classification for an RDA protocol. We have also added more detailed descriptions of our morphospaces in the supplement (with accompanying data figures – biplots, loadings, illustrations etc) to better explain what the plots are showing in an anatomical/functional context. We do think the morphospaces do a really nice job of capturing tarsal anatomy, we hope by describing them and providing accompanying figures and data that readers can better intuit how these spaces work.

The PCA is completely unsupervised with no a priori assumptions so that all the datapoints are ordinated to maximise separation between them. As such they neatly illustrate the ways in which Palaeocene species differ to extant and Cretaceous mammals in their tarsal anatomy. For the DA, the extant taxa are categorised a priori and ordinated into a morphospace of reduced dimensions which best separate those categories. The fossil taxa are then ordinated into the morphospace and in doing so capture how the fossil taxa are most similar to the extant taxa. There is some overlap in morphologies that the PC and DA axes capture but overall, they differ given how the spaces are constructed. In the supplement, we have provided written descriptions for the first three axes for the PC and DA morphospaces. These descriptions (plus data figures) show that both our PC and DA morphospaces are capturing morphologies that gibe well with the existing literature regarding astragalar and calcaneal functional anatomy.

We note that in the DA morphospaces, some of the Palaeocene taxa are widely dispersed around the extant taxa. As such their locomotor classifications could be compromised. We do not consider this a failure of the analysis, but rather that some Palaeocene species co-opt some pretty unusual tarsal morphologies compared to our extant sample. Our measurements are all homologous and our phylogenetic bracket is appropriate (consider that the Chen and Wilson paper was comparing Mesozoic mammal species to an extant sample largely comprised of therians albeit with more general measurements but still making some assumptions of functional homology). Nevertheless, we have been cautious in our language, using phrases such as 'most similar' rather than absolute terms. We also assert that where fossil species fall out in divergent positions, their tarsal anatomy can still be assessed within the functional framework of the morphospace and relative to the other fossil taxa (which was a major objective of this project – to be able to compare contemporaneous (ish) fossil taxa to one another).

We do acknowledge that discretely classifying locomotor behaviour is inherently difficult and may not be the best approach. Future methods, making use of phylogenetically flexible discriminant would be an interesting avenue of enquiry but it is not currently possible. That said, our RDA performs well. Disagreement with resampled classifications are often explained by overlapping locomotor behaviours and the DA morphospace nicely illustrates locomotor behaviour as a continuum. That is why we have included the PCA, DA and RDA classifications in combination – in conjunction with one another they capture the similarities/differences/variability in tarsal anatomy and provide a suite of evidence for figuring out what Palaeocene mammals were doing and why they are often so weird.

Regarding the reviewer's concerns over Fig. S6 (now Fig. S10): this figure is using PC scores combined with the RDA classifications – as such it is showing the differences between the extant and extinct taxa when grouped by locomotor group which is largely governed by separation along PC1. Our objective with this figure is to show that although the extinct taxa are similar to the extant taxa by the features captured in the DA morphospace, they are still differentiated by the features captured by the PCA and how similar/different the extant vs extinct locomotor groups are.

- To better dissect the performance of the LDA, it is important to report the posterior probabilities of the classifications (especially for the extinct taxa). It may well be that for some taxa at least their posterior probabilities are teetering on the precipice of falling into a different locomotor classification, potentially explaining some of what I'm noting above. I'm not sure if PAST can compute posterior probabilities, but the R package 'mass' can do this and there are plenty of online examples showing how this can be done.

This is a good suggestion which we have followed. We were conducting the LDA in PAST is easy to implement but pretty limited and does not provide posteriors. In switching over to the MASS and klar packages in R we were able to utilise discriminant analysis in a more effective manner for our dataset.

3. I have a few more minor statistical quibbles as well:

- It doesn't appear that the effect of phylogeny was taken into account (correct me if I am wrong), but it probably should. For instance, given that you have three main groups in your sample (Cretaceous, Palaeogene, extant), and that these roughly occupy different parts of the mammal family tree, might we not expect them to show some differences simply due to phylogeny? Moreover, what if locomotor habit showed some correlation with phylogeny in the extant sample (I suspect it might) – would this not influence how we make predictions for extinct species?

Phylogeny was not taken into account in the previous submission. We made reference to this in the 'future work section of our discussion'. Given that we do not have a comprehensive phylogeny for Palaeocene mammals and the uncertainty regarding extant ordinal relationships we think this issue of phylogeny would be best tackled at a future date once a well-resolved phylogeny has been established. Indeed, this is an objective of our on-going research. For this project, we have used a broader taxonomic sample to limit phylogenetic signal (this in part, is a reason for our current sample size). However, we understand the reviewer's point, and have endeavoured to implement it as best we can at the current time. We now provide Blomberg's K statistics to show that phylogeny has a limited effect on the distribution of extant taxa (whose phylogenetic relationships are well understood, in comparison to the Palaeocene taxa) in our morphospace. We note that a multivariate

K test - k.mult test (Adams 2014) does exist; however, at this time a dependency R package for this test (conducted using Phylocurve) has been orphaned for R v4.0.2 and as such we are not able to conduct this test (we will continue check this up to the point of final submission for publication).

- Disparity analyses. Could you give some justification as to why you used the particular method you did? In reviews of my own work, I have been persuaded to use Procrustes distance regression (geomorph package) for something that, on the face of it, appears similar to what you are doing. I'm not saying what you did was wrong; I'm just keen to hear why you used that particular method.

We have provided a brief descriptor for each metric in L127. We added an additional variance metric (mean pairwise distance) to the revised study to quantify the density of species within a group relative to each other, rather than their mean. We provide a longer rationale for our choices in the methods text in the supplement (p8-9).

4. Text on lines 176–187. I very much welcome the attempt here to more mechanistically link the statistical shape variation to biomechanical aspects that are of more direct relevance to locomotor behaviour. Yet the statements made here – relating to the type and magnitude of joint mobility, stability, or how load is borne – are unsubstantiated, and the reader has to take these as given. Appropriate literature needs to be cited here, and/or quantitative data presented to back up what is being said here. Otherwise it's just here-say and hand-waving. Given the issues I've noted above in regards to the LDA, having a more mechanistic grasp on what the morphologies (or parts thereof) actually mean in relation to locomotion could provide an alternative, perhaps even better, way of assessing locomotor ecology in the extinct taxa sample.

In line with the reviewer's suggestion we have provided more detailed descriptions of our morphospaces. Given the page limitations of the journal, we have provided brief summaries in the main manuscript and put the longer text in the supplement. We have described the morphological changes and functional implications for the first three axes derived from the PCA and DA and also provided brief descriptions for each locomotor group as defined by our morphospaces. The axes descriptions are supported by the literature and backed up by references. We have also added a number of supplementary figures including biplots and loadings barplots as well as morphospace figures (Fig. S5 and S8) annotated with exemplar tarsals and tarsal schematic illustrating the first three axes.

5. This may just reflect my ignorance, but if the 'size + shape' PCA gives a similar arrangement of taxa as per the 'shape only' PCA, and PC1 in the 'size + shape' analysis is strongly influenced by size, is there actually anything to be gained from the 'size + shape' analysis? That is, can the 'shape only' analysis alone suffice?

After consideration, we agree with the reviewer (and review below) and removed the 'size + shape' analysis from the manuscript as its purpose was becoming tangential to the main objectives of the paper.

6. Placentals vs. eutherians: these two terms are used interchangeably in the main text and supplementary information, but I think it would help fluency (especially for the non-specialist) if just one of these were used throughout.

While we understand what the reviewer is suggesting, we prefer to retain the use of eutherian and placental. The use of these terms is deliberate and serves a useful purpose, they are well-defined and easily looked up if the reader is unsure.

7. Minor nit-pick on page 3 of the supplementary text: "We did not include extant representatives for Chiroptera given they do not use their hindlimbs for sustained locomotion on land." This ignores the mystacinid bats, which spend most of their time on the ground. I'm not asking you to go and measure these (although it would be interesting to see how their morphology compares with your sample), but a bit clearer wording is necessary.

Noted and edited!

8. Elsewhere in the supplementary text, some sentences at the moment ramble a bit like a train of thought; these require a bit more attention regarding phrasing and the use of commas and such. Also, be careful of using a capital 'C' for the adjective 'cladotherian'.

In revising supplement, we have edited the text to be pithier and to the point.

9. Some datapoints used in the study related to a composite derived from multiple specimens for a particular extinct species. This is an understandable caveat when working with fossil material. What proportion of the sample comprised data for such composite specimens? Also, can such specimens be explicitly highlighted in the supplementary datasets please (for future reference by others, mainly)?

The specimen numbers are listed in Supplementary dataset and references are provided. It is not as clear cut as different specimen numbers equating to a composite taxon as some bones have been accessioned individually. Therefore, we would prefer not to simplify this information to a percentage.

10. The figures are superb and I commend the authors on a job well done there. Just a couple of minor notes:

- Fig. 1a was not cited in the text (the first item cited is Fig. 1b).
- Please check that the colour schemes used are suitable for colour-blind readers.

Thank you! We ended up revising the morphospaces as 2D plots given that the other two reviewers did not like the 3D versions (although did keep 3D versions in the supplement given that we think they are a good way to convey a morphospace – Figs. S5&8). Figure 1 has been broken up into 2 figures given it had sections pertaining to the methods and results and is now cited in the correct order. All figures are suitable for colour blind readers. We also added different datapoint shapes to aid visualisation.

11. Tables:

- Table 1: For the permutation test results, there's no need to use scientific notation for the sum of ranges P-values.
- Table 2: For the permutation test results, there's no need to use scientific notation for the P-values.

Corrected

12. In Dataset S1, the original measurement for the bones are not presented, but I'd like to see them included here. Not only does this aid repeatability of the present study, but such data can be useful for future studies. Also, please have a legend for the abbreviations; I can work out what they stand for, but it should be easily displayed in the file itself.

We have provided the raw measurements used for the regression analyses. In the previous submission, we used indexed data extracted from Chen and Wilson (2015) and did not have raw data for all the taxa.

13. I have also added a number of comments regarding wording and so forth in the annotated file attached.

We have addressed these comments in the new version of the manuscript. Below we have added a few comments on these changes.

Comment on L34, starting 'The nature of mammal diversification following the K-Pg extinction is contentious' Comment saying that the use of 'nature' is a bit ambiguous. We're not just referring to the tempo and mode of diversification, the meaning of which I would consider more applicable to a phylogenetic context. We use the word 'nature' to keep the meaning of this sentence deliberately broad – their phylogeny remains contentious but also aspects of their biology e.g. anatomy, functional morphology, ecology, diet, locomotion etc.

Comment on L35 regarding sampling bias/exposure/biogeography. We added a clause to say 'largely known from the northern hemisphere'.

Referee 2

Comments to the Author(s)

This manuscript makes important contributions to our understanding of both mammalian functional morphology and Paleocene mammalian ecology. The manuscript demonstrates that a discriminant

function analysis (DFA) of linear tarsal measurements can distinguish functional groups among modern placental mammals with substantial accuracy. When the resulting discriminant function is used to classify Paleocene mammals, sampled taxa are recovered as either semi-fossorial or (non-cursorial) terrestrial, a somewhat surprising conclusion in at least a few cases (e.g., *Purgatorius*, **recently considered arboreally adapted**). This manuscript has profound positive implications for our ability to infer locomotor ecology from isolated tarsal remains and for our understanding of locomotor diversity and disparity in early Cenozoic mammals. Unfortunately, these important results are obscured by more problematic aspects of the authors' study. Substantial revisions are required before this manuscript is ready for publication. Major concerns are described below. The attached files detail additional issues with the main and supplemental text.

Regarding *Purgatorius*:

We have added some text to our supplement discussing our classification *Purgatorius*, as well as two other taxa: *Ectoganus* and *Dissacus*, which had unusual classifications in light of other aspects of their anatomy.

We have also tried to be as clear as possible in our language that the RDA classifications are a guide and not absolute determinations – we never intended for them to be taken as such but appreciate that in editing this manuscript for publication some of the nuance in our discussion of our findings may have been lost. To this end we have included a statement in paragraph starting L342, specifically L351. Also in paragraph starting L366.

1. The manuscript **places too much emphasis on the results of a principal components analysis (PCA), at the expense of the DFA**. PCA is informative in visualizing a morphospace and is best used for data exploration. It recovers successive combinations of variable loadings that account for the most variance in the overall sample, without regard to user-defined categories. In contrast, a DFA looks for the combination of variable loadings that best sorts samples into user-defined categories. In the case of this study, the discriminant analysis finds the combination of factor loadings that best sorts the taxa into locomotor categories, regardless of the contribution of that variation to overall variation.

In fact, when I graph CF1 versus CF2, the DFA does an impressive job of discriminating functional categories among extant placentals. Aside from locomotor categories that blend into each other (scansorial and arboreal) or are frequently noted as having extensive functional overlap (semifossorial and semiaquatic), locomotor groups are well separated by the DFA. When Paleocene taxa are added, a few forms extend somewhat beyond the modern data, but most taxa fall within the envelope formed by extant placentals. In contrast, **the PCA is much messier**. Along PC1 and PC2, Fig. S2a demonstrates extensive overlap between every locomotor category aside from terrestrial. It certainly doesn't look like something that could be used to confidently infer locomotor preferences in mammals of unknown habits. **Making it worse is the fact that there is very little overlap between Paleocene and modern mammals**. Most Paleocene mammals fall outside of the morphospace occupied by any locomotor group.

Given this contrast, the DFA results should be front and center in the manuscript. Instead, the results and discussion place far too much emphasis on the PCA results. **For instance, the authors describe and interpret the variables with the largest loadings along PC1-3, but never indicate what variables have the largest loadings along CF1-3**. Those are going to be the variables that are most relevant to assessing locomotor function. Additionally, an extensive discussion of morphological disparity is predicated on the assumption that **morphological disparity is reflective of locomotor diversity**. However, the **PCA results demonstrate that overall morphological disparity is poorly correlated with locomotor diversity**. It might be worth looking at disparity in variables with substantial loadings in CF1 and CF2, but overall morphological disparity doesn't appear to be reflective of locomotor diversity.

We agree with the reviewer's very helpful comment and have revised the manuscript to place equal emphasis on the PCA and DA and now provide morphospace plots, statistical tests and disparity analyses for both. We retain the PCA and give it equal weight to the DA in our study given that it illustrates and provides quantification to the tarsal morphologies which best separate Palaeocene mammals from extant and Cretaceous mammals. Given the PCA is an unsupervised ordination method – this is an interesting result in and by itself (also noted by reviewer 1). We have also provided more detailed descriptions of the first three morphospace axes with accompanying figures to better explain how the morphospaces work.

The purpose of the PCA is to show how Palaeocene mammals differ from extant and Cretaceous mammals. The purpose of the DA morphospaces are to compare Palaeocene mammals to extant locomotor groups and provide locomotor predictions for the extinct mammals. We have reworded our manuscript to better clarify these objectives.

2. The sample of Paleocene eutherians included needs to be expanded. The sample used in this study is not representative of the taxonomic diversity of Paleocene mammals known from published tarsal remains. For instance, there are nine “condylarths”, but only one plesiadapiform. Why are Plesiadapis, Dryomomys, Carpolestes, and Ignacius excluded? Several major groups for which the tarsus has been described are excluded entirely (e.g., Pseudictopidae, Arctostylopidae, Dinocerata, Adapisoriculidae, Oxyaenidae). As a result, the documented diversity/disparity of Paleocene eutherian tarsals is not reflected by the sample used in this manuscript. This casts doubt on the authors’ conclusions concerning diversity and disparity in Paleocene eutherian tarsal morphology and locomotor ecology.

In addition, some additional accounting for which specific taxa were chosen should be provided. I can understand not including every condylarth, but why were specific taxa chosen and not others? Why are peripitychids represented by both Peripitychus and Ectoconus but not Hemithlaeus? Why are the Tiupampa condylarths excluded?

We have taken on the reviewer’s suggestion and revised our Palaeocene sample as best we can; it now includes 40 species. We have included a number of specimens of Eocene age known for tarsal bones which have a generic temporal range that extends back into the Palaeocene. We recognise that there are number of Palaeocene taxa for which tarsal are known but we were not able to include in this project due to not being able to either access the specimens in person (now not possible for the indefinite future due to Covid travel restrictions and museum closures) or recover the necessary measurements from the literature or online repositories. However, we are confident that the addition of new taxa will drastically change the main conclusions of this study; after all, our main conclusion is that Palaeocene mammals have a large variety of locomotor behaviours, and increased sampling would only amplify that conclusion rather than dampen it. Furthermore, we hope that by providing all the raw data, a detailed methodology, and accompanying code that other researchers will be able to replicate and incorporate other specimens if they so wish.

Of the suggestions listed we were not able to recover all measurements for:

Dinocerata, but recognise that potentially useable tarsals are known for *Probathyopsis harrisorum* and *Prodinoceras efremovi*.

Oxyaenidae, but recognise that potentially useable tarsals are known for *Dipsalidictis krausi* and *D. platypus*.

Tillodonts, but recognise that potentially useable tarsals are known for *Azygonyx gunnelli*.

We were not able to recover a full set of measurements for the Tiupampa condylarths from the literature and also recognise that there are some dubious associations with those specimens. Although perhaps our dataset might prove useful towards resolving some of those issues.

We were not able to recover all the measurements for *Hemithlaeus* from our specimen images (taken during a previous trip to the AMNH) or the published literature. Sarah Shelley speaking here: Believe me when I say this is probably more frustrating for me as someone who has done a lot work on peripitychids although I would predict it would fall out in close proximity to *Protungulatum* and adding it won’t likely do anything too drastic to the morphospaces or disparity analyses.

3. Sampled material of two included taxa (Colbertia and Carodnia) come from a locality (Itaboraí) that consensus now considers to be Eocene in age (e.g., Gelfo et al., 2009; Woodburne et al., 2014; Antoine et al., 2015; Rangel et al., 2019). Carodnia does occur in faunas that are considered Paleocene (the “zona de Carodnia” in Patagonia), although tarsals of the Paleocene species are not known. Colbertia is not known from Paleocene faunas. What is the justification for including either taxon? With regard to Carodnia, if you are including tarsals from Eocene representatives of genera with Paleocene species, why not Azygonyx, Apheliscus, Haplomylus, Chriacus, or Didymictis?

We have revised our sample to include Eocene aged specimens which have generic ranges which extend back into Palaeocene as were able to include most of the taxa suggested here.

4. I don't think there's enough Cretaceous material available to support the role its been given in this study. Of the five Cretaceous taxa included, only Deccanolestes is known from the last 10 million years of the Cretaceous. The remainder includes three taxa from more than 25 ma before the K/T boundary, two of which are non-eutherians. I don't see how you can meaningfully compare Paleocene eutherians with their Cretaceous antecedents on this basis.

We have chosen to retain the Cretaceous taxa. Their inclusion in the morphospaces provides a useful temporal context without any negative implications for interpretation of those spaces. Rigorous statistical tests have been used to account for sampling when conducting the disparity analyses and we are transparent about the limitations of our results (we would like to highlight that no reviewer had any concerns regarding these tests). The distribution of the Cretaceous species in morphospace also highlights the tarsal diversity exhibited by Cretaceous cladotherians despite all being relatively small in size and likely arboreal (which is interesting!). This distribution may well be due, in part, to the timescale and evolutionary history of the sampled taxa, but we are upfront about that fact (e.g. L304, L324) and there is little we can do about it at present beyond resampling procedures given the paucity of fossil tarsals from this time period – would it be better that we ignore this data until we seemingly have enough data to include them or better to include them with clear caveats and conservative resampling procedures? We would not be surprised that if we had not included any Cretaceous species, we would have been asked to by at least one of the reviewers to add them; neither of the other reviewers raised concerns regarding our Cretaceous sample.

5. The “size + shape” analyses don't add anything to the manuscript and should be removed. Aside from confirming the well-known maxim that PC1 represents size in an untransformed PCA, the “size + shape” analyses don't tell us anything beyond what “shape”-only analyses already revealed. The “size + shape” DFA actually performs slightly worse than the “shape”-only DFA in correctly classifying modern taxa. Bizarrely, it flips the largest early Cenozoic taxon in the analysis (Carodnia) from terrestrial (which seems fairly reasonable) to arboreal (which does not).

Agreed—a keen comment! We have cut the ‘size+shape’ analyses and given more emphasis to the DA alongside the PCA on the shape data.

6. The three axis graphs in Figure 2 are visually appealing but very difficult to read, undermining the utility of the figure. Visualizing three axes at once can be valuable when the graph is interactive and the viewer can manipulate and rotate the image. When they're compressed into a static, 2-D image, as they are here, they're simply difficult to interpret. For instance, part 2a doesn't convey how much separation there is between functional groups because the Y and Z axes are smooshed together. A 3D plot is visually more interesting than bivariate plots, but scientifically, its much less useful. Figs. 2a-b should be replaced with bivariate plots, while 2c, showing results of the “size + shape” analysis, should be dropped entirely.

We did provide the data and methods for reproducing the 3D plots which were isometrically scaled to the data but concede that 2D visuals are appropriate and easier to compile.

7. The introduction and discussion lean too heavily on papers that range from 30 to over 100 years old and do not adequately represent current thinking on Paleocene mammals. Aside from Rose (2009), citations supporting the authors claims that Paleocene mammals are viewed as “primitive” or “archaic” are uniformly outdated. Paleocene mammals are not as well understood as Eocene and younger mammals, largely due to more limited material, but I don't think its fair to claim that they are presently viewed as primitive or uniformly generalized.

In revising this manuscript we have edited these sections to hopefully remedy the reviewers concerns with specific mention of more recent studies (e.g. paragraph starting L43, L271) but we do think that the overall narrative of the introduction and discussion does still stands. We, the authors, as researchers who work on Palaeocene mammals and you the reviewers, recognise that there is considerable morphological diversity in the Palaeocene but do not think that Palaeocene mammals get as much recognition outside of the Palaeocene research-sphere as their weirdness might warrant. We think that by comparing Palaeocene species to extant mammals it is easy to come to the conclusion that they are relatively unspecialised based on their robust anatomy, but, as we have shown here there is diversity within that robustness that is not easily comparable to extant forms.

We do state that there have been major developments in our understanding of Palaeocene mammal functional morphology and ecology over the past decade or two (L274). However, many of these more quantitative studies are focussed on a single/select number of species (particularly euarchontans). Studies such as Muizon's seminal monograph on *Alcidedorbignya* provide incredibly detailed anatomical information with functional interpretations but even then, they end up summarising *Alcidedorbignya* as a 'generalised terrestrial mammal with good climbing ability'. Furthermore, a large portion of the taxa on which the 'archaic', 'primitive' etc. stereotype was based have not been given recent reconsideration and end up being described as terrestrial generalists or similar.

A major objective for this study was to provide a means to quantifiably recognise the similarities and differences between as many Palaeocene species as we could (particularly those which have not been well-described and/or are often considered to be 'generalised terrestrial mammals'). We think we have accomplished this objective and in doing so have gained novel insights into Palaeocene mammal palaeobiology that fits with the narrative of our paper. On the one hand Palaeocene mammals are unusual, robustly built animals, more like each other than many extant mammals which has historically lead them to be described as 'primitive' and 'archaic', on the other they are exhibiting diverse morphologies and achieving similar functional diversity to extant mammals (defined and distinguished by their robustness).

Additional author responses to MS comments:

Line 15 in first submission. Your DFA indicates that all of the taxa you sampled were either semifossorial or terrestrial. Does that really imply a radiation in behavior as opposed to morphology?

The disparity analyses quantify the morphospace occupation regardless of how the taxa are classes in that space. Our morphospaces show that morphology correlated with function and behaviour in extant mammals and it is a logical step to apply this to fossil mammals. Furthermore, we do not consider the locomotor predictions to be absolute classifications. General edits to the entire MS are intended to make this clearer.

Line 19. Are they really unsurpassed?

'Unsurpassed' falls pretty low on the list of synonyms for inimitable. We have edited to abstract given the changes to our MS but kept this word because we think it appropriate and conveys what we are trying to say – they were unique, individual, distinctive, quirky.

Line 26. Aren't you arguing the opposite? If Mesozoic mammals were arboreal and Paleocene mammals were semifossorial and terrestrial, Mesozoic mammals didn't really establish the ecomorphological diversity of later mammals.

No, our analysis was limited to cladotherian mammals, here we are referring to Mammalia as a whole. Removed 'extant' from previous sentence to make this clearer.

Line 27. What crown groups are you referring to? There are a lot of crown mammals in the Mesozoic, but there weren't members of most generic crown groups until the later Neogene. You probably mean something in between, but you need to specify.

Prefixed 'crown mammal groups' with 'extant'.

Line 32. I'm not sure how true this is. Several of the taxa included in your analyses have been subject to relatively detailed functional analyses (e.g., Szalay and Decker, 1974; Thewissen, 1991; Argot, 2013; Chester et al., 2015; Muizon et al., 2015). For many of the other taxa (e.g., *Dissacus navajovius*), there are detailed functional studies of close relatives (e.g., Zhou et al., 1992; O'Leary and Rose, 1995), even if the sampled taxon has not been examined.

This is why we say 'relatively little is known'. We have added text to our MS highlighting previous work and the novelty of this study in the third paragraph of the introduction.

Line 40. You've already typecast them as such at least twice in the first two pages of this manuscript.

We use 'archaic' in quote to imply that they are 'so-called archaic'. We have now written this out in full to prevent confusion.

Line 42. Specifically, "they" (the antecedent of which appears to be the "Paleocene mammals" of the previous sentence) would have constituted the entirety of mammalian diversity for the first half of the first third of Cenozoic evolutionary history.

Edited to reflect that we are referring to Palaeocene eutherians therefore not the entirety of mammalian diversity.

Line 50. How is Simpson (1931) "recent" in this context? For that matter, why is a paper on an Eocene genus (*Metacheiromys*) relevant here?

Deleted Simpson and also deleted 'recent' given temporal overlap between all references.

Line 70. Numerous species are measured from specimens where the tarsal bones are accessioned individually. In some cases, it either stated in a publication or in the specimen metadata that the bones are from one individual. In other instances, the bones have been recovered via screenwashing or over several field seasons and are assumed to be associated. Where association is less certain we have compared the sizes of the entire specimens as well as relative proportions, facets etc, and the overall appearance/preservation of the specimens (or photos/models thereof) to decide whether we can composite it for analysis.

Line 106. Why use such a conservative p value? I think its probably a wise idea, but some explanation is warranted.

A p value alpha threshold of 0.005 is a standard threshold. We have provided the BFB values as a more intuitive explanation for that value. The references cited for the BFB also discuss p value thresholds.

Line 121. Is it fair to compare a cladotherian Cretaceous sample that spans almost 50ma with a eutherian Paleocene sample spanning 10ma with a placental modern sample that represents a snapshot in time?

We have provided a more detailed response to this question elsewhere in this document. Briefly, the comparisons are what the fossil record will allow and we have provided rigorous resampling procedures on both the disparity analyses and the subsequent disparity tests.

Line 173. For a PCA, that's not a lot (in reference to the total variance captured by the first three PC axes).

It is lower than the variance captured by an analysis which includes size but a dataset incorporating 29 variables for a broad taxonomic sample and reduced size effects it is a reasonable. Chester et al. (2015) recovered similar axial variances on a tarsal multivariate dataset.

Line 174. Given that the PCA doesn't define clear-cut functional groups, this discussion of factor loadings would be much more interesting in the context of canonical variables.

In revising the manuscript, we have provided detailed axial descriptions for the both the PCA and DA

Line 246. This is a strawman that relies heavily on "historical" citations. The use of "primitive" and "advanced", in particular, have gone out of vogue since the advent of numerical cladistics.

We justified our narrative in previous sections of this response document and have added statements with references on recent studies and where the novelty of our study lies. We have acknowledged that the opening paragraph of our discussion relies on historical references, we clearly state this (L38), and we acknowledge that there are more recent studies which have inferred more specialised anatomies,

behaviours etc for particular Palaeocene mammals. However, a large portion of the taxa on which the 'archaic', 'primitive' etc. stereotype was based have not been given recent reconsideration and end up being described as terrestrialist generalists etc. Just because some work has improved upon an outdated historical stereotype doesn't mean that stereotype no longer exists (particularly among a broader audience who are less familiar with Palaeocene mammals).

Line 256. Do modern mammals with weak to little trochleation of the astragalus have more robust skeletons than mammals with more trochleation of the astragalus? Ptilocercus and Cynocephalus spring to mind, and neither strikes me as particularly robust. Also, how does a loose cruropedal joint explain robustness of the humerus and femur?

We added text to this statement to say 'weak to little trochleation in taxa otherwise thought to be more terrestrially adapted.

We stated elsewhere on the skeleton. The robustness of the femur would serve to stabilise the hip (and possibly the knee – the size of the knee ligaments is harder to infer). Increased robustness of the humerus through the proximal tuberosities, deltopectoral region, medial and lateral epicondyles etc (all of which tend to be relatively large in Palaeocene species) would serve to stabilise the shoulder, elbow and wrist through the various soft tissue attachments.

L261. You should emphasize the convergent/parallel acquisition of more restrictive joint morphologies as a factor that would potentially complicate phylogenetic assessment.

I'm not sure we have enough information to state much more than what we have here without incorporating phylogenetic data. Extant mammals tend to exhibit more restrictive joint morphologies compared to Palaeocene mammals but there a diverse range of morphologies within a more 'restrictive' morphotype

What about the time component to this? Vincelestes is roughly mid-Early Cretaceous in age. Deccanolestes is late Maastrichtian. Your five Cretaceous mammals span more than 50 million years. That, in and of itself may explain why Cretaceous mammals display more disparity than Paleocene mammals.

We have included a caveat regarding this around L324 stating: 'We note there may be a phylogenetic and temporal component to this diversity but the lack of fossil material limits further investigation at present.'

Line 306. Lines 311-313 imply that you are saying here that Paleocene mammals may have remained semifossorial. That isn't coming across here. Be more explicit.

Edited.

Line 308. Please specify what selective pressures were unique to the Paleocene?

References are provided. We are referring to the Palaeocene as a subtropical biome with a climate punctuated by hyperthermals. We mention this elsewhere in the manuscript as well.

Line 308. Given the bulk of the taxa in your sample, could this be the laurasiathere morphotype?

That is an interesting idea particularly given the divergent position of taxa like *Afrodon*. However, if there was a Laurasian signal in the dataset, we might then expect to see some similarities between extant afrotheres compared to xenarthrans and boreoeutherians which we don't. It is an interesting thought to explore though.

Line 316. Can you make some sort of more concrete hypothesis? This sentence is extremely vague. Other than digging ability, what aspect of a robust morphology would have allowed certain mammals to survive the K/T event? Did the shock wave from the impact shatter tight joints? Did osseous stabilizers require more nutrients so that mammals that had them starved after the K/T event? I'm having trouble coming up with a plausible connection.

We have expanded on this section in the manuscript and provided references to add support to our hypothesis (L355).

L334. Again, you're confusing the PCA results, which don't appear to correlate strongly with locomotor mode, with the DFA results, which filter through the data to find a strong locomotor signal.

We are referring to both, which serve different purposes based on how they are constructed.

L355. Why not look at the role of phylogeny in driving patterns of morphospace occupation in modern mammals? It should be straightforward to take your DFA (or PCA) data and graph it with taxa labeled by order instead of locomotion to look for phylogenetic patterns. I tried it and I don't really see any.

Good suggestion! We have done this using Blomberg's K statistic.

L358. Please explain what you mean a bit more.

We do not think additional explanation is required here – we'll use phylogenetic datasets as a source of postcranial characters tied to locomotor behaviour to investigate morphospaces at finer time scales.

Supplement comments:

Regarding dermopterans. They're not that extreme. Certainly, Chester et al. (2015) were able to include dermopterans without much problem.

Chester et al. (2015) included dermopterans within a small euarchontan sample. We found the tarsal proportions (e.g. the elongate calcaneal ectal facet and short calcaneal tuber) caused *Cynocephalus* to fall out as an outlier and distort our morphospace. Given that we're not investigating the tarsal morphology of gliding mammals with regard to Palaeocene species we chose to exclude them. We do note that Beard (Beard 1993) has suggested that plesiadapiforms are dermopterans; however, this has been thoroughly argued against by Bloch (Bloch and Silcox 2001, Bloch et al. 2007). *Mixodectes* has been suggested to have an affinity with Dermoptera, summary in (Rose 2008), but we were not able to recover enough tarsal measurements to include *Mixodectes* in our dataset.

Gaudin et al. (2009) does not support this [lack of a sustentacular facet]. The sustentacular facet in extant pangolins is small and weirdly positioned, but no weirder than in hyracoids.

We have corrected our text to not say the sustentacular is absent. However, we still exclude Pholidota as it is difficult to confidently delimit the proportions of the astragalar and calcaneal sustentacular facets particularly on the calcaneum (considerably more so than *Procavia*) based on the specimens we have access to (see below). Furthermore, we did try and include two pangolins as a preliminary version of our tarsal dataset but they would fall out as extreme outliers due to their unusual facets and ended up impairing visualisation of the main portion of the morphospace (assuming our conservative measurements were captured correctly).

There are some significant exclusions. There are no primates with elongate calcanei (e.g., *Tarsius* or *Galago*), pinnipeds, fossorial "insectivores" (talpids, chrysochlorids), suspensory xenarthrans, etc. I don't think this is an unreasonable sample for inferring metric correlates of locomotor behavior, but I don't think this sample adequately captures placental tarsal disparity. In particular, some broad locomotor categories (aquatic, fossorial, gliding) are absent, and generally extreme morphologies are excluded.

We start this section with text saying what taxa and locomotor groups we're interested in and why in our sample. In the MS we define our extant sample by what it includes and state that we regard our extant and extinct samples as broadly equivalent so we're comparing like-to-like but acknowledge that some extant locomotor categories are not included. We have avoided morphological outliers to allow for better examination of the main region of morphospace occupied. We refer to our extant sample as the extant morphospace, extant disparity results etc. to facilitate readability once we have defined what we mean by extant mammals in relation to our sample.

This is not true. Szalay (1994) illustrates tarsals of a number of Cretaceous metatherians. The tarsus of *Asioryctes* is also known, as are tarsals of some of the Yixian therians.

We acknowledge this error and have corrected this statement. We also note here that we were not able to obtain all of the tarsal measurements required from illustrations in Szalay (1994)

References cited in this review that are not cited in the manuscript:

Antoine, P.-O., G. Billet, R. Salas Gismondi, J. V. Tejada-Lara, P. Baby, S. Brusset, and N. Espurt. 2015. A new *Carodnia* Simpson, 1935 (Mammalia, Xenungulata) from the early Eocene of northwestern Peru and a phylogeny of xenungulates at species level. *Journal of Mammalian Evolution* 22:129-140.

Gaudin, T. J., R. J. Emry, and J. R. Wible. 2009. The phylogeny of living and extinct pangolins (Mammalia, Pholidota) and associated taxa: a morphology based analysis. *Journal of Mammalian Evolution* 16:235-305.

Gelfo, J. N., F. J. Goin, M. O. Woodburne, and C. d. Muizon. 2009. Biochronological relationships of the earliest South American Paleogene mammalian faunas. *Palaeontology* 52:251-269.

Rangel, C. C., L. M. Carneiro, L. P. Bergqvist, É. V. Oliveira, F. J. Goin, and M. J. Babot. 2019. Diversity, affinities and adaptations of the basal sparassodont *Patene* (Mammalia, Metatheria). *Ameghiniana* 56:263-289.

Szalay, F. S. 1994. *Evolutionary History of the Marsupials and an Analysis of Osteological Characters*. Cambridge University Press, Cambridge, 481 pp.

Thewissen, J. G. M. 1991. Limb osteology and function of the primitive Paleocene ungulate *Pleuraspidothierium* with notes on *Tricuspidon* and *Dissacus*. *Geobios* 24:483-495.

Woodburne, M. O., F. J. Goin, M. Sol Raigemborn, M. Heizler, J. N. Gelfo, and É. V. Oliveira. 2014. Revised timing of the South American early Paleogene land mammal ages. *Journal of South American Earth Sciences* 54:109-119.

Referee 3

Overview

I think this is an important and interesting contribution to the field of mammalian palaeobiology, and I would be happy to see this published in *Proceedings B* in due course. The authors investigate the morphological disparity and locomotor mode of Paleocene and some Cretaceous mammals by comparing these taxa to modern mammals. The authors have chosen appropriate measurements from a highly functional limb element which is able to capture morphological disparity across eutherian and stem therian mammals. The element chosen also allows for a good sample of Paleocene mammals to be included. The authors use a

suite of statistical tests to rigorously assess their claims which is commendable (although see comment 1a. regarding their LDA) and strengthens their manuscript.

The paper is well written overall, with points clearly explained. I think that this paper would be of interest to a wide audience, and that Proceedings B is a good fit.

Main comments

Assigning fossils to locomotor groups using an LDA

1a. The **number of wrongly assigned extant taxa in the jackknife classification is a little worrying**. It suggests that entire regions of morphospace for some locomotor groups are populated by only one species, whereby if that species is removed, organisms with this morphology are wrongly assigned. The authors argue that as they use the full dataset when assigning fossils, and that the full dataset has representatives of each tarsal morphology, this is not an issue. However, the authors admit that their sampling of modern mammals is just over 1%, therefore **I believe it would be quite difficult to confidently know that the entire breadth of tarsal morphologies** had been captured for each locomotor mode, and I would suggest that it is unlikely to be true (although I am sure every effort has been made to sample fairly and inclusively, **I imagine it is near impossible to capture the breadth of tarsal ecomorphology with just over 1% of the extant sample. Nearly half of this sample are carnivores**, and I do not agree that the **full ecomorphology of rodents has been captured with 11 species**, despite these species exhibiting a range of locomotor types). **Accurate reconstruction of a fossil species locomotor mode** would be particularly difficult where there is **no overlap with extant species in the CF dataset**, which seems to be the case along **PC1 at least for some of the Paleocene mammals** (although it is difficult to tell in 3D). Furthermore, where a locomotor group is overwhelmingly occupied by mammals from a single clade (i.e. see **cursorial**), I imagine it would be difficult to assign species from any clade that is particularly phylogenetically removed, as presumably phylogeny is not completely removed from tarsal morphology.

Note that the discriminant analyses component of our study now makes use of an RDA using the `klAR` package in R which has given us much greater freedom in how we conduct our analyses. We have provided a full rationale for our methodology in the manuscript and accompanying supplement with appropriate references.

Regarding the resampled misclassification percentages:

When the RDA model is assessed using cross-validation and bootstrapping, the correct classification rates drop but this percentage value is not equivalent to a percentage value failure of the dataset in 1 iteration e.g. ~50% of our 73 extant species are not necessarily misclassified in a single analysis. The percentage value is equivalent to the average of the misclassified observations across the number of folds. Note also that the principal purpose of these resampling procedures is validate the predictive power of the model with regard to how well the optimisation parameters (gamma and lambda) work for the data. The classification rate is calculated for lambda and gamma values generated by the optimisation algorithm to test the predictive power of these lambda and gamma values on subsets of the data. The model will ultimately use the best optimised values. Validation is particularly useful when dealing with large to infinite datasources but that is not the case here - we are sampling from a finite extant morphological dataset. Visual inspection of the morphospaces show they are capturing the diversity of tarsal morphologies and functions exhibited by the extant mammals relevant to this study. Furthermore, predictions of locomotor behaviour for fossil species are derived from the apparent RDA using the full dataset which performs well. Adding additional extant taxa to our dataset will do little to improve the predictive power of the whole dataset, it will however, introduce a greater phylogenetic signal likely reducing the locomotor signal which we want to avoid. We also note here that other locomotor studies on tarsal measurements have record similar cross-validation percentage values (see Ginot et al. 2016) – this isn't to say that this validates our numbers but does show that it isn't an unusual occurrence for discriminant analyses on morphological data.

In revising this manuscript and using an RDA classification procedure, we have provided the posterior values for the training data (extant sample) to better show where the classifications may be more labile. The cross validation and bootstrapping resampling procedure no longer outputs misclassifications (as tables or error matrices) for the extant locomotor groups given that these percentage values are averages for many iterations. We did investigate whether it was possible to call the gamma and lambda values for the

average misclassification percentage for the cross validated and bootstrapped data in order to run a RDA model with those parameters, but we could not find a way to do this.

Regarding our extant sample:

We assert that our extant sample does a good job of capturing the tarsal morphologies and provide good morphospace coverage of the locomotor diversity relevant to the fossil taxa we are interested in. Although it is worth noting we did add an additional taxon when rerunning the revised analyses (*Sciurus* – we had previously struggled to find a specimen in the CMNH collection with both the astragalus and calcaneum. This is a problem we encountered with a lot of the smaller bodied taxa, the carpals and tarsals are often left in the skin).

We acknowledge that we have sampled little over 1% of extant species but when you consider how species are defined and how variable those distinctions can be, the metric of species percentage sampled is perhaps inappropriate given that subtle phenotypic or genotypic specific differences likely show little correlation with tarsal morphology. Reconsidering our sample as a generic percentage (67 unique genera out of a possible 945 [excluding bats, whales etc]) we have sampled just over 7% (we have added both values to our methods text in the supplement). Visual inspection of our morphospace and examination of extreme taxa in those spaces show that we have a good sample covering the breadth of tarsal morphologies exhibited by the extant mammals pertinent to our study. We have provided more detailed morphospace descriptions and figures to better explain our findings with references to existing literature discussing the range of tarsal functional morphologies exhibit my mammals. Where extinct taxa fall away from extant taxa, we do not expect there to be many more extant species that could fill that gap. The divergent positions between the fossils and living mammals is due to a lack of extant taxa with analogous tarsal morphologies. In those cases, the morphospaces still provide a useful functional framework with which to interpret the fossil taxa.

Regarding carnivores and rodents:

Carnivores are incredibly diverse in tarsal morphology and locomotor strategies, possibly more so than any other extant order. Consider the locomotor strategies of a lion compared cheetah compared to an otter compared to a bear compared to a badger compared to a binturong – they are incredibly diverse. Whereas, rodents are relatively more conserved given their smaller size although still exhibit a diversity of morphologies (e.g. see Ginot et al. 2006 – assessed ecomorphology of rodent tarsals using multivariate analyses on a sample of 32 unique rodent species) which we have aimed to capture without introducing too much phylogenetic signal given our focus is not directed towards just rodents.

Regarding reconstruction of fossil species locomotor mode:

We have provided additional text in our methodology explaining how the different morphospaces are constructed and why dissimilarity in the PC spaces is an interesting result. In the discriminant analyses, where fossil taxa this is due to a lack of extant taxa with analogous tarsal morphologies/functions. In those cases, the morphospaces still provide a useful functional framework with which to interpret the fossil taxa in combination with qualitative assessment of the fossils. We have provided more figures of our DF morphospaces to assist in their interpretation and recognition of their limitations regarding the classifications. We have also provided posterior values for the fossil classifications. More, generally in the paper we were and continue to be careful with our language when talking about the fossil classifications and have numerous sections in the discussion (e.g. paragraphs starting L342 and L366) where we explicitly refer to the limitations of inferences based on the RDA alone.

Cursorial classification:

In this revised manuscript our analysis classifies *Ectoganus* as cursorial. This is a surprising locomotor classification for this taxon but does make sense functionally (discourse provided on p17 supplement discussion text). Additionally, we have provided Blomberg's K statistics to show that phylogeny has a limited effect on the extant taxa in our morphospaces due to our carefully curated sampling.

Accurate reconstruction of a fossil species locomotor mode would be particularly difficult where there is **no overlap with extant species in the CF dataset**, which seems to be the case along **PC1 at least for some of the Paleocene mammals**

The authors also discuss the issues surrounding discretising what are often complex locomotor behaviours, and there are a number of classifications which could be considered controversial, or at the least highly variable. For example, *Erethizon dorsatum* could be considered arboreal, and there are a number of Carnivora species that could be considered cursorial. Without collecting additional data, one possible test of how robust the fossil assignments are could be to re-assign marginal taxa to their other possible locomotor modes and see whether this changes the predictions for the Paleocene and Cretaceous taxa. **I think at the very least, the predicted fossil locomotor modes should be discussed with extreme caution.** I do not find it unrealistic that these taxa could have been primarily terrestrial and semi-fossorial, but neither am I convinced by the LDA. This isn't necessarily a huge problem for the paper as a whole. There are a number of interesting results regarding morphospace occupation, and the focus could be centred on these results, with assertions about the possible locomotor modes a more secondary point.

We agree, and appreciate this comment. Caution is warranted when we discuss specific locomotor predictions/classifications for specific Palaeocene taxa. In revising this manuscript, we have improved our classification methodology and supplementary results (posterior values etc) and been cautious in how we discuss our results with regard to the locomotor classifications. The main objective of this study was to be able to quantifiably compare tarsals for a broad sample of Palaeocene mammals. We have tried to remain as circumspect as possible in our wording while also clearly presenting and discussing our findings. And we try to focus on the *spread* of taxa in the morphospaces (which the reviewer recognises as the main thrust of our paper) rather than specific locomotor classifications for extinct species.

1b. Please explain in the supplement the methodology for the resubstitution and jackknife classifications (for anyone unfamiliar with LDAs) and report the percentage predictive accuracy (is 88.89% for the resubstitution classification? Please make clear if so [line 151]). Consider putting the success rate (percentage) in your confusion matrix in the supplement (Tables S3, S4, S9, S10).

Done

PCAs

1c. For the shape dataset, the authors z transform each species to minimise the effect of size. For the size + shape dataset, the authors do not z transform their data. It makes sense not to z transform each species in the size + shape dataset, but as a result PC1, which correlates with tarsal size, accounts for 95% of the variance. It is standard to z transform each trait (rather than species) before subjecting them to a PCA in order to stop one trait from dominating the analysis (i.e. if one trait ranges from 0.1 – 1, but another trait ranges from 1-10, the trait that ranges from 1-10 will carry more 'weight' in the PCA analysis. It might be that PC1 (tarsal size) describes such a large proportion of data not because the relative variance is higher, but because the absolute values are on a larger scale. Presumably, z transforming each trait would not 'remove' the size signal for each species (individual species which are larger will still have relatively larger trait values compared to species that are smaller [similarly, presumably z transforming each species would not completely remove the effect of having traits on different scales – but perhaps the scales are close enough that it is not an issue.]), but it may allow size to be analysed on a level playing field to shape. Body size changes across the boundary could be investigated using a single body size proxy. If there is a reason that the size + shape trait data is not z transformed, please explain.

This point is no longer applicable with the removal of the 'size+shape' dataset. However, a PCA on correlation matrix is equivalent to z-transforming the data by trait. Data with widely varying scales would be better ordinated using a correlation matrix. However, our raw tarsal variables have relatively similar ranges and scales so using a correlation matrix on the 'size+shape' does little to change how taxa are ordinated in morphospace – PC1 still accounts for ~95% of the total variance and is strongly correlated with size.

1d. At the top of page 16, the authors discuss Paleocene morphospace occupation in comparison to their extant sample. It may be worth noting here that there are a number of locomotor types (and therefore presumably morphologies) known among extant mammals that were left out of their analyses (for reasons explained in their supplement). **Therefore, they should be clear to discuss morphospace occupation relative to their extant sampling, not extant mammals more generally.** I understand the rationale

behind excluding them, nevertheless, they are ecomorphologies currently known from Recent mammals which are not recognised among Paleocene mammals and perhaps should be mentioned briefly in the discussion (e.g. fully aquatic, flying, gliding etc [although of course absence of evidence for these morphologies among Paleocene mammals is not necessarily evidence of absence – but we must work with what we have]). Alternatively, discuss around line 325.

Should add a sentence to say when that we refer to extant mammals in regard to our results we referring to out extant mammal sample

Yes, good suggestion – we have been more cautious in our language to specify we are comparing Palaeocene mammals to extant taxa included in our sample e.g. L367.

Data Visualisation

1d. Although I appreciate the time and effort that has gone into producing visually appealing 3D figures (Fig. 2a-c), I find it near impossible to accurately assess the spread of points in morphospace. In particular along PC2 and PC3. I would prefer to see 2D figures in the main text. Perhaps you could consider moving the 3D figures to the supplement (failing this, ensure there are 2D versions of all your 3D main text figures in the supplement [e.g. like Figure S1]). Particularly, I would like to see 2D figures for your Canonical Functions morphospace (these are not in the in the supplement either). Although the proposed fossil locomotor groups and extant locomotor groups are shown in sup. fig. S2, these are the PCA analyses, and the Paleocene taxa show nearly no overlap with any modern groups. **It would be interesting to see the same groupings (fossil and extant mammal assigned groups) in CF morphospace** (consider using the same colour for locomotor mode, but a different shape for fossil and extant mammals – i.e. green circle = extant arboreal, same green but diamond = fossil arboreal).

The morphospaces are now illustrated in 2D with additional versions of the Discriminant Analysis morphospace plots.

Line by line suggestions

2a. Throughout: 'Paleocene' is misspelt throughout. Even in English, Paleocene is never spelt with an 'a' as it is the Pal-Eocene (the 'old Eocene').

We think that the spelling of 'palaeo' has become a matter of opinion at this point given how language evolves to forget or accentuate inconsistencies and regional idiosyncrasies. British English spelling retains the second 'a' in 'palaeo' even if technically wrong based on Schimper's etymology of the word (although we have never been able to access the original publication to verify this). Recent publications in Proceedings B spell 'palaeo' with the second 'a'. We have retained the second 'a' given that this is a British journal and the first author (Shelley) is a British English writer and have used British English spelling elsewhere the manuscript; however, we will leave the final decision to discretion of the journal editor.

2b. Line 74: 'Extant taxa were classified into six locomotor groups following the literature'. As this is such an integral part of your study, I would list the locomotor groups here in the main text.

We are running very close to the word limit and decided to leave the text as is. The locomotor groups are clearly illustrated and listed with the figures and the information is easy to find elsewhere in the manuscript.

2c. Line 76: '...and PanTHERIA Database'. Consider 'and **the** PanTHERIA Database' Done

2d. Line 81: '...as a means for mitigating against...'. Doesn't read well. Perhaps '**in order to mitigate**'.

N/A – text removed during revisions

2e. Line 124: '...dispRity' package55'. 55?

Corrected

2f. Line 134: '...tests to further test...'. Consider '...tests to further **assess**...'

Done

2g. Line 232: '...reveals the same range trends...' This doesn't read well.

N/A – text removed during revisions

2h. Line 254-259. Long sentence. Consider breaking up.

Done

2i. Line 264: '...such as the double pulley astragalus observed in Artiodactyla.' Reference?

Gingerich 2001

2b. Line 274: Word missing from following sentence: '...resulting in greater observed range morphospace occupation (Fig. 4).'

Corrected

2k. Line 297: 'Then following the K-Pg extinction...'. Sentence beginning with 'Then'.

Corrected

2l. Line 312: '...but the animals did not...'. Consider '... but the animals **may not have...**'.

Done

2m. Page 7 of supplement. Word missing from following sentence: '...when resampled this rate drops due to areas of previously morphospace being excluded.'

N/A – text removed during revisions

2n. Page 7 of supplement. Word missing from following sentence: '...but for the purpose looking at eutherian locomotion at a broader scale...'

Corrected – 'but for the of purpose of looking'

2o. Page 8 of supplement: '...don't necessarily reflect inaccuracy in the data.' Change to '...**do not** necessarily reflect inaccuracy in the data.'

2p. Page 8 of supplement: '...note that although extant hippotamids spend much time in bodies of water they can't actually swim.' Change to '...note that although extant hippotamids spend much **of their** time in bodies of water they **cannot** actually swim.'

N/A – text removed during revisions

2q. Page 9 of the supplement. 'Variance dissimilarity decreases from the Cretaceous...'. Change to '**Morphological** dissimilarity decreases from the Cretaceous..'

N/A – text removed during revisions

2r. Page 9 of supplement: '...between the Palaeocene compared to the extant sample.' Change to '...between the Palaeocene **and** the extant sample.'

N/A – text removed during revisions

2s. Page 14 of supplement. Missing space in the following Figure caption: 'Figure S2. Tarsal 'shape'morphospace plots'.

N/A – text removed during revisions

Appendix E

Dr. S L. Shelley
School of GeoSciences
Grant Institute
University of Edinburgh
Edinburgh
EH9 3FE

Sarah.shelley@ed.ac.uk

15th February 2021

Dear Editors,

In September of last year, my co-authors, Stephen Brusatte and Thomas Williamson, and I submitted a manuscript titled 'Unique locomotor habits in Early Paleocene mammals after the end-Cretaceous mass extinction' (RSPB-2020-2330) for consideration at Proceedings of the Royal Society B: Biological Sciences. We are pleased that two of the three reviewers find our manuscript ready to be published, with a few minor revisions, which we have addressed. The second reviewer had more detailed comments which seem to be the one hurdle to having our paper accepted. We have revised our study following the comments and suggestions of reviewer 2 and we are now ready to submit our revised study.

Our paper, for the first time, examines the locomotor diversity of a broad diversity of Paleocene mammals living after the end-Cretaceous extinction, including the ancestors of living placental mammals. Our study finds a novel and interesting result: Paleocene mammals exhibited their own diverse and distinctly robust morphologies that belie their historical stereotype as poorly adapted precursors to the evolution of the extant mammal orders. Rather, they flourished in a post-extinction evolutionary radiation and prospered through their own bespoke body plans.

The principal concern raised by reviewer 2 centered around the use of the discriminants analysis model to predict locomotor behaviour in fossil eutherians using an extant placental sample. We have addressed this concern with the addition of extant marsupials to our dataset and now provide supplementary morphospaces and disparity analyses with the associated tests on the data when the extant sample is differentiated into placental and marsupial subgroups. Overall, the results of our study still hold, Paleocene mammals occupy a significantly separated from extant placental and marsupial mammals. The addition of marsupials has altered some of our locomotor predictions for fossil taxa but has provided better context for interpreting the functional morphology of the fossil species. We would also like to reiterate that the main purpose of our paper is not to predict exact locomotor habits of extinct mammals, but to assess the diversity of locomotory habits, and how these changed after the end-Cretaceous extinction, in the same way previous studies have assessed diversity of species, body size, and morphological disparity in Paleocene mammals.

With these revisions, we feel that our paper is now substantially stronger. We thank the reviewers for their detailed and helpful comments, the editors for inviting these reviewers and feel that Proceedings of the Royal Society B: Biological Sciences would be an ideal fit for our paper.

Yours Faithfully,

On behalf of the authors

Sarah L. Shelley

Appendix F

Unique locomotor habits in Paleocene mammals after the end-Cretaceous mass extinction

Response to comments:

Editor:

Currently the paper does an excellent job of demonstrating the links between tarsal morphology and broad locomotor habit in extant mammals, but I agree with reviewer 2 that it **falls short of providing a clear predictive picture of the locomotor habits of extinct taxa**. It is of course interesting that **extinct forms appear different to extant forms morphologically, but this view was qualitatively supported to an extent previously**.

We have expanded our extant sample to include a number of marsupial species across the locomotor groups following the suggestion of reviewer 2. Overall, our results hold. Paleocene mammals are significantly separated from both extant marsupials and placentals and the disparity trends remain the same. The addition of marsupials had altered some of our fossil predictions and some of the problem taxa highlighted by reviewer 2 are now classified in way that is more in line with the existing literature (more details are provided below).

That said, We reiterate that the main purpose of our paper is not to predict exact locomotor habits of extinct mammals, but to assess the *diversity* of locomotory habits, and how these changed after the end-Cretaceous extinction, in the same way previous studies have assessed diversity of species, body size, and morphological disparity. We do not intend for the locomotor predictions to be taken as absolutes. They should be considered in conjunction with the anatomy, morphospaces and RDA posterior values to give a more nuanced insight into the locomotor behaviours exhibited by Paleocene mammals. We do state in the manuscript that discretely categorising locomotor behaviours is difficult but nonetheless it serves a purpose. We also note that the discriminant analysis that predicts locomotor categories is one of two multivariate methods we use to produce morphospaces to assess locomotor diversity. Our PCA does not involve locomotor category prediction at all. The fact that our PCA and discriminant morphospaces and disparity results are similar shows that our methods and overall conclusions are not compromised by the fact that one method produces discrete locomotor category predictions that might be subject to error.

We would like to address the statement that the uniqueness of Paleocene mammals has been qualitatively supported to an extent previously. Anatomical and functional insights on Paleocene mammals have previously either been limited to qualitative inferences or focused on a specific extinct taxon or a small and/or closely related sample of extinct taxa. These findings challenge the concept that Paleocene mammals were non-specialised and indicate a wider range of morphological diversity than previously alluded but this has not been tested or quantified. The novelty of our study is derived from the fact that we have incorporated a broad taxonomic sample of 40 Paleocene species including many taxa which have not been well-described in the recent literature or quantitatively analysed. Furthermore, the inclusion of many Paleocene taxa allows for quantifiable comparisons and differentiation between taxa within the functional context provided by the extant sample and the temporal context provided by including the Cretaceous taxa (this was a major impetus for this study). Where Paleocene taxa have been described (e.g. *Purgatorius*) our results do match up with anatomical descriptions and functional inferences supporting our results. Our study goes further by providing quantifiable context to qualitative statements and offers a broader context to fossil interpretations by allowing for comparisons to a large sample of other Paleocene taxa.

The issue comes with the lack of a clear translation to differences/similarities in function and locomotor ecologies.

We do not understand this statement. We have provided detailed descriptions of our morphospaces and the morphologies associated with different locomotor behaviours. Where fossil taxa diverge from extant taxa (which in itself is an interesting result), their tarsal morphology can still be considered and interpreted within the functional context of the morphospaces. We have also presented several hypotheses in the discussion section as to why this might be happening.

A further minor issue for me, not really discussed by the reviewers, is the addition of the long bone robusticity analysis (Fig 2). I'm not sure this is particularly robust with an $N = 4$ for the extinct taxa, and I don't think its legitimate to extend the best-fit for this group beyond the range of those four taxa.

This is a good point and we understand. We have removed the best-fit line for the Paleocene sample and moved the regression analyses to the supplement in order to make space within the main text for other key suggestions put forward by the reviewers. We want to retain this information in the supplement because it supports our interpretation of Paleocene placentals having robust skeletons compared to modern taxa.

Referee: 1

I have attached the manuscript with some very minor grammatical/punctuation suggestions annotated throughout, and I encourage the authors to carefully run their eye over the supplementary text as well in this regard. I do not feel the need to see any further revisions (should they be required), and I look forward to seeing the paper in print.

We are pleased this reviewer found our manuscript ready to be published and thank them for their helpful comments and suggestions.

Responses to comments in PDF:

L203 (now L192) Do you mean digitigrade?

Not really, some Paleocene taxa are digitigrade but in general, most have a heel-elevated stance (somewhere between full plantigrady and digitigrady).

L264 (now L252) non-avian?

We do mean to say all dinosaurs were dominant prior to the extinction

Referee 2:

The revised version of this manuscript is a significant improvement over the first draft. The inclusion of an expanded Paleocene sample is a welcome addition, and it paints a clearer picture of the distribution of Paleocene eutherians in tarsal morphospace. The manuscript now also strikes a more appropriate balance between the PCA and DFA analyses. However, I think there are still **significant issues that need to be addressed, largely centered on the validity of the locomotor predictions made by the DFA**. I've outlined my primary concerns below, with additional comments included in the attached pdfs.

1. While the DFA does an impressive job of separating extant taxa by locomotor groups, I'm still concerned by how many fossil taxa fall outside of the discriminant morphospace defined by extant taxa. Many of the "arboreal" and "semifossorial" taxa, in particular, seem to be well outside the envelope defined by all extant taxa, which suggests that their morphology may be too divergent for locomotor patterns to be accurately inferred based on the morphology of extant taxa. Even when fossil taxa do fall within the extant morphospace, many are classified into groups that seem nonintuitive, at least based on the DF1 versus DF2 plot. The "terrestrial" taxa, for instance, include several taxa (Protungulatum, Goniacodon, Vincelestes) that look like they would be more appropriately classified as semifossorial or semiaquatic based on where they plot in discriminant function morphospace.

I think you need to investigate whether it's appropriate to use DFA to classify unknowns that fall well outside the range of the known comparative sample. **As great a job as the discriminant functions do of accurately classifying extant eutherians, I'm not sure it's an appropriate sample for classifying Cretaceous and Paleocene mammals.** You wouldn't use a regression derived from shrews to estimate the mass of an elephant. I'm concerned that what you're doing here is more or less equivalent.

We understand the reviewer's comments, and note that this issue applies to any discriminant analysis: there will be unknown samples that do not fall within the range of all of the known samples in the analysis, and thus there will be additional uncertainty in accurately predicting the assignment of these unknowns. If all we were trying to do was to predict the locomotor category for each extinct mammal in our analysis, then this would be a more concerning issue. But this is only a small part of our paper, and our focus is instead on the *diversity* of locomotor types, which is well encapsulated by our morphospaces. It is this spread of diversity, not whether, say, *Purgatorius* is specifically predicted to be terrestrial or arboreal, that is important. We have been very clear about our methods and the uncertainties in assigning locomotor categories.

The RDA uses all the discriminant functions, not just the two/three axes visualised therefore the morphospaces are only displaying a portion of data and the position of taxa in a given morphospace may not intuitively match with its predicted classification. When investigating this further by looking at the discriminant functions for each axis for a given taxon it is often more clear why they've been given their classification. For example, *Goniacodon* has a relatively low DF1 value for a terrestrialist, however, its position along the other 4 functions causes it to be classified as terrestrial. This overlap is also reflected in the priors for the predictions. We've made it very clear in the MS that the predictions are only predictions and should be considered alongside the morphospaces and existing literature (e.g. L323).

Regardless of the classifications, the morphospaces provide a graphical means for inferring tarsal functions for the fossil taxa and highlight useful extant analogues. Working up from first principles, we selected an appropriately bracketed extant sample covering a wide range of tarsal morphologies, we've taken detailed measurements of those tarsal and demonstrated that we've accurately captured tarsal morphologies associated with locomotor behaviour. That some Paleocene species fall in divergent positions does not invalidate the method or render the classifications un-useful when taken into consideration with the morphospaces and other lines of evidence (as outlined in the text).

We find the ‘shrews to elephants’ comment misleading. We are not directly comparing elephants to shrews using a regression, we are using a multivariate dataset that spans ‘shrews’ to ‘elephants’ to analyse a similar range of fossil taxa of varying body size. This is a widely used practice and is at the heart of any discriminant analysis. That some of the fossil taxa fall outside the range of the modern sample ***is in itself interesting*** and a major conclusion of our study. We’re sorry, but it seems like the reviewer is saying that our results would only be valid and publishable if all of the extinct mammals fell within the morphospace ranges of the extant mammals. But we’re dealing with fossils, and this is not realistic, and it’s also not what the data show. The Paleocene mammals in our dataset, in some cases, are truly different from extant mammals and that is important!

One potentially relevant issue is whether extant placentals are the ideal extant comparator for Cretaceous “cladotheres” and even some Paleocene eutherians. Horovitz (2000) notes that some Cretaceous eutherians lack important features that characterize most or all extant placentals, including a pulley-shaped astragalar trochlea, complete encasement of the astragalus by the fibula, presence of a plantar tubercle, and a straight calcaneal tuber. In this regard, Cretaceous “cladotheres” and Paleocene eutherians are more similar to extant marsupials than placentals. A modern sample that includes both marsupials and placentals would be more appropriate than a placental-only sample. I would recommend adding a sample of marsupials across the same locomotor categories to the modern sample. If you can still get good separation and classification of modern locomotor categories, the fossil categories would be more robust.

We appreciate the suggestion and feel that it is very helpful and has made our manuscript stronger. We were able to include a sample of extant marsupials for each locomotor category following the suggestion of the reviewer. We now provide supplementary morphospaces, PERMANOVAs and disparity analyses that differentiate the extant sample into marsupials and placentals. The fossil classifications are derived from an extant sample which incorporates marsupials. With the addition of marsupials, our results maintain statistically significant morphospace separation between groups and a high correct classification rate for the extant taxa.

“Horovitz (2000) notes that some Cretaceous eutherians lack important features that characterize most or all extant placentals, including a pulley-shaped astragalar trochlea, complete encasement of the astragalus by the fibula, presence of a plantar tubercle, and a straight calcaneal tuber. In this regard, Cretaceous “cladotheres” and Paleocene eutherians are more similar to extant marsupials than placentals.”

This statement is not strictly true. Paleocene mammals do have eutherian features, and morphologically they look far more placental-like than marsupial-like. Extant marsupials achieve pedal mobility primarily at the upper ankle (cruropedal) joint whereas placentals achieve throughout the tarsus to varying degrees depending on requirement. Paleocene mammals possess a high degree of mobility throughout the ankle joint – they possess a high degree of mobility at the upper and mid ankle joints like extant marsupials but also possess features associated with more forceful and efficient tarsal movements. It is this combination and associated morphologies that sets them apart from marsupials and placentals.

2. Related to the first point, a number of classifications of fossil taxa are unexpected and conflict with previous assessments of the locomotor ecology of these taxa. You’ve focused on three taxa (Purgatorius, Ectoganus, Dissacus), but there are several other classifications that are equally, if not more concerning, including Pachyaena and Kulbeckia, both of which have been considered cursorial based on holistic assessments of postcranial anatomy (O’Leary and Rose, 1995; Chester et al., 2010; Averianov and Archibald, 2017). In these and other instances,

the fact that overall postcranial anatomy does not support the locomotor classification derived from the tarsal DFA should be a cause for concern, as it serves as a partial test of the validity of the DFA classifications.

Additionally, I'm concerned by the number of changes in predicted locomotor ecology between the first draft and the current draft. Of the 25 fossil taxa that were included in the original draft, the predicted locomotor group of 7 (28%) has changed in the revised draft. I assume this reflects the change from LDA to RDA, but it doesn't leave me with much confidence in the reliability of your results.

We will state, once again, that the main focus of our paper is not predicting the locomotor categories for Paleocene mammals; instead, we are looking at the diversity of locomotor abilities based on functionally useful ankle metrics. When it comes to assigning taxa to categories, discriminant function analyses always have uncertainty and will always have taxa that are incorrectly assigned, hence the need for the bootstrap analyses and cross validation.

Regarding *Pachyaena*:

Firstly, our RDA now classifies this taxon as terrestrial. The addition of marsupials with relatively more robust tarsal morphologies in a variety of locomotor groups appears to have helped.

Secondly, we have used *Pachyaena gracilis*. The conclusions outlined in O'Leary and Rose 1995 pertain to *Pachyaena* at a generic level but as their figures illustrate there are notable differences in size and morphology between the species.

Our results are concordant with the descriptions and conclusions in O'Leary and Rose 1995

- In our PC morphospaces, the mesonychids are divergent for reasons noted in the supplementary text - the unusual combination of Paleocene mammal-like features (high degree of mobility between astragalus and calcaneum) with some cursorial features such as elongation of the calcaneal tuber.
- In our DF morphospace, *Pachyaena* is positioned between extant carnivores and ungulates with DF values that correspond to the morphology.

Regarding *Kulbeckia*, now classified as terrestrial:

Averianov and Archibald (2017) write:

"The long and distally fused tibia and fibula indicate a cursorial mode of life. The upper ankle joint of Kulbeckia has the complete separation of medial and lateral astragalotibial articulations. This character might be also present in Zalambdalestes. The astragalar canal is present in Kulbeckia but lost in Zalambdalestes. The calcaneus has a cone-like calcaneoastagalar facet, long calcaneal tuber, and a large and nearly transversely oriented calcaneocuboid facet. The long calcaneal tuber is another evidence of cursorial specialization of Kulbeckia."

These features are captured by the morphospace position of *Kulbeckia*. It falls close to *Pedetes* in DF morphospace, and in proximity to a cluster of scansorial/arboreal taxa and is now classified as terrestrial.

We have been very clear that the locomotor predictions should be taken in conjunction with the morphospaces, posteriors etc. We don't mean to be rude, but we feel like our entire paper is being judged on whether the locomotor category predictions for each fossil taxon are perfect, and not on the main thrust of our paper, which is on how locomotor diversity changed after the end-Cretaceous extinction and in the lead-up to extant placental diversity.

Regarding the number of changes in predicted locomotor ecology between the first draft and the current draft:

We changed method and added data. RDA is a machine learning method, the input has changed, therefore the output likely will to. It is up to the user to determine the integrity of the input data and interpret the output. We hope that the additional of marsupials has alleviated the reviewers concerns in this regard. Overall, the morphospaces are still very similar and interpretations of the axes are the same, the classifications are changing a bit but the differences between analyses are reflected in the posteriors. It would be suspicious if the results didn't change a bit. The fact our data works well classifying the extant sample, provides greater confidence in our fossil classifications.

3. The key paragraph in the discussion from lines 339 to 361 requires extensive modification. First, you're trying to have it both ways. On the one hand, maybe the DFA is accurately classifying Paleocene mammals and they were robust because they were semifossorial. On the other hand, maybe the DFA isn't accurate and Paleocene mammals only looked similar to modern semifossors due to robusticity unrelated to locomotor ecology. The latter would undermine the significance of the DFA.

Now L328. We're not trying to have it both ways. We're raising two different hypotheses to explain our results, because we do not put absolute confidence in the results of the **predictive** model. We are being honest about the limitations of the locomotor category predictions here—exactly what the reviewer wants us to be doing.

The argument that a lack of tightly interlocking joints was beneficial during the early stages of the post-K/T eutherian adaptive radiation is interesting, but it needs quite a bit of improvement. First, as written, it comes across as almost Lamarkian (selection for acquired myological adaptations). I don't think that's what you're arguing, but it needs to be rewritten for clarity. Additionally, its not clear why this would specifically benefit eutherians. Paleocene multituberculates and metatherians have similarly loose joints (if anything, they're less tightly interlocking). Why were they unable to radiate to the extent seen in eutherians?

Thank you for these suggestions. We have edited this section, however, if the reviewer could be more specific as to where they think it needs improvement/rewriting, we are open to suggestions.

Metatherians achieved tarsal mobility at the upper and mid ankle joints whereas Paleocene eutherians were experimenting with different morphologies throughout the tarsus and evolving ways in which to achieve more forceful and efficient locomotion (long levers, different loading strategies etc) but without reducing upper/mid ankle mobility. It is possible other mammals were doing this too, we haven't tested this as it's beyond the scope of this current project. We're not saying mobile joints were the principal driver behind the eutherian adaptive radiation but coupled with other physiological and adaptive traits (e.g. changes in diet, body size), they may have helped placentals exploit new niches. We have added a sentence to this effect in the MS (L354).

Although it may be a moot point given the problems with the DFA classifications, you might want to consider an alternative link between postcranial morphology and eutherian success. Robertson et al. (2004) have argued that burrowing and submersion were important

mechanisms for surviving the initial K/T event. If many K/T interval eutherians were semifossorial as suggested by the DFA results, this might explain the differential survival of eutherians.

We do say this, but have added an explicit sentence to that effect and cite Robertson et al. (2013) (L332). We further expand on this by hypothesising that the loss of arboreal habitats following the impact may have favoured the survival of more ground-dwelling species.

4. The basis for assigning extant taxa to locomotor groups also needs to be more rigorously documented. While you give a general overview, it's impossible to determine how the locomotor group of any particular taxon was determined. For instance, was *Atilax*, which is a marsh dwelling herpestid that falls out close to semiaquatic taxa in the DFA determined to be terrestrial based on the literature, comparisons to closely related species, or observations based on behavior. The same question could be asked of *Pekania pennanti* (you should update the generic assignment, by the way) which you classify as terrestrial. It falls out with scansorial/arboreal taxa in the DFA and Heinrich and Rose (1995) classify it as scansorial.

We state at the end of the paper that the classification of locomotor behaviour into discrete categories is not ideal but serves a purpose. This is an issue that concerns every study of mammal locomotion that discusses locomotor categories. We have researched every taxon included in this dataset and made a decision based on the information available. For example, *Pekania* fits the terrestrial locomotor category as defined here (spends majority of time on the ground for feeding, shelter, rest, reproduction and socializing but may be able to climb, run, swim, dig on occasion). Observation of *Pekania* shows it spends the majority of its time on the ground and Powell (1981) in *Mammalian Species* notes that its time spent climbing is often overstated, however, it can also invert its hindfeet to descend surfaces headfirst. Our RDA classifies *Pekania* as scansorial and the posteriors reflect the overlap between terrestrial and scansorial behaviours. Given the overall success rate of the RDA, a few borderline classifications based on overlapping life behaviours are immaterial. Regarding the marsh mongoose – we said it was terrestrial for the purpose of the RDA lives on land but can swim as noted in the literature and can also dig – a terrestrial generalist classification makes the most sense. Our morphospaces capture these overlapping behaviours and the priors support the classification based on the model.

5. I'm not sure how appropriate it is to apply disparity analysis to DFA scores is. I'm having trouble finding examples of similar analyses in the literature, and it doesn't make much sense intuitively, especially given the questions surrounding the applicability of the DFA to the fossil sample. If the DFA loadings don't reliably correlate with locomotion in Cretaceous/Paleocene mammals, I'm not sure what disparity in those loadings would even mean. I would suggest adding citations to the literature used to determine that this analysis was appropriate.

The disparity metrics we have used quantify the distribution of datapoints in a scatter plot (morphospace) so there is no reason for it not to be appropriate. The DFA data shows how the fossil mammals are most similar to extant mammals so in this regard we think the comparable disparity levels are important to include. This is common practice in the literature. For example, in their major paper on Mesozoic mammal evolution, Chen and Wilson (2015) used variance metrics on Linear Discriminant scores.

6. Some statistical information is missing or at least not obvious. With regard to the DFA, how many terms are used in the classifying equation? In the Supplemental Information, scores for 5 linear discriminant functions are provided. Did all 5 contribute to the equation?

Added (back in) (L123). We previously removed to meet word limit as using all axes is the standard.

In the previous draft, you stated that "We used all 29 PC axes to calculate morphological disparity". That statement has been removed, but nothing has replaced it. How many PC and LD scores were used in the disparity analyses? Was anything done to scale the axes to account for the different amounts of variation explained by each axis? For instance, PC1 accounts for ~23% of total variance, while PC29 accounts for 0.084%. Was variance along these axes treated equivalently by the disparity analyses? I would think that variance along PC1 should be given much more weight than variance along PC29.

We have added a statement regarding the number of PC axes back in, see above (L123). Scaling the axes is rarely if ever done in disparity analyses. That's why we often take sum and products of the range/variance. If we do a product, we would normalise by taking the root of however many axes used. But regardless, what we did is standard and serves to describe morphospace occupation.

Responses to comments in MS PDF:

L172 (now L158) Could you clarify what this means. Are they significantly separated along each axis, or is this solely a cumulative assessment? If it's the latter, is there significant separation along each of PC1, PC2, and PC3?

The PERMANOVA provides pairwise comparisons of subgroups of the PC scores (for all 29 axes). We are running the PERMANOVAs on all 29 axes i.e. 100% of the variance.

L192 (now L178) I don't think this assessment of the functional significance of DF2 is accurate. Does an animal like Potto really have more rapid but less powerful actions than an artiodactyl?

It is correct, *Perodicticus* possesses an astragalus and calcaneum with a more equal length effort and lever arm corresponding to a joint which favours more rapid movement over forceful movement. Many prosimian primates use leaping behaviours, *Perodicticus* is one of very few prosimian primates not known to leap but it's tarsal morphology may be reflecting its evolutionary history and hasn't needed to modify due to the rather motile existence of the animal. Additionally, the potto is omnivorous and will hunt (creep up and rapidly grab) fast prey items like lizards and bats – in that sense a tarsus capable of short, rapid bursts of movement may be useful for lunging.

L194 (now L181) Taxa that do a lot of inversion/eversion at the subastragalar joint have substantial asymmetry in the length of the astragalar versus calcaneal ectal facets. If DF3 reflects inversion and eversion at the subastragalar joint, shouldn't A10 and C8 have meaningful loadings? Both have essentially no relevance to DF3.

This is because there is not a clear trend between facet asymmetry and locomotor behaviour given locomotor group is an a priori in determining the distribution of taxa in morphospace.

The below boxplot illustrates this. Other variates better capture tarsal inversion/eversion that define and distinguish the all the mammals sampled across all the locomotor groups.

1. Boxplots (with 95% confidence intervals) of the difference between astragalar and calcaneal ectal facet length (z-transformed) for mammal sample grouped by locomotor mode

L200 (now L189) This suggests that the morphology of Cretaceous/Paleocene mammals may be too different from living mammals to make reliable functional inferences.

We disagree with this statement. The position of the Paleocene taxa in morphospace can still be interpreted within the functional framework provided by the extant taxa. The kinetic principles are scalable. The divergent positions for some species (like the mesonychids) reflects their unusual anatomy which is mentioned in the literature. For example, O’Leary and Rose 1995 compare *Pachyaena* to extant ungulate and carnivoran cursors and conclude *Pachyaena* is a weird amalgam of both with some ‘condylarth-like’ features as well. Our morphospaces agree with these observations and provide quantitative context on how this taxon differs from other Paleocene taxa and extant mammals.

L216 (now 204) Is this meaningful? Essentially, all this says is that the centroid of fossil semi-fossorial taxa is significantly different from the centroids of fossil terrestrial and arboreal taxa. There still appears to be substantial overlap between individual taxa.

It’s an appropriate test for the data and provides a numerical value on group differences with data permutations to avoid possible biases. In conjunction with the other analyses it provides a meaningful contribution to the paper.

L224 (now L211) This isn’t an accurate summary. The DFA shows many if not most fossil mammals outside (sometimes substantially) the functional morphospace of extant eutherians. The fact that the DFA classifies all of the fossil taxa into locomotor groups doesn’t mean that those classifications are correct.

This section has been edited to better summarise our findings.

“The LDA visualisation and RDA classification show that the fossil species share similar tarsal functions to extant mammals in terms of stance and movement and loading through the pes, but are often exploiting locomotor strategies using different combinations of morphologies.”

L227 (now L213). If this is going to be included in a summary paragraph, it needs to be stated more clearly in the preceding paragraphs.

It is starting on L152 with reference to the supplement.

L265 (now L253) This still comes across as a strawman argument, and I'm not sure it needs to be here at all. The framing device of Paleocene mammalian tarsals not being considered collectively is more than adequate as a justification for this study.

We've previously explained our reasoning for setting up the discussion as we have and provide text discussing more recent advances in the field. Nevertheless, we have chosen to keep the narrative as it is and note that neither of the other reviewers took issue with how we have framed our discussion.

L276 (now L263) This sentence is trying to do way too much, and it ends up so hedged with caveats that it minimizes the significance of the study. Change it to something simpler like:

"Furthermore, the inclusion of many Palaeocene taxa allows for quantified comparisons to be made between taxa."

Not sure what you mean by 'hedge with caveats'. We have included extant taxa to provide functional context based on observed behaviour and Cretaceous taxa to add temporal context and provide *some* information about what happened over the K-Pg boundary and the radiation of eutherians.

L286 (now L274) This seems to suggest that Paleocene eutherians were more skeletally robust than Cretaceous therians. Can you cite evidence in support of this assertion?

Not at present. We are not aware of a published study that compares Cretaceous therians to Paleocene eutherians and provides data that quantifies (or can be used to quantify) robustness. Ideally, would require measurements for calculating body mass estimates using the Campione and Evans equation and limb bone measurements. If the reviewer knows of any published studies, we are open suggestions.

This statement also implies that Paleocene eutherians were distinctive in having relatively loose joints. At least with regard to the tarsus, I don't think that's likely to be the case. Multituberculates and metatherians have ankles that, at least to my eye, look even less stable than early eutherians.

Extant marsupials achieve pedal mobility primarily at the upper ankle (cruropedal) joint and mid-ankle joint (between the astragalus and calcaneum) whereas placentals achieve a greater amount of mobility through the tarsus at the upper, mid and lower ankle joints to varying degrees depending on requirement (often less at the upper joint). Paleocene mammals fall somewhere in between, they possess a high degree of mobility throughout the ankle joint – with a high degree of mobility at the upper and mid ankle joints like extant marsupials but are also experimenting with more forceful and efficient locomotor styles (albeit, not to the same

extent as placentals and while maintaining a more mobile tarsus). It is this combination of morphologies that sets them apart from marsupials and placentals.

Finally, can you present evidence that non-tarsal joints were relatively loose? In particular, since you've presented evidence that Paleocene eutherians had robust humeri and femora, can you present evidence that their shoulders, elbows, hips, or knees were less tightly interlocking than in extant eutherians?

We cited papers that provide anatomical descriptions of Paleocene species (L276) a quantitative analysis is beyond the scope of the current project.

Line 297 (now L286) I'm really having trouble seeing this. There is extensive overlap between Cretaceous and Paleocene taxa in both the PCA and DFA analyses. Deccanolestes has an extremely low score for PC1. Otherwise, the Cretaceous taxa seem to occupy a portion of the morphospace occupied by the Paleocene taxa.

Separate as found by the PERMANOVA when using all axes. The three plots are more useful in this regard for illustrating the dimensionality of the data. See Figures S7 and S10. Can also generate interactive plots in R using the rgl package.

L298 (now L287) How do you reconcile classification of *Kulbeckia* as arboreal with the consensus (Szalay and Sargis, 2006; Chester et al., 2010; Averianov and Archibald, 2017) that *Kulbeckia* was cursorial, a conclusion supported by studies of more complete skeletons of younger zalambdalestids?

No longer applicable given terrestrial classification for *Kulbeckia* but also see comment on *Kulbeckia* above.

L300 (now L289) What non-cladotherians disappeared at the K/T boundary? Multituberculate diversity was certainly impacted, but did it affect locomotor diversity? Otherwise, non-cladotherian groups present in the Maastrichtian survived the K/T boundary (meridolestans, monotremes, gondwanatheres). Jehol-era mammals may show considerable locomotor diversity, but most of those groups were extinct long before the K/T boundary.

True, we have edited this text in the revised version.

L309 (now L298) This is really awkward phrasing. If you mean size disparity, say size disparity. If not, please explain.

We do mean 'range disparity' as in Sum of Range. Edited to say 'We hypothesise that the increase in spread of taxa in morphospace (sum of range disparity),...'

Line 312 (now L301) As an aside, that seems extremely unlikely

We do say changes in dietary ecology *may* have been delayed until the Eocene.

L323 (now L312) OK, so what you're saying is that Paleocene mammals occupied a larger region of tarsal morphospace than Cretaceous mammals, but the density of the occupied morphospace (variance) is comparable. That is really not coming through in the earlier

discussion. In particular, I'm really having trouble making sense of lines 305-308 in this context.

Correct. We're not quite sure how to edit this. We describe the disparity plots in the results and have gone through and discussed them in the discussion. You are correct in your summary of the range/variance disparity so we've got our point across... If you have a specific place in the text where it would be useful to edit in the phrasing, then we're open to suggestions.

L327 I'm not sure this is an appropriate use of the DFA loadings.

Addressed above.

Line 330 (now L319) This is really awkward. I think you're saying that Paleocene mammals had different morphologies than extant mammals with similar locomotor ecologies. If that's true, it really undermines the value of the DFA classification.

We're saying some Paleocene mammals fall on the periphery of the extant data, not all. We've edited the language to provide clarity and the following sentence urges caution when considering the locomotor predictions.

L332 (now L321) If the results of the DA aren't reliable, it undermines both the ancillary analyses based on the DA (e.g., the DA disparity analysis) and the broader conclusions about Paleocene mammal locomotor evolution that rely on the DA.

The RDA results are unreliable, they're predictions and should be taken in conjunction with the PC and DF morphospaces and the posteriors derived from the RDA.

The disparity analyses on the DF morphospaces are not affected by the results of the RDA. They are two independent methods.

The broader conclusions of our paper are not reliant on the RDA predictions. We discuss semi-fossoriality/terrestriality as potentially beneficial to surviving the extinction and provide clear caveats to our hypotheses (i.e. some Paleocene species may have exhibited behaviours most similar to extant semi-fossors) but weren't necessarily semi-fossorial.

L336 (now L326) Something more comprehensive would be appropriate. In the examples cited, you essentially seem willing to go to the mat to defend a the locomotor classification of *Purgatorius*, which is known from tarsals but not other postcrania. For *Ectoganus*, which is more completely known, you defer to a more holistic assessment. For other fossil taxa documented by non-tarsal material, how do your classifications compare to existing assessments? This would be at least a limited test of the reliability of the DFA classifications.

We're not arguing for a semi-fossorial classification for *Purgatorius* but trying to illuminate why it is being classified as such based on its tarsal morphology.

We have added a total evidence summary table to the supplement.

L341 (now L331) I don't think the hedging is helpful. It makes this paragraph very difficult to read, and it indicates a lack of confidence in your analyses.

We ‘hedging’ our statements regarding the results of a *predictive* model that is specifically about locomotor categories, one small part of our paper. Surely, given the other comments in this review, being more cautious is advisable? We have confidence in the predictive model We do have confidence that, in combination with each, the methods and analyses are an appropriate treatment of the data currently available. We have provided hypotheses as to why there are so many Paleocene semi-fossors/terrestrialists and we think these hold based on current understanding.

L345 (now L340) This digression doesn't need to be here.

We think it is an important distinction to make given the prevalence of the term ‘primitive eutherian condition’ in the literature and the controversy regarding the time of eutherian vs. placental divergence.

L350 (now L345) Then why did the model classify them as semifossors?

The centroid of the Paleocene semi-fossors is significantly separated from the centroid of the extant semi-fossors but relative to the other extant locomotor groups the Paleocene species classified as semi-fossorial are most similar to extant semi-fossors. The extant model shows extant semi-fossors can be distinguished by tarsal measures, a substantial of Paleocene species share these traits but at the same time are differentiated from extant semi-fossors by the fact they possess highly mobile tarsal articulations.

L374 (now L371) You just said their locomotor strategies are not easily compared to extant mammals.

Deleted ‘with similar locomotor strategies

L382 (now 379) Why didn't they survive in areas where dense tropical forest persisted? Why are extant mammals inhabiting dense tropical forest more gracile?

We don't know. Maybe some Paleocene lineages did survive longer in refuges but there isn't enough data to test this. There are other factors at play too – immigration, dietary shifts associated with new environments and neurosensory changes.

Are tropical mammals more gracile? Compared to Paleocene species, yes but compared to other extant placentals, I'm not sure. Extant gracile tropical forest mammals could have evolved from a more gracile open-habitat ancestor and migrated into forested regions.

Line 400 (now L397) What constitutes a basic placental bauplan?

Something similar that outline in O'Leary et al. 2013. We have added this reference to the text.

Responses to comments in Supplement PDF:

In reference to locomotor allocations: This needs to be more rigorously documented. For instance, why is *Atilax* classified as terrestrial? It falls out in a region of discriminant morphospace otherwise occupied by semiaquatic mammals, which seems like it might be a reasonable classification of the marsh mongoose.

In text on Purgatorius: Redundant

Deleted.

Can you back this up with anything more rigorous? Intuition is not a substitute for data.

Have added a reference to statement and edited language.

Tarsal features or just tarsals?

Tarsals will do

Can you describe specific features that support this inference? Chester et al. illustrate all of the archontan taxa analyzed in this manuscript side by side. I'm not seeing any obvious features of the tarsus of Purgatorius that would indicate more mobility and loading laterally than in the other taxa.

The mediolaterally broad astragalar body with a well-developed, fibular facet and large ectal and sustentacular facets facilitating a greater arc of rotation with relatively taller trochlear keels (compared to other euarchontans) stabilising against over-rotation. Broad and long sustentacular in particular would have permitted a high degree of rotation

Chester writes:

Purgatorius has a saddle-shaped astragalar ectal facet that articulates with and rotates along a longer, moderately proximodistally aligned calcaneal ectal facet (Fig. 1). This morphology suggests a pronounced capacity for inversion and eversion of the foot, which is supported further by the presence of a well-developed distal calcaneal sustentacular facet and a distally extensive astragalar sustentacular facet that contacts the navicular facet. These distal articular regions would have come into close contact only during strong inversion of the foot. Such movements are facilitated further at the transverse tarsal joint of Purgatorius by the rounded, concave, gliding articulation of the calcaneocuboid facet and its fairly transverse orientation and by the pronounced, rounded navicular facet on the medial side of the astragalar head.

Chester et al. specifically state that inversion/eversion was occurring at the subastragalar joint. There is no mention of inversion at the crurotarsal joint. It seems like you're inferring mobility at the crurotarsal joint based on DF3 value, not based on the morphology of the tarsus.

DF3 captures the movement between the astragalus and calcaneum (=inversion and eversion) as well. This is described in the description of the axis in the supplement.

Purgatorius has a taller lateral keel allowing a great arc of rotation of the lateral side of the crus which would cause the pes to invert (independently from movement between the astragalus and calcaneum) and in a well-stabilised manner – stabilised by large fibular. This is more like extant semi-fossors than arborealists which either quite rounded trochlear keels or more symmetrical keels.

The comments here are reasonable, although some ground sloths appear to have combined a hook on the tuber with a plantigrade morphology.

Do you have more information on this? How are the ground sloths interpreted to be plantigrade? Some armadillos have enlarged medially inflected tuber hooks – could be a xenarthran morphology.

One concern is the contrast between *Ectoganus* and *Purgatorius*. In the case of *Purgatorius*, you have tarsals but no other postcrania, so you accept the DFA results even though they differ from prior assessments. In the case of *Ectoganus*, non-tarsal anatomy constrains your functional assessment, so you're effectively rejecting the results of the DFA.

We're not rejecting the results, we're explaining why we're getting the results we are and why they should be taken in consideration of the other information available – i.e. they should not be dismissed because they're different to previous interpretations, nor accepted without consideration. We have now provided a summary table in the supplement for all the fossil taxa.

Honestly, the classification of *Pachyaena* is much more worrisome. Thewissen was discussing postcrania of a late Paleocene species of *Dissacus* that has a more specialized morphology than *D. navajovius*. O'Leary and Rose (1995), on the other hand, concluded that all species of *Pachyaena*, including *P. gracilis*, were cursorial.

We have provided discussion on *Pachyaena* in the above reponse text and now also provide a supplementary total evidence table for the fossil taxa (Table S15). The addition of marsupials has also changed the classification for some 'worrisome' taxa.

Appendix G

We thank the editor and the reviewer for their continued feedback and attention to detail, and their patience with the revision process. We feel that this last round of edits has made the paper substantially stronger. In contrast to previous rounds of major revision, these comments were more moderate, and only one of our reviewers still had outstanding questions. The main remaining issues from the editor and reviewer mostly had to do with the precision of our language, particularly how our title and portions of our discussion implied that Paleocene mammals had unique locomotor habits (when our results are more nuanced), and in how we discussed the different morphospaces and their implications. The reviewer also had an important methodological comment about outliers in morphospace, and suggestions for improving our figures. Additional comments were minor edits to the text. Specific details are provided below.

Associate Editor Board Member

Comments to Author:

Thank you for revising and resubmitting this interesting work on tarsal morphology in extinct and extant mammals. This is the third submission of the paper, and again represents an improvement on the previous iteration. In particular, expansion of the extant marsupial data set deepens the analysis and much of the discussion around the nature and significance of the findings are now much improved. However, the reviewer still has, in my opinion, a number of legitimate concerns that need to be addressed, but these are now probably less than “major” in scale. The first couple of points raised by the reviewer overlap to an extent with comments I raised about the previous version, which the authors didn’t feel were entirely clear. The title of your paper is “Unique locomotor habits in Palaeocene mammals” so what, exactly, is the unique “habit(s)” (which I interpret to be a synonym for habitat or locomotor ecology) that they had? What unique habit/habitat/locomotor ecology were they living in that no other mammal has ever done before or since? In the current version, this very, very high predictive bar is almost solely set by the title but in earlier versions it was also present in places within the text too. But the analysis did not and does not reach that bar because you are **directly analysing morphology in an attempt to infer function and then from that infer habit/ecology**; you’re not directly analysing mechanics and you’re not directly analysing functional performance within an ecology or habitat (because you can’t in fossils). My previous comment that there was “a lack of a clear translation to differences/similarities in function and locomotor ecologies” largely reflects the fact that (mostly, but certainly not always) you had not over-interpreted your results in an attempt to reach that bar: you mostly discuss MORPHOLOGICAL diversity/disparity patterns, and not functional/ecological ones. In this version I read through the discussion and found only a few sentences that explicitly talk about locomotor habit/ecology rather than morphology (prior to the final conclusion/summary paragraph). I think this is fine as long as that is where the expectation is set throughout, and I agree wholeheartedly that the morphological analyses and results are novel and very interesting by themselves, but I felt previously that you (by yourselves) were setting the bar too high. In the current version **the title and the final sentence of the paper (“thus indicating distinctive locomotor habits and unexpected ecomorphological diversity”) are still guilty of this in my opinion**. These issues can be addressed by simply rewording, and, for example by separating the solid aspect of your data (morphology) from the inference-based aspect of your work when you set the study up:

“Here, we investigate the locomotor ecology of Paleocene eutherians, in comparison to a sample of Cretaceous cladotherians and extant therian mammals.” You could reword to say something like “investigate diversity of morphology in X and Y ways and subsequently these analyses to assess it’s relationship to locomotor ecology.”

Something like that, so that you are explicitly separate the morphological aspect from the “bigger picture” functional and/or ecological aspects then what you have achieved will be clearer and more fairly represented. I think this approach combined with the suggestions in the reviewers comment 1 would clear these issues up.

Thank you for clarifying on your previous comments and we agree with everything said here. As such we have changed the title of the paper and also edited the language in the manuscript to more clearly state when we’re referring to the morphospaces, disparity results, RDA etc. (in line with some comments raised by reviewer 2) and how they reflect locomotor behaviour.

Referee: 2

Comments to the Author(s).

Overall, this is a much improved draft of this manuscript. The addition of extant marsupials to the modern dataset has substantially improved the analysis, and I’m now comfortable that DFA provides insights into the locomotor behavior of Paleocene mammals. I agree with the authors that its not necessary that every locomotor classification be perfectly aligned with other data. My major concern, that too many taxa were falling well outside the DFA morphospace defined by extant placentals, has been addressed. I think this is now a reasonable dataset to investigate patterns of locomotor diversification. There are still issues to be addressed. There are several broad issues as well as smaller issues presented with line numbers in lieu of attaching an annotated copy.

1. The broad conclusion that correlation between the tarsal morphology of extant mammals and locomotor behavior can be used to infer patterns behavior in Paleocene mammals is sound. However, this conclusion is repeatedly undercut (e.g., **lines 321-323, 346-348, 365-368**) by statements that the relationship between morphology and locomotion in Paleocene mammals may not have been the same as in extant mammals. You don’t clearly explain where this skepticism about your results is coming from. As a reader, I’m left with the impression that you don’t have much confidence in your analyses.

If I’m not mistaken, I think the basic issue you’re trying to confront is as follows: extant semifossorial mammals have robust tarsals; Paleocene mammals have robust tarsals across the board; the robusticity of Paleocene mammal tarsals could be predisposing them to be misclassified as semifossorial (or in the semifossorial-adjacent area of terrestriality). **If that is the case, it needs to be addressed clearly and directly.** State the problem and the evidence for it, explain why you don’t (or do) think it undermines your conclusions, and describe potential future tests that could address the issue.

We acknowledge this concern and have edited sections of the discussion to make our statements regarding our results and the different hypothesis clearer. It’s not that we don’t have confidence

in our analyses, it's more that we want to acknowledge and account for the uncertainty of the predictive methods.

The sections in question are:

L321: This distribution is explained by Paleocene species co-opting unusual morphologies compared to the extant sample to meet the functional requirements of a locomotion.

L346: We emphasise that the tarsal morphologies of the Paleocene taxa might also have been convergent with extant terrestrialists and semi-fossors to an extent, and the animals may not have necessarily exhibited similar behaviours

We have edited the paragraph containing the above two sections to more clearly state the results and then present the different hypotheses (section now beginning at L342)

Regarding what was L365:

However, the distribution of taxa in morphospace shows some Paleocene species exhibit tarsal morphologies intimating at diversification towards different locomotor behaviours than their classification implies although still differentiated from extant species by their overall robustness and lack of tarsal stability

We have deleted this entire paragraph as it is no longer relevant.

2. I agree with your argument that individual classifications of extinct taxa don't need to be overanalyzed and that the overall pattern is more important. However, I think you could strengthen your presentation by **briefly discussing the outlier taxa**. There are three taxa that are outliers along discriminant axis 1 (*Orthaspidotherium*, *Pleuraspidotherium*, and *Tribosphenomys*) and three that are outliers along axis 2 (*Afrodon*, *Bustylus*, and *Carpolestes*). It would be worth noting that two of the three in each category are closely related, since it makes the overall proportion of outliers less dramatic. You should also consider excluding or reanalyzing another outlier, **Tribosphenomys**. While the other outlier taxa have genuinely weird tarsals, the tarsals illustrated for *Tribosphenomys* don't seem to be unusual. Its surprising that its so divergent. I think the problem may be the source of data. Based on data set S1, *Tribosphenomys* was scored from Meng and Wyss (2001). However, the calcaneal figures in Meng and Wyss are not oriented in a way that should allow them to be used for this study. The dorsal and ventral views are actually dorsomedial and ventrolateral views, respectively. Among other things, you shouldn't be able to see the entire plantar tubercle in dorsal view. At least 4 measurements used in the study (C3, C4, C5 and C12) cannot be reliably measured from the photos in Meng and Wyss (2001). I would suggest replacing the calcaneal data from Meng and Wyss with measurements taken from Fostowicz-Frelik et al. (2018) or excluding *Tribosphenomys* from the analysis.

This is a good catch. We remeasured *Tribosphenomys* and there was enough of a difference that we decided to rerun the entire analysis out of extreme caution. The position of *Tribosphenomys* has changed slightly and the RDA predictions are slightly different, however, the overall results are the same. We thank the reviewer for the attention to detail to notice this issue.

We have also added Mahalanobis distances that quantifiably assess divergent taxa and identify statistical outliers. These have been provided and discussed in the supplement.

3. Your use of the term morphospace can be confusing (e.g., line 375). It's not always clear if it applies to the overall morphospace captured by the PCA or the functional morphospace defined by the DFA or both. **Defining and using distinct terms to discuss those would make the discussion easier to follow.**

Also a good catch. We have specified when discussing PC versus DF morphospace or both.

4. My previous criticism of how extant locomotor classifications were determined still needs to be addressed. I specifically requested that this be documented in the manuscript. If Powell (1981) was used to determine the locomotor category of *Pekania*, it needs to be cited somewhere. Addition of a column to **Dataset S1** after “**LOCOMOTOR GROUP**” with **references for each modern taxon** would be ideal.

These have now been provided (cited in Dataset S1 with full references in the supplementary text file).

5. The addition of more taxa and their effect on the discriminant analysis is making Fig. 1b, d, and f and Fig. S4b, d, and f difficult to read. In particular, the extinct taxa are obscuring the pattern of the extant taxa. I would suggest adding either a column to Fig. S4 or a separate figure showing all of the extant taxa without the fossil taxa included.

We have added another figure into supplement with just the extant taxa differentiated into placentals and marsupials

6. Figs. S12d-e and S13d-e are impossible to read given fonts and the density of text, markers, and lines. These graphs (especially the “e” portions) need to be presented at a larger scale so they're legible.

We have now used taxon identification numbers (corresponding to numbers in Dataset S1) to identify datapoints in these figures.

Additional comments:

Lines 155-157: In the previous draft, these distinctions also applied to placentals. Is this no longer true? [in reference to morphospace position of Paleocene taxa compared to extant placentals and marsupials]

The construction of the sentences is different, and the distinctions are not quantified in these summary sentences – they're all relative.

In previous version, we wrote:

“Relative to the extant sample, Palaeocene mammals are widely dispersed in PC locomotor morphospace, possessing a comparatively more robust and mobile astragalocalcaneal

articulation, [**Palaeocene mammals possess**] a variably mobile to more stable cruropedal joint, and a range of relative ectal and sustentacular facet proportions (Figs. 3-4).” [**as do placentals but they’re still separated from placentals by robustness and mobility as the astralocalcaneal joint**]

With the addition of marsupials, we wrote:

Relative to the extant sample, Paleocene species are widely dispersed in PC morphospace (Figs. 1-2). They are differentiated from extant placental mammals in that they possess a comparatively more robust and mobile astragalocalcaneal articulation (Figs. SX, SX, SX). In this regard, they are more similar to our extant marsupial sample; however, they are differentiated from marsupials by possessing a variably mobile to more stable cruropedal joint and a range of relative ectal and sustentacular facet proportions (Figs. 1-2, SX, SX, SX) [**which is more like the placental sample**].

We have slightly tweaked this section in the current submission but do not think there is much to edit here to reconcile the two versions given how the data have changed.

Line 156: Remove “,” after “joint”. There are only two items in the list.

Edited

Lines 213-215: I think this should be reworded for emphasis. What the LDA is indicating is that, despite having distinctive morphologies, most fossil taxa appear to be exploiting locomotor strategies similar to extant mammals.

Now L240 and edited to say:

‘Despite their distinctiveness in PC morphospace, the LDA visualisation and RDA classification show that the fossil species share similar tarsal functions to extant mammals in terms of stance and movement and loading through the pes, but are often exploiting locomotor strategies using different combinations of morphologies distinguished by their underlying robustness’

Lines 249-250: The statement described here should be added.

Oops

Have followed through and added a brief sentence in the main MS.

Lines 309-311: I don’t think this is the correct interpretation of the pattern you’ve observed. Putting the two values (range and variance) together, they indicate that Paleocene mammals occupied the same area of morphospace as Cretaceous mammals (range), but at a greater density (variance). That’s consistent with greater numerical diversity, which makes sense given that you’ve sampled more Paleocene taxa. I don’t think it implies that Cretaceous mammals were exploiting niches in different ways.

Data have been bootstrapped and rarefied, so the influence of sample size is lessened. The Cretaceous taxa are inferred to be terrestrial plus one semi-fossorial but are still spread out in morphospace. There is tarsal diversity (which may also be attributable, in part, to phylogeny or

age, as noted). We feel that our original statement holds, but we have added a statement to remind the reader that the data have been corrected for sampling bias.

Now reads: We note that although the data presented here has been correct for sampling bias, there may be a phylogenetic and temporal component to this diversity but the lack of fossil material limits further investigation at present.

Lines 319-321: Can you document this connection? Right now, this reads as speculation. It seems possible, but some evidence should be provided. In particular, do Paleocene taxa with strong negative scores along PC1 and PC2 (further from the modern sample) tend to be on the periphery of the DFA functions?

The multivariate analyses document this. The PCA shows how the taxa differ, with the Paleocene species being distinguished from the extant (placentals) by a more robust tarsus and a more mobile astragalocalcaneal joint. The DFA shows how the Paleocene taxa are similar to the extant sample by arraying them over the extant taxa when grouped by locomotor mode. Even so, some Paleocene taxa are still divergent.

That said a divergent position in PC morphospace does not have to correspond to a divergent position in DF morphospace given that they are different methods (most importantly, that DF includes information on pre-assigned locomotor categories for the extant species, whereas PC does not) so a comparison between PC1 (robustness/mobility) versus DF1 (stance) isn't necessarily going to show what you're asking

Instead, we have provided Mahalanobis distances/scaled Euclidean distances. This shows something interesting: Paleocene taxa which are statistical outliers in PC morphospace are also outliers in DF morphospace. We thank the reviewer for inspiring us to do this, and feels that it clarifies the issue they are raising.

Lines 333-335: Simplify, change:

“these ground-dwelling locomotor strategies (or morphologies most similar to those exhibited by extant mammals that exploit these locomotor strategies) were key adaptive traits for”

to:

"ground-dwelling locomotor strategies were key to"

I don't think the qualifications are needed here, they just make the statement more difficult to parse.

Edited

Lines 342-342: Change:

"semi-fossorial(-like) bauplan (retaining some traits of that morphotype)"

to:

"ground dwelling bauplan"

Again, the qualifications aren't necessary.

Edited

Lines 349-350: I renew my objection to including this statement despite your response. This is a completely circular argument. You use discriminant scores to classify taxa as either terrestrial or semifossorial. You then take the discriminant scores for taxa assigned to each group and use them to demonstrate that the mean values are significantly different. To the extent that this says anything at all, it's that the discriminant functions do a good job of reliably separating functional groups.

We think the reviewer has misunderstood. It's comparing like to like, extant terrestrialists to extinct ones for two locomotor groups.

We have edited and restructured paragraph/section to make it more clear what we're saying (now L381).

Line 350: It's not clear what scenario the phrase "In this scenario" is referring to. It doesn't make sense with regard to the previous sentence.

Have edited to say:

In the scenario that Paleocene species exhibit convergent morphologies with extant semi-fossors and terrestrialists, it might have been the robust morphologies themselves and associated reliance on soft tissues for joint stabilisation that promoted survival and diversification around the extinction.

Lines 353-354: Please discuss how this hypothesis could be tested. If you're correct, Paleocene eutherians should be more robust than Paleocene metatherians or multituberculates, which would account for their divergent fates. More robust placental subclades should exhibit differential survival in the Paleocene. More robust mammals should show greater survival across the K/T boundary. I doubt any of these predictions can currently be tested, but **you should indicate what data could be used to test this hypothesis in the future.**

R2: 'If you're correct, Paleocene eutherians should be more robust than Paleocene metatherians or multituberculates'

This statement makes the assumption that eutherians were more successful than metatherians and multituberculates because of robustness, which isn't what we're saying. We don't know this and haven't tested it. We're hypothesising why Paleocene eutherians are different/robust compared to extant eutherians. Postcranial robustness plus other factors (diet, body size, physiological factors – reproductive strategies, growth strategies, greater metabolic efficiency) allowed placentals to do well in their own right. Maybe Paleocene metatherians were more robust than extant ones and Paleocene multituberculates were more robust than Cretaceous antecedents as well—but our dataset cannot test these hypotheses, which are outside of the scope of this paper and would require a much more expansive dataset. It is a great idea for future follow-up work.

Have edited this section to address comment (L381 onwards)

Line 358: Change "given rise to" to "have favored development of"

Edited

Lines 370-373: Aside from Adapisoriculidae and Pleuraspidotheriidae, the fact that Paleocene mammals fall within or very close to the functional morphospace defined by the DFA suggests that their locomotor strategies are easily compared to extant mammals.

This section has been deleted

Line 374: Change “amount” to “area”

Edited

Lines 374-375: Again, this is one place where discriminating between overall and functional morphospace would be very helpful. It's not clear if you're referring to overall tarsal morphospace or functional morphospace.

Have now distinguished morphospaces throughout the manuscript. Thanks again for pointing out where our previous language was unclear.

Line 376: Delete “warmer and”. It's surprising that Paleocene mammals showed so much locomotor diversity because the global environment was more homogenous, not because it was warmer. The warmer climate may have driven the greater homogeneity, but the mere fact that it was warmer isn't why it's surprising that Paleocene mammals showed so much locomotor diversity.

Edited

Lines 378-379: Delete “which were nonetheless punctuated by hyperthermals”. Again, this is distracting from your point.

Edited

Lines 385-394: I don't think this needs to be here. If you were focused on the specific locomotor reconstructions of individual taxa, this would be appropriate. Since you're looking at broader patterns and trying to deemphasize individual taxa, this just breaks the flow of the discussion.

Agreed and deleted. The point is made in the supplement anyway.

Lines 397-398: Again, aside from Pleuraspidotheriidae and Adapisoriculidae, where is the evidence for distinctiveness in locomotion? Paleocene mammals certainly occupy a distinctive area of tarsal morphospace, but functionally they seem comparable to extant mammals given the amount of overlap in the DFA.

We have now reframed the way in which we discuss our results and how we describe morphological distinctiveness and how this related to locomotor diversity.

SI, page 24, bottom, “Higher DF3 scores are associated with morphologies that permit a greater degree of rotational movement between the astragalus and calcaneum”: Wouldn't movement between the astragalus and calcaneum facilitate inversion, which is associated with climbing? Yes, which is what we're saying.

Artiodactyls are really the exception here where the subastragalar joint is involved in terrestrial locomotion. We're not sure we follow. Perhaps R2 is looking at PC morphospaces where artiodactyls do have high PC3 scores? More typically, terrestrial taxa lock down the subastragalar joint and its arboreal/scansorial taxa that have mobility here. Which is what DF3 shows, artiodactyls have midrange DF3 values associated with limited inversion and eversion.